# Computing Optimal Equilibria and Mechanisms via Learning in Zero-Sum Extensive-Form Games

**Brian Hu Zhang**[*]
Carnegie Mellon University
bhzhang@cs.cmu.edu

**Gabriele Farina**[*]
MIT
gfarina@mit.edu

**Ioannis Anagnostides**
Carnegie Mellon University
ianagnos@cs.cmu.edu

**Federico Cacciamani**
DEIB, Politecnico di Milano
federico.cacciamani@polimi.it

**Stephen McAleer**
Carnegie Mellon University
smcaleer@cs.cmu.edu

**Andreas Haupt**
MIT
haupt@mit.edu

**Andrea Celli**
Bocconi University
andrea.celli2@unibocconi.it

**Nicola Gatti**
DEIB, Politecnico di Milano
nicola.gatti@polimi.it

**Vincent Conitzer**
Carnegie Mellon University
conitzer@cs.cmu.edu

**Tuomas Sandholm**
Carnegie Mellon University
Strategic Machine, Inc.
Strategy Robot, Inc.
Optimized Markets, Inc.
sandholm@cs.cmu.edu

## Abstract

We introduce a new approach for *computing* optimal equilibria and mechanisms via learning in games. It applies to extensive-form settings with any number of players, including mechanism design, information design, and solution concepts such as correlated, communication, and certification equilibria. We observe that *optimal* equilibria are minimax equilibrium strategies of a player in an extensive-form zero-sum game. This reformulation allows us to apply techniques for learning in zero-sum games, yielding the first learning dynamics that converge to optimal equilibria, not only in empirical averages, but also in iterates. We demonstrate the practical scalability and flexibility of our approach by attaining state-of-the-art performance in benchmark tabular games, and by computing an optimal mechanism for a sequential auction design problem using deep reinforcement learning.

## 1 Introduction

What does it mean to *solve* a game? This is one of the central questions addressed in game theory, leading to a variety of different solution concepts. Perhaps first and foremost, there are various notions of *equilibrium*, strategically stable points from which no rational individual would be inclined to deviate. But is it enough to compute, or indeed *learn*, just any one equilibrium of a game? In two-player zero-sum games, one can make a convincing argument that a single equilibrium in fact constitutes a complete solution to the game, based on the celebrated minimax theorem of von Neumann [97]. Indeed, approaches based on computing minimax equilibria in

---

[*]Equal contribution.

37th Conference on Neural Information Processing Systems (NeurIPS 2023).

two-player zero-sum games have enjoyed a remarkable success in solving major AI challenges, exemplified by the recent development of superhuman poker AI agents [10, 11].

However, in general-sum games it becomes harder to argue that *any* equilibrium constitutes a complete solution. Indeed, one equilibrium can offer vastly different payoffs to the players than another. Further, if a player acts according to one equilibrium and another player according to a different one, the result may not be an equilibrium at all, resulting in a true *equilibrium selection problem*. In this paper, therefore, we focus on computing an *optimal* equilibrium, that is, one that maximizes a given linear objective within the space of equilibria. There are various advantages to this approach. First, in many contexts, we would simply prefer to have an equilibrium that maximizes, say, the sum of the players' utilities—and by computing such an equilibrium we also automatically avoid Pareto-dominated equilibria. Second, it can mitigate the equilibrium selection problem: if there is a convention that we always pursue an equilibrium that maximizes social welfare, this reduces the risk that players end up playing according to different equilibria. Third, if one has little control over how the game will be played but cares about its outcomes, one may like to understand the space of all equilibria. In general, a complete picture of this space can be elusive, in part because a game can have exponentially many equilibria; but computing extreme equilibria in many directions—say, one that maximizes Player 1's utility—can provide meaningful information about the space of equilibria.

That being said, many techniques that have been successful at computing a single equilibrium do not lend themselves well to computing optimal equilibria. Most notably, while *no-regret* learning dynamics are known to converge to different notions of *correlated equilibria* [49, 38, 39, 47], little is known about the properties of the equilibrium reached. In this paper, therefore, we introduce a new paradigm of learning in games for *computing* optimal equilibria. It applies to extensive-form settings with any number of players, including information design, and solution concepts such as correlated, communication, and certification equilibria. Further, our framework is general enough to also capture optimal mechanism design and optimal incentive design problems in sequential settings.

**Summary of Our Results**  A key insight that underpins our results is that computing *optimal* equilibria in multi-player extensive-form games can be cast via a Lagrangian relaxation as a two-player zero-sum extensive-form game. This unlocks a rich technology, both theoretical and experimental, developed for computing minimax equilibria for the more challenging—and much less understood—problem of computing optimal equilibria. In particular, building on the framework of Zhang and Sandholm [100], our reduction lends itself to mechanism design and information design, as well as an entire hierarchy of equilibrium concepts, including *normal-form coarse correlated equilibria (NFCCE)* [79], *extensive-form coarse correlated equilibria (EFCCE)* [31], *extensive-form correlated equilibria (EFCE)* [98], *communication equilibria (COMM)* [36], and *certification equilibria (CERT)* [37]. In fact, for communication and certification equilibria, our framework leads to the first learning-based algorithms for computing them, addressing a question left open by Zhang and Sandholm [100] (*cf.* [40], discussed in Appendix B).

We thus focus on computing an optimal equilibrium by employing regret minimization techniques in order to solve the induced bilinear saddle-point problem. Such considerations are motivated in part by the remarkable success of no-regret algorithms for computing minimax equilibria in large two-player zero-sum games (*e.g.*, see [10, 11]), which we endeavor to transfer to the problem of computing optimal equilibria in multi-player games.

In this context, we show that employing standard regret minimizers, such as online mirror descent [91] or counterfactual regret minimization [106], leads to a rate of convergence of $T^{-1/4}$ to optimal equilibria by appropriately tuning the magnitude of the Lagrange multipliers (Corollary 3.3). We also leverage the technique of *optimism*, pioneered by Chiang et al. [18], Rakhlin and Sridharan [87] and Syrgkanis et al. [94], to obtain an accelerated $T^{-1/2}$ rate of convergence (Corollary 3.4). These are the first learning dynamics that (provably) converge to optimal equilibria. Our bilinear formulation also allows us to obtain *last-iterate* convergence to optimal equilibria via optimistic gradient descent/ascent (Theorem 3.5), instead of the time-average guarantees traditionally derived within the no-regret framework. As such, we bypass known barriers in the traditional learning paradigm by incorporating an additional player, a *mediator*, into the learning process. Furthermore, we also study an alternative Lagrangian relaxation which, unlike our earlier approach, consists of solving a sequence of zero-sum games (*cf.* [30]). While the latter approach is less natural, we find that it is preferable when used in conjunction with deep RL solvers since it obviates the need for solving games with large reward ranges—a byproduct of employing the natural Lagrangian relaxation.

**Experimental results**   We demonstrate the practical scalability of our approach for computing optimal equilibria and mechanisms. First, we obtain state-of-the-art performance in a suite of 23 different benchmark game instances for seven different equilibrium concepts. Our algorithm significantly outperforms existing LP-based methods, typically by more than one order of magnitude. We also use our algorithm to derive an optimal mechanism for a sequential auction design problem, and we demonstrate that our approach is naturally amenable to modern deep RL techniques.

## 1.1   Related work

In this subsection, we highlight prior research that closely relates to our work. Additional related work is included in Appendix B.

A key reference point is the recent paper of Zhang and Sandholm [100], which presented a unifying framework that enables the computation via linear programming of various mediator-based equilibrium concepts in extensive-form games, including NFCCE, EFCCE, EFCE, COMM, and CERT.[2] Perhaps surprisingly, Zhang et al. [101] demonstrated that computing optimal communication and certification equilibria is possible in time polynomial in the description of the game, establishing a stark dichotomy between the other equilibrium concepts—namely, NFCCE, EFCE, and EFCCE—for which the corresponding problem is NP-hard [98]. In particular, for the latter notions intractability turns out to be driven by the imperfect recall of the mediator [101]. Although imperfect recall induces a computationally hard problem in general from the side of the mediator [19, 59], positive parameterized results have been documented recently in the literature [103].

Our work significantly departs from the framework of Zhang and Sandholm [100] in that we follow a learning-based approach, which has proven to be a particularly favorable avenue in practice; *e.g.*, we refer to [26, 16, 78, 77, 104] for such approaches in the context of computing EFCE. Further, beyond the tabular setting, learning-based frameworks are amenable to modern deep reinforcement learning methods (see [70, 71, 62, 69, 51, 76, 57, 13, 52, 73, 72, 86, 105, 42], and references therein). Most of those techniques have been developed to solve two-player zero-sum games, which provides another crucial motivation for our main reduction. We demonstrate this experimentally in large games in Section 4. For multi-player games, Marris et al. [70] developed a scalable algorithm based on *policy space response oracles (PSRO)* [62] (a deep-reinforcement-learning-based double-oracle technique) that converges to NFC(C)E, but it does not find an optimal equilibrium.

Our research also relates to computational approaches to static auction and mechanism design through deep learning [27, 86]. In particular, similarly to the present paper, Dütting et al. [27] study a Lagrangian relaxation of mechanism design problems. Our approach is significantly more general in that we cover both static and *sequential* auctions, as well as general extensive-form games. Further, as a follow-up, Rahme et al. [86] frame the Lagrangian relaxation as a two-player game, which, however, is not zero-sum, thereby not enabling leveraging the tools known for solving zero-sum games. Finally, in a companion paper [102], we show how the framework developed in this work can be used to *steer* no-regret learners to optimal equilibria via nonnegative vanishing payments.

## 2   Preliminaries

We adopt the general framework of *mediator-augmented games* of Zhang and Sandholm [100] to define our class of instances. At a high level, a mediator-augmented game explicitly incorporates an additional player, the *mediator*, who can exchange messages with the players and issue action recommendations; different assumptions on the power of the mediator and the players' strategy sets induce different equilibrium concepts, as we clarify for completeness in Appendix A.

**Definition 2.1.** A *mediator-augmented, extensive-form game* $\Gamma$ has the following components:

1. a set of players, identified with the set of integers $[\![n]\!] := \{1, \ldots, n\}$. We will use $-i$, for $i \in [\![n]\!]$, to denote all players except $i$;
2. a directed tree $H$ of *histories* or *nodes*, whose root is denoted $\varnothing$. The edges of $H$ are labeled with *actions*. The set of actions legal at $h$ is denoted $A_h$. Leaf nodes of $H$ are called *terminal*, and the set of such leaves is denoted by $Z$;

---

[2]Notably missing from this list is the *normal-form correlated equilibrium (NFCE)*, the complexity status of which (in extensive-form games) is a long-standing open problem.

3. a partition $H \setminus Z = H_{\mathsf{C}} \sqcup H_0 \sqcup H_1 \sqcup \cdots \sqcup H_n$, where $H_i$ is the set of nodes at which $i$ takes an action, and $\mathsf{C}$ and $0$ denote chance and the mediator, respectively;
4. for each agent[3] $i \in [\![n]\!] \cup \{0\}$, a partition $\mathcal{I}_i$ of $i$'s decision nodes $H_i$ into *information sets*. Every node in a given information set $I$ must have the same set of legal actions, denoted by $A_I$;
5. for each agent $i$, a *utility function* $u_i : Z \to \mathbb{R}$; and
6. for each chance node $h \in H_{\mathsf{C}}$, a fixed probability distribution $c(\cdot\,|h)$ over $A_h$.

To further clarify this definition, in Appendix A we provide two concrete illustrative examples: a single-item auction and a welfare-optimal correlated equilibrium in normal-form games.

At a node $h \in H$, the *sequence* $\sigma_i(h)$ of an agent $i$ is the set of all information sets encountered by agent $i$, and the actions played at such information sets, along the $\varnothing \to h$ path, excluding at $h$ itself. An agent has *perfect recall* if $\sigma_i(h) = \sigma_i(h')$ for all $h, h'$ in the same infoset. We will use $\Sigma_i := \{\sigma_i(z) : z \in Z\}$ to denote the set of all sequences of player $i$ that correspond to terminal nodes. We will assume that all *players* have perfect recall, though the *mediator* may not.[4]

A *pure strategy* of agent $i$ is a choice of one action in $A_I$ for each information set $I \in \mathcal{I}_i$. The *sequence form* of a pure strategy is the vector $\boldsymbol{x}_i \in \{0,1\}^{\Sigma_i}$ given by $\boldsymbol{x}_i[\sigma] = 1$ if and only if $i$ plays every action on the path from the root to sequence $\sigma \in \Sigma_i$. We will use the shorthand $\boldsymbol{x}_i[z] = \boldsymbol{x}_i[\sigma_i(z)]$. A *mixed strategy* is a distribution over pure strategies, and the sequence form of a mixed strategy is the corresponding convex combination $\boldsymbol{x}_i \in [0,1]^{\Sigma_i}$. We will use $X_i$ to denote the polytope of sequence-form mixed strategies of player $i$, and use $\Xi$ to denote the polytope of sequence-form mixed strategies of the mediator.

For a fixed $\boldsymbol{\mu} \in \Xi$, we will say that $(\boldsymbol{\mu}, \boldsymbol{x})$ is an *equilibrium* of $\Gamma$ if, for each *player* $i$, $\boldsymbol{x}_i$ is a best response to $(\boldsymbol{\mu}, \boldsymbol{x}_{-i})$, that is, $\max_{\boldsymbol{x}_i' \in X_i} u_i(\boldsymbol{\mu}, \boldsymbol{x}_i', \boldsymbol{x}_{-i}) \leq u_i(\boldsymbol{\mu}, \boldsymbol{x}_i, \boldsymbol{x}_{-i})$. We do *not* require that the mediator's strategy $\boldsymbol{\mu}$ is a best response. As such, the mediator has the power to commit to its strategy. The goal in this paper will generally be to reach an *optimal (Stackelberg) equilibrium*, that is, an equilibrium $(\boldsymbol{\mu}, \boldsymbol{x})$ maximizing the mediator utility $u_0(\boldsymbol{\mu}, \boldsymbol{x})$. We will use $u_0^*$ to denote the value for the mediator in an optimal equilibrium.

**Revelation principle** The *revelation principle* allows us, without loss of generality, to restrict our attention to equilibria where each player is playing some fixed pure strategy $\boldsymbol{d}_i \in X_i$.

**Definition 2.2.** The game $\Gamma$ satisfies the *revelation principle* if there exists a *direct* pure strategy profile $\boldsymbol{d} = (\boldsymbol{d}_1, \ldots, \boldsymbol{d}_n)$ for the players such that, for all strategy profiles $(\boldsymbol{\mu}, \boldsymbol{x})$ for all players including the mediator, there exists a mediator strategy $\boldsymbol{\mu}' \in \Xi$ and functions $f_i : X_i \to X_i$ for each player $i$ such that:

1. $f_i(\boldsymbol{d}_i) = \boldsymbol{x}_i$, and

2. $u_j(\boldsymbol{\mu}', \boldsymbol{x}_i', \boldsymbol{d}_{-i}) = u_j(\boldsymbol{\mu}, f_i(\boldsymbol{x}_i'), \boldsymbol{x}_{-i})$ for all $\boldsymbol{x}_i' \in X_i$, and *agents* $j \in [\![n]\!] \cup \{0\}$.

The function $f_i$ in the definition of the revelation principle can be seen as a *simulator* for Player $i$: it tells Player $i$ that playing $\boldsymbol{x}_i'$ if other players play $(\boldsymbol{\mu}, \boldsymbol{d}_{-i})$ would be equivalent, in terms of all the payoffs to all agents (including the mediator), to playing $f(\boldsymbol{x}_i')$ if other agents play $(\boldsymbol{\mu}, \boldsymbol{x}_{-i})$. It follows immediately from the definition that if $(\boldsymbol{\mu}, \boldsymbol{x})$ is an $\varepsilon$-equilibrium, then so is $(\boldsymbol{\mu}', \boldsymbol{d})$—that is, every equilibrium is payoff-equivalent to a direct equilibrium.

The revelation principle applies and covers many cases of interest in economics and game theory. For example, in (single-stage or dynamic) mechanism design, the direct strategy $\boldsymbol{d}_i$ of each player is to report all information truthfully, and the revelation principle guarantees that for all non-truthful mechanisms $(\boldsymbol{\mu}, \boldsymbol{x})$ there exists a truthful mechanism $(\boldsymbol{\mu}', \boldsymbol{d})$ with the same utilities for all players.[5] For correlated equilibrium, the direct strategy $\boldsymbol{d}_i$ consists of obeying all (potentially randomized) recommendations that the mediator gives, and the revelation principle states that we can, without loss of generality, consider only correlated equilibria where the signals given to the players are what actions they should play. In both these cases (and indeed in general for the notions we consider in this

---

[3]We will use *agent* to mean either a player or the mediator.

[4]Following the framework of Zhang and Sandholm [100], allowing the mediator to have imperfect recall will allow us to automatically capture optimal correlation.

[5]In a mechanism design context, a strategy for the mediator $\boldsymbol{\mu}$ induces a mechanism; here we slightly abuse terminology by referring to $(\boldsymbol{\mu}, \boldsymbol{d})$ also as a mechanism.

paper), it is therefore trivial to specify the direct strategies $\boldsymbol{d}$ without any computational overhead. Indeed, we will assume throughout the paper that the direct strategies $\boldsymbol{d}$ are given. Further examples and discussion of this definition can be found in Appendix A.

Although the revelation principle is a very useful characterization of optimal equilibria, as long as we are given $\boldsymbol{d}$, all of the results in this paper actually apply regardless of whether the revelation principle is satisfied: when it fails, our algorithms will simply yield an *optimal direct equilibrium* which may not be an optimal equilibrium. Under the revelation principle, the problem of computing an optimal equilibrium can be expressed as follows:

$$\max_{\boldsymbol{\mu} \in \Xi} u_0(\boldsymbol{\mu}, \boldsymbol{d}) \quad \text{s.t.} \quad \max_{\boldsymbol{x}_i \in X_i} u_i(\boldsymbol{\mu}, \boldsymbol{x}_i, \boldsymbol{d}_{-i}) \le u_i(\boldsymbol{\mu}, \boldsymbol{d}) \ \ \forall i \in [\![n]\!].$$

The objective $u_0(\boldsymbol{\mu}, \boldsymbol{d})$ can be expressed as a linear expression $\boldsymbol{c}^\top \boldsymbol{\mu}$, and $u_i(\boldsymbol{\mu}, \boldsymbol{x}_i, \boldsymbol{d}_{-i}) - u_i(\boldsymbol{\mu}, \boldsymbol{d})$ can be expressed as a bilinear expression $\boldsymbol{\mu}^\top \mathbf{A}_i \boldsymbol{x}_i$. Thus, the above program can be rewritten as

$$\max_{\boldsymbol{\mu} \in \Xi} \quad \boldsymbol{c}^\top \boldsymbol{\mu} \quad \text{s.t.} \quad \max_{\boldsymbol{x}_i \in X_i} \boldsymbol{\mu}^\top \mathbf{A}_i \boldsymbol{x}_i \le 0 \ \ \forall i \in [\![n]\!]. \tag{G}$$

Zhang and Sandholm [100] now proceed by taking the dual linear program of the inner maximization, which suffices to show that (G) can be solved using linear programming.[6]

Finally, although our main focus in this paper is on games with discrete action sets, it is worth pointing out that some of our results readily apply to continuous games as well using, for example, the discretization approach of Kroer and Sandholm [60].

## 3 Lagrangian relaxations and a reduction to a zero-sum game

Our approach in this paper relies on Lagrangian relaxations of the linear program (G). In particular, in this section we introduce two different Lagrangian relaxations. The first one (Section 3.1) reduces computing an optimal equilibrium to solving a *single* zero-sum game. We find that this approach performs exceptionally well in benchmark extensive-form games in the tabular regime, but it may struggle when used in conjunction with deep RL solvers since it increases significantly the range of the rewards. This shortcoming is addressed by our second method, introduced in Section 3.2, which instead solves a *sequence* of suitable zero-sum games.

### 3.1 "Direct" Lagrangian

Directly taking a Lagrangian relaxation of the LP (G) gives the following saddle-point problem:

$$\max_{\boldsymbol{\mu} \in \Xi} \min_{\substack{\lambda \in \mathbb{R}_{\ge 0}, \\ \boldsymbol{x}_i \in X_i : i \in [\![n]\!]}} \boldsymbol{c}^\top \boldsymbol{\mu} - \lambda \sum_{i=1}^n \boldsymbol{\mu}^\top \mathbf{A}_i \boldsymbol{x}_i. \tag{L1}$$

We first point out that the above saddle-point optimization problem admits a solution $(\boldsymbol{\mu}^*, \boldsymbol{x}^*, \lambda^*)$:

**Proposition 3.1.** *The problem* (L1) *admits a finite saddle-point solution* $(\boldsymbol{\mu}^*, \boldsymbol{x}^*, \lambda^*)$. *Moreover, for all fixed* $\lambda > \lambda^*$, *the problems* (L1) *and* (G) *have the same value and same set of optimal solutions.*

The proof is in Appendix C. We will call the smallest possible $\lambda^*$ the *critical Lagrange multiplier*.

**Proposition 3.2.** *For any fixed value $\lambda$, the saddle-point problem* (L1) *can be expressed as a zero-sum extensive-form game.*

*Proof.* Consider the zero-sum extensive-form game $\hat{\Gamma}$ between two players, the *mediator* and the *deviator*, with the following structure:

1. Nature picks, with uniform probability, whether or not there is a deviator. If nature picks that there should be a deviator, then nature samples, also uniformly, a deviator $i \in [\![n]\!]$. Nature's actions are revealed to the deviator, but kept private from the mediator.

---

[6]Computing optimal equilibria can be phrased as a linear program, and so in principle Adler's reduction could also lead to an equivalent zero-sum game [2]. However, that reduction does not yield an *extensive-form* zero-sum game, which is crucial for our purposes; see Section 3.

2. The game $\Gamma$ is played. All players, except $i$ if nature picked a deviator, are constrained to according to $\boldsymbol{d}_i$. The deviator plays on behalf of Player $i$.
3. Upon reaching terminal node $z$, there are two cases. If nature picked a deviator $i$, the utility is $-2\lambda n \cdot u_i(z)$. If nature did not pick a deviator, the utility is $2u_0(z) + 2\lambda \sum_{i=1}^n u_i(z)$.

The mediator's expected utility in this game is

$$u_0(\boldsymbol{\mu}, \boldsymbol{d}) - \lambda \sum_{i=1}^n [u_i(\boldsymbol{\mu}, \boldsymbol{x}_i, \boldsymbol{d}_{-i}) - u_i(\boldsymbol{\mu}, \boldsymbol{d})]. \qquad \square$$

This characterization enables us to exploit technology used for extensive-form zero-sum game solving to compute optimal equilibria for an entire hierarchy of equilibrium concepts (Appendix A).

We will next focus on the computational aspects of solving the induced saddle-point problem (L1) using regret minimization techniques. All of the omitted proofs are deferred to Appendices D and E.

The first challenge that arises in the solution of (L1) is that the domain of the minimizing player is unbounded—the Lagrange multiplier is allowed to take any nonnegative value. Nevertheless, we show in Theorem D.1 that it suffices to set the Lagrange multiplier to a fixed value (that may depend on the time horizon); appropriately setting that value will allow us to trade off between the equilibrium gap and the optimality gap. We combine this theorem with standard regret minimizers (such as variants of CFR employed in Section 4.1) to guarantee fast convergence to optimal equilibria.

**Corollary 3.3.** *There exist regret minimization algorithms such that when employed in the saddle-point problem* (L1)*, the average strategy of the mediator $\bar{\boldsymbol{\mu}} := \frac{1}{T} \sum_{t=1}^T \boldsymbol{\mu}^{(t)}$ converges to the set of optimal equilibria at a rate of $T^{-1/4}$. Moreover, the per-iteration complexity is polynomial for communication and certification equilibria (under the nested range condition [100]), while for NFCCE, EFCCE and EFCE, implementing each iteration admits a fixed-parameter tractable algorithm.*

Furthermore, we leverage the technique of *optimism*, pioneered by Chiang et al. [18], Rakhlin and Sridharan [87], Syrgkanis et al. [94], to obtain a faster rate of convergence.

**Corollary 3.4** (Improved rates via optimism)**.** *There exist regret minimization algorithms that guarantee that the average strategy of the mediator $\bar{\boldsymbol{\mu}} := \frac{1}{T} \sum_{t=1}^T \boldsymbol{\mu}^{(t)}$ converges to the set of optimal equilibria at a rate of $T^{-1/2}$. The per-iteration complexity is analogous to Corollary 3.3.*

While this rate is slower than the (near) $T^{-1}$ rates known for converging to some of those equilibria [24, 34, 85, 3], Corollaries 3.3 and 3.4 additionally guarantee convergence to *optimal* equilibria; improving the $T^{-1/2}$ rate of Corollary 3.4 is an interesting direction for future research.

**Last-iterate convergence**   The convergence results we have stated thus far apply for the *average* strategy of the mediator—a typical feature of traditional guarantees in the no-regret framework. Nevertheless, an important advantage of our mediator-augmented formulation is that we can also guarantee *last-iterate convergence* to optimal equilibria in general games. Indeed, this follows readily from our reduction to two-player zero-sum games, leading to the following guarantee.

**Theorem 3.5** (Last-iterate convergence to optimal equilibria in general games)**.** *There exist algorithms that guarantee that the last strategy of the mediator $\boldsymbol{\mu}^{(T)}$ converges to the set of optimal equilibria at a rate of $T^{-1/4}$. The per-iteration complexity is analogous to Corollaries 3.3 and 3.4.*

As such, our mediator-augmented paradigm bypasses known hardness results in the traditional learning paradigm (Proposition D.2) since iterate convergence is no longer tied to Nash equilibria.

## 3.2   Thresholding and binary search

A significant weakness of the above Lagrangian is that the multiplier $\lambda^*$ can be large. This means that, in practice, the zero-sum game that needs to be solved to compute an optimal equilibrium could have a large reward range. While this is not a problem for most tabular methods that can achieve high precision, more scalable methods based on reinforcement learning tend to be unable to solve games to the required precision. In this section, we will introduce another Lagrangian-based method for solving the program (G) that will not require solving games with large reward ranges.

Specifically, let $\tau \in \mathbb{R}$ be a fixed threshold value, and consider the bilinear saddle-point problem

$$\max_{\boldsymbol{\mu} \in \Xi} \min_{\substack{\boldsymbol{\lambda} \in \Delta^{n+1}, \\ \boldsymbol{x}_i \in X_i : i \in [\![n]\!]}} \boldsymbol{\lambda}_0(\boldsymbol{c}^\top \boldsymbol{\mu} - \tau) - \sum_{i=1}^{n} \boldsymbol{\lambda}_i \boldsymbol{\mu}^\top \mathbf{A}_i \boldsymbol{x}_i, \tag{L2}$$

where $\Delta^k := \{\boldsymbol{\lambda} \in \mathbb{R}^k_{\geq 0} : \mathbf{1}^\top \boldsymbol{\lambda} = 1\}$ is the probability simplex on $k$ items. This Lagrangian was also stated—but not analyzed—by Farina et al. [30], in the special case of correlated equilibrium concepts (NFCCE, EFCCE, EFCE). Compared to that paper, ours contains a more complete analysis, and is general to more notions of equilibrium.

Like (L1), this Lagrangian is also a zero-sum game, but unlike (L1), the reward range in this Lagrangian is bounded by an absolute constant:

**Proposition 3.6.** *Let $\Gamma$ be a (mediator-augmented) game in which the reward for all agents is bounded in $[0,1]$. For any fixed $\tau \in [0,1]$, the saddle-point problem (L2) can be expressed as a zero-sum extensive-form game whose reward is bounded in $[-2,2]$.*

*Proof.* Consider the zero-sum extensive-form game $\hat{\Gamma}$ between two players, the *mediator* and the *deviator*, with the following structure:

1. The deviator picks an index $i \in [\![n]\!] \cup \{0\}$.
2. If $i \neq 0$, nature picks whether Player $i$ can deviate, uniformly at random.
3. The game $\Gamma$ is played. All players, except $i$ if $i \neq 0$ and nature selected that $i$ can deviate, are constrained to play according to $\boldsymbol{d}_i$. The deviator plays on behalf of Player $i$.
4. Upon reaching terminal node $z$, there are three cases. If nature picked $i = 0$, the utility is $u_0(z) - \tau$. Otherwise, if nature picked that Player $i \neq 0$ can deviate, the utility is $-2u_i(z)$. Finally, if nature picked that Player $i \neq 0$ cannot deviate, the utility is $2u_i(z)$.

The mediator's expected utility in this game is exactly

$$\boldsymbol{\lambda}_0 u_0(\boldsymbol{\mu}, \boldsymbol{d}) - \sum_{i=1}^{n} \boldsymbol{\lambda}_i [u_i(\boldsymbol{\mu}, \boldsymbol{x}_i, \boldsymbol{d}_{-i}) - u_i(\boldsymbol{\mu}, \boldsymbol{d})]$$

where $\boldsymbol{\lambda} \in \Delta^{n+1}$ is the deviator's mixed strategy in the first step. $\qquad \square$

The above observations suggest a binary-search-like algorithm for computing optimal equilibria; the pseudocode is given as Algorithm 1. The algorithm solves $O(\log(1/\varepsilon))$ zero-sum games, each to precision $\varepsilon$. Let $v^*$ be the optimal value of (G). If $\tau \leq v^*$, the value of (L2) is 0, and we will therefore never branch low, in turn implying that $u \geq v^*$ and $\ell \geq v^* - \varepsilon$. As a result, we have proven:

**Theorem 3.7.** *Algorithm 1 returns an $\varepsilon$-approximate equilibrium $\boldsymbol{\mu}$ whose value to the mediator is at least $v^* - 2\varepsilon$. If the underlying game solver used to solve (L2) runs in time $f(\Gamma, \varepsilon)$, then Algorithm 1 runs in time $O(f(\Gamma, \varepsilon) \log(1/\varepsilon))$.*

---

**ALGORITHM 1:** Pseudocode for binary search-based algorithm

1 **input:** game $\Gamma$ with mediator reward range $[0,1]$, target precision $\varepsilon > 0$
2 $\ell \leftarrow 0, u \leftarrow 1$
3 **while** $u - \ell > \varepsilon$ **do**
4 $\quad$ $\tau \leftarrow (\ell + u)/2$
5 $\quad$ run an algorithm to solve game (L2) until either
6 $\quad\quad$ (1) it finds a $\boldsymbol{\mu}$ achieving value $\geq -\varepsilon$ in (L2), or
7 $\quad\quad$ (2) it proves that the value of (L2) is $< 0$
8 $\quad$ **if** *case (1) happened* **then** $\ell \leftarrow \tau$
9 $\quad$ **else** $u \leftarrow \tau$
10 **return** *the last $\boldsymbol{\mu}$ found*

---

The differences between the two Lagrangian formulations can be summarized as follows:

1. Using (L1) requires only a single game solve, whereas using (L2) requires $O(\log(1/\varepsilon))$ game solves.
2. Using (L2) requires only an $O(\varepsilon)$-approximate game solver to guarantee value $v^* - \varepsilon$, whereas using (L1) would require an $O(\varepsilon/\lambda^*)$-approximate game solver to guarantee the same, even assuming that the critical Lagrange multiplier $\lambda^*$ in (L1) is known.

Which is preferred will therefore depend on the application. In practice, if the games are too large to be solved using tabular methods, one can use approximate game solvers based on deep reinforcement learning. In this setting, since reinforcement learning tends to be unable to achieve the high precision required to use (L1), using (L2) should generally be preferred. In Section 4, we back up these claims with concrete experiments.

## 4 Experimental evaluation

In this section, we demonstrate the practical scalability and flexibility of our approach, both for computing optimal equilibria in extensive-form games, and for designing optimal mechanisms in large-scale sequential auction design problems.

### 4.1 Optimal equilibria in extensive-form games

We first extensively evaluate the empirical performance of our two-player zero-sum reduction (Section 3.1) for computing seven equilibrium solution concepts across 23 game instances; the results using the method of Section 3.2 are slightly inferior, and are included in Appendix H. The game instances we use are described in detail in Appendix F, and belong to following eight different classes of established parametric benchmark games, each identified with an alphabetical mnemonic: B – Battleship [30], D – Liar's dice [68], GL – Goofspiel [88], K – Kuhn poker [61], L – Leduc poker [93], RS – ridesharing game [101], S – Sheriff [30], TP – double dummy bridge game [101].

For each of the 23 games, we compare the runtime required by the linear programming method of Zhang and Sandholm [100] ('LP') and the runtime required by our learning dynamics in Section 3.1 ('Ours') for computing $\varepsilon$-optimal equilibrium points.

Table 1 shows experimental results for the case in which the threshold $\varepsilon$ is set to be 1% of the payoff range of the game, and the objective function is set to be the maximum social welfare (sum of player utilities) for general-sum games, and the utility of Player 1 in zero-sum games. Each row corresponds to a game, whose identifier begins with the alphabetical mnemonic of the game class, and whose size in terms of number of nodes in the game trees is reported in the second column. The remaining columns compare, for each solution concept, the runtimes necessary to approximate the optimum equilibrium point according to that solution concept. Due to space constraints, only five out of the seven solution concepts (namely, NFCCE, EFCCE, EFCE, COMM, and CERT) are shown; data for the two remaining concepts (NFCCERT and CCERT) is given in Appendix G.

We remark that in Table 1, the column 'Ours' reports the minimum across the runtime across the different hyperparameters tried for the learning dynamics. Furthermore, for each run of the algorithms, the timeout was set at one hour. More details about the experimental setup are available in Appendix G, together with finer breakdowns of the runtimes.

We observe that our learning-based approach is faster—often by more than an order of magnitude—and more scalable than the linear program. Our additional experiments with different objective functions and values of $\varepsilon$, available in Appendix G, confirm the finding. This shows the promise of our computational approach, and reinforces the conclusion that *learning dynamics are by far the most scalable technique available today to compute equilibrium points in large games*.

### 4.2 Exact sequential auction design

Next, we use our approach to derive the optimal mechanism for a sequential auction design problem. In particular, we consider a two-round auction with two bidders, each starting with a budget of 1. The valuation for each item for each bidder is sampled uniformly at random from the set $\{0, 1/4, 1/2, 3/4, 1\}$. We consider a mediator-augmented game in which the principal chooses an outcome (allocation and payment for each player) given their reports (bids). We use CFR+ [95] as learning algorithm and a fixed Lagrange multiplier $\lambda := 25$ to compute the optimal communication

Table 1: Experimental comparison between our learning-based approach ('Ours', Section 3.1) and the linear-programming-based method ('LP') of Zhang and Sandholm [100]. Within each pair of cells corresponding to 'LP' *vs* 'Ours,' the faster algorithm is shaded blue while the hue of the slower algorithm depends on how much slower it is. If both algorithms timed out, they are both shaded gray.

| Game | # Nodes | NFCCE | | EFCCE | | EFCE | | COMM | | CERT | |
|---|---|---|---|---|---|---|---|---|---|---|---|
| | | LP | Ours | LP | Ours | LP | Ours | LP | Ours | LP | Ours |
| B2222 | 1573 | 0.00s | 0.00s | 0.00s | 0.01s | 0.00s | 0.02s | 2.00s | 1.49s | 0.00s | 0.02s |
| B2322 | 23 839 | 0.00s | 0.01s | 3.00s | 0.69s | 9.00s | 1.60s | timeout | 4m 41s | 2.00s | 1.24s |
| B2323 | 254 239 | 6.00s | 0.33s | 1m 21s | 14.23s | 3m 40s | 44.87s | timeout | timeout | 37.00s | 40.45s |
| B2324 | 1 420 639 | 38.00s | 2.73s | timeout | 3m 1s | timeout | 10m 48s | timeout | timeout | timeout | 6m 14s |
| D32 | 1017 | 0.00s | 0.01s | 0.00s | 0.02s | 12.00s | 0.40s | 0.00s | 0.06s | 0.00s | 0.01s |
| D33 | 27 622 | 2m 17s | 12.93s | timeout | 1m 46s | timeout | timeout | timeout | 4m 37s | 4.00s | 3.14s |
| GL3 | 7735 | 0.00s | 0.01s | 1.00s | 0.02s | 0.00s | 0.01s | timeout | 7.72s | 0.00s | 0.02s |
| K35 | 1501 | 49.00s | 0.76s | 46.00s | 0.67s | 57.00s | 0.55s | 1.00s | 0.03s | 0.00s | 0.01s |
| L3132 | 8917 | 26.00s | 0.59s | 8m 43s | 5.13s | 8m 18s | 6.10s | 8.00s | 3.46s | 1.00s | 0.10s |
| L3133 | 12 688 | 38.00s | 0.94s | 20m 26s | 8.88s | 21m 25s | 6.84s | 12.00s | 3.40s | 1.00s | 0.22s |
| L3151 | 19 981 | timeout | 15.12s | timeout | timeout | timeout | timeout | timeout | 16.73s | 2.00s | 0.21s |
| L3223 | 15 659 | 4.00s | 0.44s | 1m 10s | 2.94s | 2m 2s | 5.52s | 19.00s | 18.19s | 1.00s | 0.61s |
| L3523 | 1 299 005 | timeout | 1m 7s | timeout | timeout | timeout | timeout | timeout | timeout | timeout | 2m 58s |
| S2122 | 705 | 0.00s | 0.00s | 0.00s | 0.01s | 0.00s | 0.02s | 2.00s | 0.35s | 0.00s | 0.02s |
| S2123 | 4269 | 0.00s | 0.01s | 1.00s | 0.06s | 1.00s | 0.15s | 1m 33s | 59.63s | 1.00s | 0.15s |
| S2133 | 9648 | 1.00s | 0.02s | 3.00s | 0.11s | 3.00s | 0.49s | timeout | 12m 11s | 2.00s | 0.92s |
| S2254 | 712 552 | 1m 58s | 7.43s | timeout | 22.01s | timeout | 3m 34s | timeout | timeout | timeout | 2m 42s |
| S2264 | 1 303 177 | 3m 43s | 11.74s | timeout | 39.23s | timeout | timeout | timeout | timeout | timeout | timeout |
| TP3 | 910 737 | 1m 38s | 7.44s | timeout | 13.76s | timeout | 13.46s | timeout | timeout | timeout | 26.70s |
| RS212 | 598 | 0.00s | 0.00s | 0.00s | 0.00s | 0.00s | 0.00s | 2.00s | 0.01s | 0.00s | 0.00s |
| RS222 | 734 | 0.00s | 0.00s | 0.00s | 0.00s | 0.00s | 0.00s | 3.00s | 0.01s | 0.00s | 0.00s |
| RS213 | 6274 | timeout | 14.68s | timeout | 15.54s | timeout | 23.37s | 6m 25s | 8.74s | 0.00s | 0.02s |
| RS223 | 6238 | timeout | timeout | timeout | timeout | timeout | timeout | 8m 54s | 4.00s | 1.00s | 0.01s |

equilibrium that corresponds to the optimal mechanism. We terminated the learning procedure after 10000 iterations, at a duality gap for (L1) of approximately $4.2 \times 10^{-4}$. Figure 1 (left) summarizes our results. On the y-axis we show how exploitable (that is, how incentive-incompatible) each of the considered mechanisms are, confirming that for this type of sequential settings, second-price auctions (SP) with or without reserve price, as well as the first-price auction (FP), are typically incentive-incompatible. On the x-axis, we report the hypothetical revenue that the mechanism would extract assuming truthful bidding. Our mechanism is provably incentive-compatible and extracts a larger revenue than all considered second-price mechanisms. It also would extract less revenue than the hypothetical first-price auction if the bidders behaved truthfully (of course, real bidders would not behave honestly in the first-price auction but rather would shade their bids downward, so the shown revenue benchmark in Figure 1 is actually not achievable). Intriguingly, we observed that 8% of the time the mechanism gives an item away for free. Despite appearing irrational, this behavior can incentivize bidders to use their budget earlier in order to encourage competitive bidding, and has been independently discovered in manual mechanism design recently [25, 75].

### 4.3 Scalable sequential auction design via deep reinforcement learning

We also combine our framework with deep-learning-based algorithms for scalable equilibrium computation in two-player zero-sum games to compute optimal mechanisms in two sequential auction settings. To compute an optimal mechanism using our framework, we use the PSRO algorithm [63], a deep reinforcement learning method based on the double oracle algorithm that has empirically scaled to large games such as Starcraft [96] and Stratego [72], as the game solver in Algorithm 1.[7] To train the best responses, we use proximal policy optimization (PPO) [90].

---

[7]We also tested PSRO on the Lagrangian (L1), but this proved to be incompatible with deep learning due to the large reward range induced by the multiplier $\lambda$.

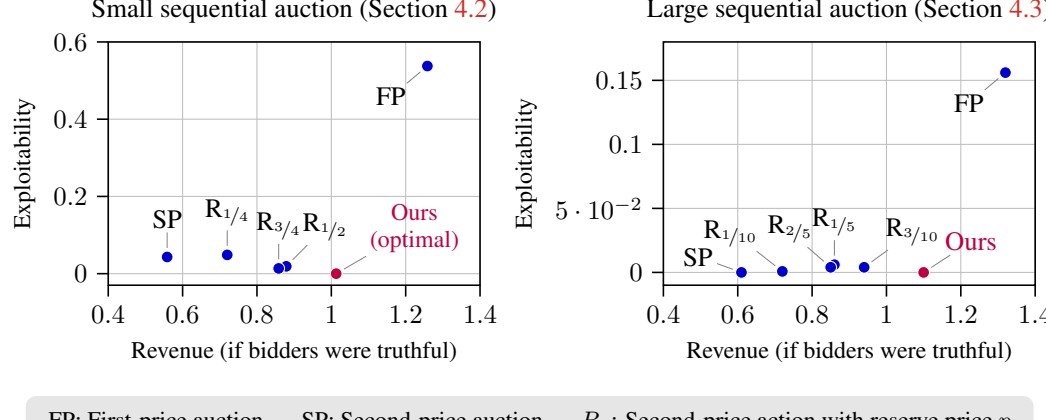

Figure 1: Exploitability is measured by summing the best response for both bidders to the mechanism. Zero exploitability corresponds to incentive compatibility. In a sequential auction with budgets, our method is able to achieve higher revenue than second-price auctions and better incentive compatibility than a first-price auction.

First, to verify that the deep learning method is effective, we replicate the results of the tabular experiments in Section 4.2. We find that PSRO achieves the same best response values and optimal equilibrium value computed by the tabular experiment, up to a small error. These results give us confidence that our method is correct.

Second, to demonstrate scalability, we run our deep learning-based algorithm on a larger auction environment that would be too big to solve with tabular methods. In this environment, there are four rounds, and in each round the valuation of each player is sampled uniformly from $\{0, 0.1, 0.2, 0.3, 0.4, 0.5\}$. The starting budget of each player is, again, 1. We find that, like the smaller setting, the optimal revenue of the mediator is $\approx 1.1$ (right-side of Figure 1). This revenue exceeds the revenue of every second-price auction (none of which have revenue greater than 1).[8]

## 5 Conclusions

We proposed a new paradigm of learning in games. It applies to mechanism design, information design, and solution concepts in multi-player extensive-form games such as correlated, communication, and certification equilibria. Leveraging a Lagrangian relaxation, our paradigm reduces the problem of computing optimal equilibria to determining minimax equilibria in zero-sum extensive-form games. We also demonstrated the scalability of our approach for *computing* optimal equilibria by attaining state-of-the-art performance in benchmark tabular games, and by solving a sequential auction design problem using deep reinforcement learning.

---

[8]We are inherently limited in this setting by the inexactness of best responses based on deep reinforcement learning; as such, it is possible that these values are not exact. However, because of the success of above tabular experiment replications, we believe that our results should be reasonably accurate.

## Acknowledgements

We are grateful to the anonymous NeurIPS reviewers for many helpful comments that helped improve the presentation of this paper. Tuomas Sandholm's work is supported by the Vannevar Bush Faculty Fellowship ONR N00014-23-1-2876, National Science Foundation grants RI-2312342 and RI-1901403, ARO award W911NF2210266, and NIH award A240108S001. McAleer is funded by NSF grant #2127309 to the Computing Research Association for the CIFellows 2021 Project. The work of Prof. Gatti's research group is funded by the FAIR (Future Artificial Intelligence Research) project, funded by the NextGenerationEU program within the PNRR-PE-AI scheme (M4C2, Investment 1.3, Line on Artificial Intelligence). Conitzer thanks the Cooperative AI Foundation and Polaris Ventures (formerly the Center for Emerging Risk Research) for funding the Foundations of Cooperative AI Lab (FOCAL). Andy Haupt was supported by Effective Giving. We thank Dylan Hadfield-Menell for helpful conversations.

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

# A  Illustrative examples of mediator-augmented games

In this section, we further clarify the framework of mediator-augmented games we operate in through a couple of examples. We begin by noting that the family of solution concepts for extensive-form games captured by this framework includes, but is not limited to, the following:

- normal-form coarse correlated equilibrium* [5, 79],
- extensive-form coarse correlated equilibrium* [31],
- extensive-form correlated equilibrium* [98],
- certification (under the *nested range condition* [46, 37]) [37, 100],
- communication equilibrium [81, 36],
- mechanism design for sequential settings, and
- information design/Bayesian persuasion for sequential settings [58].

We refer the interested reader to Zhang and Sandholm [100, Appendix G] for additional interesting concepts not mentioned above.

*Example* A.1 (Single-item auction). Consider the single-good monopolist problem studied by Myerson [80]. Each player $i \in [\![n]\!]$ has a valuation $v_i \in V_i$. Agent valuations may be correlated, and distributed according to $\mathcal{F} \in \Delta(V)$, where $V := V_1 \times V_2 \times \cdots \times V_n$. The mechanism selects a (potentially random) payment $p$ and a winner $i^* \in [\![n]\!]$. The agents' utilities are quasilinear: $u_i(i^*, p; v_i) = v_i - p$ if $i^* = i$ and 0 otherwise. The seller wishes to maximize expected payment from the agents. This has the following timeline.

1. The mechanism commits to a (potentially randomized) mapping $\phi : \boldsymbol{v} = (v_1, \ldots, v_n) \mapsto (i^*, p)$.

2. Nature samples valuations $\boldsymbol{v} = (v_1, \ldots, v_n) \sim \mathcal{F}$.

3. Each player $i \in [\![n]\!]$ privately observes her valuation $v_i$, and then decides what valuation $v_i'$ to report to the mediator.

4. The winner and payment are selected according to $\phi(\boldsymbol{v}')$.

5. Player $i^*$ gets utility $v_{i^*} - p$, while all other players get 0. The mediator obtains utility $u_0 = p$.

In this extensive-form game, the primitives from our paper are:

- a (pure) mediator strategy $\boldsymbol{\mu} \in \Xi$ is a mapping from valuation reports $\boldsymbol{v}' = (v_1', \ldots, v_n')$ to outcomes $(i^*, p)$—that is, mediator strategies are mechanisms, and mixed strategies are randomized mechanisms;

- a (pure) player strategy $\boldsymbol{x}_i \in X_i$ for each player $i \in [\![n]\!]$ is a mapping from $V_i$ to $V_i$ indicating what valuation Player $i$ reports as a function of its valuation $v_i \in V_i$;

- the direct (in mechanism design language, truthful) strategy $\boldsymbol{d}_i$ for each player $i$ is the identity map from $V_i$ to $V_i$. (Hence, in particular, $\boldsymbol{d}_i \in X_i$ is a strategy of Player $i$, so it makes sense, for example, to call $(\boldsymbol{\mu}, \boldsymbol{d}) = (\boldsymbol{\mu}, \boldsymbol{d}_1, \ldots, \boldsymbol{d}_n)$ a strategy profile.)

In particular, if profile $(\boldsymbol{\mu}, \boldsymbol{d}_1, \ldots, \boldsymbol{d}_n)$ is such that each player $i$ is playing a best response, then $\boldsymbol{\mu}$ is a *truthful* mechanism.

The conversion in Proposition 3.2 creates a zero-sum extensive-form game $\hat{\Gamma}$, whose equilibria for the mediator (for sufficiently large $\lambda$) are precisely the revenue-maximizing mechanisms. $\hat{\Gamma}$ has the following timeline:

---

*For notions of correlated equilibrium in extensive-form games, the mediator must have *imperfect recall*, and therefore the representation of the mediator's decision space $\Xi$ may not be polynomial. This is unavoidable, since the problem of computing an optimal equilibrium under these notions is NP-hard in general [98]. In this paper, we will largely ignore these concerns and assume that the representation of the mixed strategy set $\Xi$ is part of the input.

1. Nature picks, with equal probability, whether there is a deviator. If nature picks that there is a deviator, nature also selects which player $i \in [\![n]\!]$ is represented by the deviator.

2. Nature samples valuations $\boldsymbol{v} = (v_1, \ldots, v_n) \sim \mathcal{F}$.

3. If nature selected that there is a deviator, the deviator observes $i$ and its valuation $v_i \in V_i$, and selects a deviation $v_i' \in V_i$.

4. The mediator observes $(v_i', \boldsymbol{v}_{-i})$ (*i.e.*, all other players are assumed to have reported honestly) and selects a winner $i^*$ and payment $p$, as before.

5. There are now two cases. If nature selected at the root that there was to be a deviator, the utility for the mediator is $-2\lambda n u_i(i^*, p; v_i)$. If nature selected at the root that there was to be no deviator, the utility for the mediator is $2p + 2\lambda \sum_{i=1}^n u_i(i^*, p; v_i) = 2p + 2\lambda(v_{i^*} - p)$.

As our second example, we show how the problem of computing a social-welfare-maximizing correlated equilbrium (CE) in a normal-form game can be captured using mediator-augmented games.

*Example* A.2 (Social welfare-optimal correlated equilibria in normal-form games). Let $A_i$ be the action set for each player $i \in [\![n]\!]$ in the game, and let utility functions $u_i : A \to \mathbb{R}$, where $A := A_1 \times A_2 \times \cdots \times A_n$. The social welfare is the function $u_0 : A \to \mathbb{R}$ given by $u_0(\boldsymbol{a}) := \sum_{i=1}^n u_i(\boldsymbol{a})$. In the traditional formulation, a CE is a correlated distribution $\boldsymbol{\mu}$ over $A$. The elements $(a_1, \ldots, a_n)$ sampled from $\boldsymbol{\mu}$ can be thought of as profiles of action recommendations for the players such that no player has any incentive to not follow the recommendation (obedience). This has the following well-known timeline.

1. At the beginning of the game, the mediator player chooses a profile of recommendations $\boldsymbol{a} = (a_1, \ldots, a_n) \in A$.

2. Each player observes its recommendation $a_i$ and chooses an action $a_i'$.

3. Each player gets utility $u_i(\boldsymbol{a}')$, and the mediator gets utility $u_0(\boldsymbol{a}') = \sum_{i=1}^n u_i(\boldsymbol{a}')$.

In this game:

- mixed strategies $\boldsymbol{\mu}$ for the mediator are distributions over $A$, that is, they are correlated profiles;

- a (pure) strategy for player $i \in [\![n]\!]$ is a mapping from $A_i$ to $A_i$, encoding the action player $i \in [\![n]\!]$ will take upon receiving each recommendation;

- the direct strategy $\boldsymbol{d}_i$ is again the identity map (*i.e.*, each player selects as action what the mediator recommended to him/her).

In particular, if profile $(\boldsymbol{\mu}, \boldsymbol{d}_1, \ldots, \boldsymbol{d}_n)$ is such that each player $i$ is playing a best response, then $\boldsymbol{\mu}$ is a CE.

Proposition 3.2 yields the following zero-sum game whose mediator equilibrium strategies (for sufficiently large $\lambda$) are precisely the welfare-optimal equilibria:

1. Nature picks, with equal probability, whether there is a deviator. If nature picks that there is a deviator, nature also selects which player $i \in [\![n]\!]$ is represented by the deviator.

2. The mediator picks a pure strategy profile $\boldsymbol{a} = (a_1, \ldots, a_n) \in A$.

3. If there is a deviator, the deviator observes $i \in [\![n]\!]$ and the recommendation $a_i$ and picks an action $a_i'$.

4. There are now two cases. If nature selected at the root that there was to be a deviator, the utility for the mediator is $-2\lambda n u_i(a_i', \boldsymbol{a}_{-i})$. If nature selected at the root that there was to be no deviator, the utility for the mediator is $2u_0(\boldsymbol{a}) + 2\lambda \sum_{i=1}^n u_i(\boldsymbol{a}) = 2(1 + \lambda)u_0(\boldsymbol{a})$.

# B  Further related work

In this section, we provide additional related work omitted from the main body.

We first elaborate further on prior work regarding the complexity of computing equilibria in games. Much attention has been focused on the complexity of computing just any one Nash equilibrium. This has been motivated in part by the idea that if even this is hard to compute, then this casts doubt on the concept of Nash equilibrium as a whole [23]; but the interest also stemmed from the fact that the complexity of the problem was open for a long time [82], and ended up being complete for an exotic complexity class [23, 17], whereas computing a Nash equilibrium that reaches a certain objective value is "simply" NP-complete [41, 21]. None of this, however, justifies settling for just any one equilibrium in practice. Moreover, for correlated equilibria and related concepts, the complexity considerations are different. While one (extensive-form) correlated equilibrium can be computed in polynomial time even for multi-player succinct games [83, 56, 54] (under the polynomial expectation property), computing one that maximizes some objective function is typically NP-hard [83, 98]. To make matters worse, even finding one that is *strictly* better—in terms of social welfare—than the worst one is also computationally intractable [9]. Of course, our results do not contradict those lower bounds. For example, in multi-player normal-form games the strategy space of the mediator has an exponential description, thereby rendering all our algorithms exponential in the number of players. We stress again that while there exist algorithms that avoid this exponential dependence, they are not guaranteed to compute an optimal equilibrium, which is the main focus of this paper.

Moreover, in our formulation the mediator has the power to commit to a strategy. As such, our results also relate to the literature on learning and computing Stackelberg equilibria [8, 35, 66, 84, 20], as well as the work of Camara et al. [15] which casts mechanism design as a repeated interaction between a principal and an agent. Stackelberg equilibria in extensive-form games are, however, hard to find in general [65]. Our Stackelberg game has a much nicer form than general Stackelberg games—in particular, we know in advance what the equilibrium strategies will be for the followers (namely, the direct strategies, ). This observation is what allows the reduction to zero-sum games, sidestepping the need to use Stackleberg-specific technology or solvers and resulting in efficient algorithms.

In an independent and concurrent work, Fujii [40] provided independent learning dynamics converging to the set of communication equilibria in Bayesian games, but unlike our algorithm there are no guarantees for finding an optimal one. Also in independent and concurrent work, Ivanov et al. [55] develop similar Lagrangian-based dynamics for the equilibrium notion that, in the language of this paper and Zhang and Sandholm [100], is *coarse full-certification equilibrium*. Differing from ours, their paper does not present any theoretical guarantees (instead focusing on practical results).

# C  Proof of Proposition 3.1

In this section, we provide the proof of Proposition 3.1, the statement of which is recalled below.

**Proposition 3.1.** *The problem* (L1) *admits a finite saddle-point solution* $(\boldsymbol{\mu}^*, \boldsymbol{x}^*, \lambda^*)$. *Moreover, for all fixed* $\lambda > \lambda^*$, *the problems* (L1) *and* (G) *have the same value and same set of optimal solutions.*

*Proof.* Let $v$ be the optimal value of (G). The Lagrangian of (G) is

$$\max_{\boldsymbol{\mu} \in \Xi} \min_{\substack{\lambda_i \in \mathbb{R}_{\geq 0}, \\ \boldsymbol{x}_i \in X_i : i \in [\![n]\!]}} \quad \boldsymbol{c}^\top \boldsymbol{\mu} - \sum_{i=1}^{n} \lambda_i \boldsymbol{\mu}^\top \mathbf{A}_i \boldsymbol{x}_i.$$

Now, making the change of variables $\bar{\boldsymbol{x}}_i := \lambda_i \boldsymbol{x}_i$, the above problem is equivalent to

$$\max_{\boldsymbol{\mu} \in \Xi} \min_{\bar{\boldsymbol{x}}_i \in \bar{X}_i : i \in [\![n]\!]} \quad \boldsymbol{c}^\top \boldsymbol{\mu} - \sum_{i=1}^{n} \boldsymbol{\mu}^\top \mathbf{A}_i \bar{\boldsymbol{x}}_i. \tag{1}$$

where $\bar{X}_i$ is the conic hull of $X_i$: $\bar{X}_i := \{\lambda_i \boldsymbol{x}_i : \boldsymbol{x}_i \in X_i\}$. Note that, when $X_i$ is a polytope of the form $X_i := \{\mathbf{F}_i \boldsymbol{x}_i = \boldsymbol{f}_i, \boldsymbol{x}_i \geq 0\}$, its conic hull can be expressed as $\bar{X}_i = \{\mathbf{F}_i \boldsymbol{x}_i = \lambda_i \boldsymbol{f}_i, \boldsymbol{x}_i \geq 0, \lambda_i \geq 0\}$. Thus, (1) is a bilinear saddle-point problem, where $\Xi$ is compact and convex and $\bar{X}_i$ is

convex. Thus, Sion's minimax theorem [92] applies, and we have that the value of (1) is equal to the value of the problem

$$\min_{\bar{\boldsymbol{x}}_i \in X_i : i \in [\![n]\!]} \max_{\boldsymbol{\mu} \in \Xi} \quad \boldsymbol{c}^\top \boldsymbol{\mu} - \sum_{i=1}^{n} \boldsymbol{\mu}^\top \mathbf{A}_i \bar{\boldsymbol{x}}_i. \tag{2}$$

Since this is a linear program[9] with a finite value, its optimum value must be achieved by some $\bar{\boldsymbol{x}} := (\bar{\boldsymbol{x}}_1, \ldots, \bar{\boldsymbol{x}}_n) := (\lambda_1 \boldsymbol{x}_1, \ldots, \lambda_n \boldsymbol{x}_n)$. Let $\lambda^* := \max_i \lambda_i$. Using the fact that $\boldsymbol{\mu}^\top \mathbf{A}_i \boldsymbol{d}_i = 0$ for all $\boldsymbol{\mu}$, the profile

$$\bar{\boldsymbol{x}}' := (\lambda^* \boldsymbol{x}_1', \ldots, \lambda^* \boldsymbol{x}_n') \quad \text{where} \quad \boldsymbol{x}_i' = \boldsymbol{d}_i + \frac{\lambda_i}{\lambda^*}(\boldsymbol{x}_i - \boldsymbol{d}_i)$$

is also an optimal solution in (2). Therefore, for any $\lambda \geq \lambda^*$, $\boldsymbol{x}' := (\boldsymbol{x}_1', \ldots, \boldsymbol{x}_n')$ is an optimal solution for the minimizer in (L1) that achieves the value of (G), so (G) and (L1) have the same value.

Now take $\lambda > \lambda^*$, and suppose for contradiction that (L1) admits some optimal $\boldsymbol{\mu} \in \Xi$ that is not optimal in (G). Then, either $\boldsymbol{c}^\top \boldsymbol{\mu} < v$, or $\boldsymbol{\mu}$ violates some constraint $\max_{\boldsymbol{x}_i} \boldsymbol{\mu}^\top \mathbf{A}_i \boldsymbol{x}_i \leq 0$. The first case is impossible because then setting $\boldsymbol{x}_i = \boldsymbol{d}_i$ for all $i$ yields value less than $v$ in (L1). In the second case, since we know that (L1) and (G) have the same value when $\lambda = \lambda^*$, we have

$$\boldsymbol{c}^\top \boldsymbol{\mu} - \lambda \max_{\boldsymbol{x} \in X} \sum_{i=1}^{n} \boldsymbol{\mu}^\top \mathbf{A}_i \boldsymbol{x}_i < \boldsymbol{c}^\top \boldsymbol{\mu} - \lambda^* \max_{\boldsymbol{x} \in X} \sum_{i=1}^{n} \boldsymbol{\mu}^\top \mathbf{A}_i \boldsymbol{x}_i \leq v. \qquad \square$$

# D   Fast computation of optimal equilibria via regret minimization

In this section, we focus on the computational aspects of solving the induced saddle-point problem (L1) using regret minimization techniques. In particular, this section serves to elaborate on our results presented earlier in Section 3.1. All of the omitted proofs are deferred to Appendix E for the sake of exposition.

As we explained in Section 3.1, the first challenge that arises in the solution of (L1) is that the domain of Player min is unbounded—the Lagrange multiplier is allowed to take any nonnegative value. Nevertheless, we show in the theorem below that it suffices to set the Lagrange multiplier to a fixed value (that may depend on the time horizon); we reiterate that appropriately setting that value will allow us to trade off between the equilibrium gap and the optimality gap. Before we proceed, we remark that the problem of Player min in (L1) can be decomposed into the subproblems faced by each player separately, so that the regret of Player min can be cast as the sum of the players' regrets (see Corollary E.2); this justifies the notation $\sum_{i=1}^{n} \operatorname{Reg}_{X_i}^T$ used for the regret of Player min below.

**Theorem D.1.** *Suppose that Player max in the saddle-point problem* (L1) *incurs regret* $\operatorname{Reg}_\Xi^T$ *and Player min incurs regret* $\sum_{i=1}^{n} \operatorname{Reg}_{X_i}^T$ *after* $T \in \mathbb{N}$ *repetitions, for a fixed* $\lambda = \lambda(T) > 0$. *Then, the average mediator strategy* $\Xi \ni \bar{\boldsymbol{\mu}} := \frac{1}{T} \sum_{t=1}^{T} \boldsymbol{\mu}^{(t)}$ *satisfies the following:*

1. *For any strategy* $\boldsymbol{\mu}^* \in \Xi$ *such that* $\max_{i \in [\![n]\!]} \max_{\boldsymbol{x}_i^* \in X_i} (\boldsymbol{\mu}^*)^\top \mathbf{A}_i \boldsymbol{x}_i^* \leq 0$,

$$\boldsymbol{c}^\top \bar{\boldsymbol{\mu}} \geq \boldsymbol{c}^\top \boldsymbol{\mu}^* - \frac{1}{T}\left(\operatorname{Reg}_\Xi^T + \sum_{i=1}^{n} \operatorname{Reg}_{X_i}^T\right);$$

2. *The equilibrium gap of* $\bar{\boldsymbol{\mu}}$ *decays with a rate of* $\lambda^{-1}$:

$$\max_{i \in [\![n]\!]} \max_{\boldsymbol{x}_i^* \in X_i} \bar{\boldsymbol{\mu}}^\top \mathbf{A}_i \boldsymbol{x}_i^* \leq \frac{\max_{\boldsymbol{\mu}, \boldsymbol{\mu}' \in \Xi} \boldsymbol{c}^\top (\boldsymbol{\mu} - \boldsymbol{\mu}')}{\lambda} + \frac{1}{\lambda T}\left(\operatorname{Reg}_\Xi^T + \sum_{i=1}^{n} \operatorname{Reg}_{X_i}^T\right).$$

As a result, if we can simultaneously guarantee that $\lambda(T) \to +\infty$ and $\frac{1}{T}\left(\operatorname{Reg}_\Xi^T + \sum_{i=1}^{n} \operatorname{Reg}_{X_i}^T\right) \to 0$, as $T \to +\infty$, Theorem D.1 shows that both the optimality gap (Item 1) and the equilibrium gap

---

[9]This holds by taking a dual of the inner minimization.

(Item 2) converge to 0. We show that this is indeed possible in the sequel (Corollaries 3.3 and 3.4), obtaining favorable rates of convergence as well.

It is important to stress that while there exists a bounded critical Lagrange multiplier for our problem (Proposition 3.1), thereby obviating the need for truncating its value, such a bound is not necessarily polynomial. For example, halving the players' utilities while maintaining the utility of the mediator would require doubling the magnitude of the critical Lagrange multiplier.

Next, we combine Theorem D.1 with suitable regret minimization algorithms in order to guarantee fast convergence to optimal equilibria. Let us first focus on the side of Player min in (L1), which, as pointed out earlier, can be decomposed into subproblems corresponding to each player separately (Corollary E.2). Minimizing regret over the sequence-form polytope can be performed efficiently with a variety of techniques, which can be classified into two basic approaches. The first one is based on the standard online mirror descent algorithm (see, *e.g.*, [91]), endowed with appropriate *distance generating functions (DGFs)* [32]. The alternative approach is based on regret decomposition, in the style of CFR [106, 29]. In particular, given that the players' observed utilities have range $O(\lambda)$, the regret of each player under suitable learning algorithms will grow as $O(\lambda\sqrt{T})$ (see Proposition E.1). Furthermore, efficiently minimizing regret from the side of the mediator depends on the equilibrium concept at hand. For NFCCE, EFCCE and EFCE, the imperfect recall of the mediator [101] induces a computationally hard problem [19], which nevertheless admits fixed-parameter tractable algorithms [103] (Proposition E.3). In contrast, for communication and certification equilibria the perfect recall of the mediator enables efficient computation for any extensive-form game. As a result, selecting a bound of $\lambda := T^{1/4}$ on the Lagrange multiplier, so as to optimally trade off Items 1 and 2 of Theorem D.1, leads to the following conclusion.

**Corollary 3.3.** *There exist regret minimization algorithms such that when employed in the saddle-point problem* (L1)*, the average strategy of the mediator $\bar{\boldsymbol{\mu}} := \frac{1}{T}\sum_{t=1}^{T}\boldsymbol{\mu}^{(t)}$ converges to the set of optimal equilibria at a rate of $T^{-1/4}$. Moreover, the per-iteration complexity is polynomial for communication and certification equilibria (under the nested range condition [100]), while for NFCCE, EFCCE and EFCE, implementing each iteration admits a fixed-parameter tractable algorithm.*

Furthermore, we leverage the technique of *optimism*, pioneered by Chiang et al. [18], Rakhlin and Sridharan [87], Syrgkanis et al. [94] in the context of learning in games, in order to obtain faster rates. In particular, using optimistic mirror descent we can guarantee that the sum of the agents' regrets in the saddle-point problem (L1) will now grow as $O(\lambda)$ (Proposition E.4), instead of the previous bound $O(\lambda\sqrt{T})$ obtained using vanilla mirror descent. Thus, letting $\lambda = T^{1/2}$ leads to the following improved rate of convergence.

**Corollary 3.4** (Improved rates via optimism)**.** *There exist regret minimization algorithms that guarantee that the average strategy of the mediator $\bar{\boldsymbol{\mu}} := \frac{1}{T}\sum_{t=1}^{T}\boldsymbol{\mu}^{(t)}$ converges to the set of optimal equilibria at a rate of $T^{-1/2}$. The per-iteration complexity is analogous to Corollary 3.3.*

We reiterate that while this rate is slower than the (near) $T^{-1}$ rates known for converging to some of those equilibria [24, 34, 85, 3], Corollaries 3.3 and 3.4 additionally guarantee convergence to *optimal* equilibria; improving the $T^{-1/2}$ rate of Corollary 3.4 is an interesting direction for the future.

**Last-iterate convergence** The results we have stated thus far apply for the *average* strategy of the mediator—a typical feature of traditional guarantees in the no-regret framework. In contrast, there is a recent line of work that endeavors to recover *last-iterate* guarantees as well [22, 45, 1, 14, 6, 99, 64, 43, 67, 44]. Yet, despite many efforts, the known last-iterate guarantees of no-regret learning algorithms apply only for restricted classes of games, such as two-player zero-sum games. There is an inherent reason for the limited scope of those results: last-iterate convergence is inherently tied to Nash equilibria, which in turn are hard to compute in general games [23, 17]—let alone computing an optimal one [41, 21]. Indeed, any given joint strategy profile of the players induces a product distribution, so iterate convergence requires—essentially by definition—at the very least computing an approximate Nash equilibrium.

**Proposition D.2** (Informal)**.** *Any independent learning dynamics (without a mediator) require superpolynomial time to guarantee $\varepsilon$-last-iterate convergence, for a sufficiently small $\varepsilon = O(m^{-c})$, even for two-player $m$-action normal-form games, unless $\mathsf{PPAD} \subseteq \mathsf{P}$.*

There are also unconditional exponential communication-complexity lower bounds for uncoupled methods [7, 53, 89, 48], as well as other pertinent impossibility results [50, 74] that document the inherent persistence of limit cycles in general-sum games. In contrast, an important advantage of our mediator-augmented formulation is that we can guarantee last-iterate convergence to optimal equilibria in general games. Indeed, this follows readily from our reduction to two-player zero-sum games, for which the known bound of $O(\lambda/\sqrt{T})$ for the iterate gap of (online) optimistic gradient descent can be employed (see Appendix E).

**Theorem 3.5** (Last-iterate convergence to optimal equilibria in general games). *There exist algorithms that guarantee that the last strategy of the mediator $\boldsymbol{\mu}^{(T)}$ converges to the set of optimal equilibria at a rate of $T^{-1/4}$. The per-iteration complexity is analogous to Corollaries 3.3 and 3.4.*

As such, our mediator-augmented learning paradigm bypasses the hardness of Proposition D.2 since last-iterate convergence is no longer tied to convergence to Nash equilibria.

# E  Omitted proofs from Section D

In this section, we provide the omitted proofs from Appendix D, which concerns the solution of the saddle-point problem described in (L1) using regret minization. Appendix E.1 then presents a slightly different approach for solving (L1) using regret minimization over conic hulls, which is used in our experiments.

We begin with the proof of Theorem D.1, the statement of which is recalled below. In the following proof, we will denote by $\mathcal{L} : \Xi \times X \ni (\boldsymbol{\mu}, (\boldsymbol{x}_i)_{i=1}^n) \mapsto \boldsymbol{c}^\top \boldsymbol{\mu} - \lambda \sum_{i=1}^n \boldsymbol{\mu}^\top \mathbf{A}_i \boldsymbol{x}_i$ the induced Lagrangian, for a fixed $\lambda > 0$.

**Theorem D.1.** *Suppose that Player max in the saddle-point problem (L1) incurs regret $\operatorname{Reg}_\Xi^T$ and Player min incurs regret $\sum_{i=1}^n \operatorname{Reg}_{X_i}^T$ after $T \in \mathbb{N}$ repetitions, for a fixed $\lambda = \lambda(T) > 0$. Then, the average mediator strategy $\Xi \ni \bar{\boldsymbol{\mu}} := \frac{1}{T} \sum_{t=1}^T \boldsymbol{\mu}^{(t)}$ satisfies the following:*

1. *For any strategy $\boldsymbol{\mu}^* \in \Xi$ such that $\max_{i \in [\![n]\!]} \max_{\boldsymbol{x}_i^* \in X_i} (\boldsymbol{\mu}^*)^\top \mathbf{A}_i \boldsymbol{x}_i^* \le 0$,*

$$\boldsymbol{c}^\top \bar{\boldsymbol{\mu}} \ge \boldsymbol{c}^\top \boldsymbol{\mu}^* - \frac{1}{T} \left( \operatorname{Reg}_\Xi^T + \sum_{i=1}^n \operatorname{Reg}_{X_i}^T \right);$$

2. *The equilibrium gap of $\bar{\boldsymbol{\mu}}$ decays with a rate of $\lambda^{-1}$:*

$$\max_{i \in [\![n]\!]} \max_{\boldsymbol{x}_i^* \in X_i} \bar{\boldsymbol{\mu}}^\top \mathbf{A}_i \boldsymbol{x}_i^* \le \frac{\max_{\boldsymbol{\mu}, \boldsymbol{\mu}' \in \Xi} \boldsymbol{c}^\top (\boldsymbol{\mu} - \boldsymbol{\mu}')}{\lambda} + \frac{1}{\lambda T} \left( \operatorname{Reg}_\Xi^T + \sum_{i=1}^n \operatorname{Reg}_{X_i}^T \right).$$

*Proof.* Let $\bar{\boldsymbol{\mu}} \in \Xi$ be the average strategy of the mediator and $\bar{\boldsymbol{x}}_i \in X_i$ be the average strategy of each player $i \in [\![n]\!]$ over the $T$ iterations. We first argue about the approximate optimality of $\bar{\boldsymbol{\mu}}$. In particular, we have that

$$\boldsymbol{c}^\top \bar{\boldsymbol{\mu}} \ge \max_{\boldsymbol{\mu} \in \Xi} \left\{ \boldsymbol{c}^\top \boldsymbol{\mu} - \lambda \sum_{i=1}^n \boldsymbol{\mu}^\top \mathbf{A}_i \bar{\boldsymbol{x}}_i \right\} - \frac{1}{T} \left( \sum_{i=1}^n \operatorname{Reg}_{X_i}^T + \operatorname{Reg}_\Xi^T \right) \tag{3}$$

$$\ge \boldsymbol{c}^\top \boldsymbol{\mu}^* - \lambda \sum_{i=1}^n (\boldsymbol{\mu}^*)^\top \mathbf{A}_i \bar{\boldsymbol{x}}_i - \frac{1}{T} \left( \sum_{i=1}^n \operatorname{Reg}_{X_i}^T + \operatorname{Reg}_\Xi^T \right) \tag{4}$$

$$\ge \boldsymbol{c}^\top \boldsymbol{\mu}^* - \lambda \sum_{i=1}^n \max_{\boldsymbol{x}_i^* \in X_i} (\boldsymbol{\mu}^*)^\top \mathbf{A}_i \boldsymbol{x}_i^* - \frac{1}{T} \left( \sum_{i=1}^n \operatorname{Reg}_{X_i}^T + \operatorname{Reg}_\Xi^T \right)$$

$$\ge \boldsymbol{c}^\top \boldsymbol{\mu}^* - \frac{1}{T} \left( \sum_{i=1}^n \operatorname{Reg}_{X_i}^T + \operatorname{Reg}_\Xi^T \right), \tag{5}$$

where (3) follows from the fact that

$$\max_{\boldsymbol{\mu}^* \in \Xi} \mathcal{L}(\boldsymbol{\mu}^*, (\bar{\boldsymbol{x}}_i)_{i=1}^n) - \min_{(\boldsymbol{x}_i^*)_{i=1}^n \in X} \mathcal{L}(\bar{\boldsymbol{\mu}}, (\boldsymbol{x}_i^*)_{i=1}^n) \le \frac{1}{T} \left( \sum_{i=1}^n \operatorname{Reg}_{X_i}^T + \operatorname{Reg}_\Xi^T \right), \tag{6}$$

in turn implying (3) since $\sum_{i=1}^{n} \max_{\boldsymbol{x}_i^* \in X_i} \bar{\boldsymbol{\mu}}^\top \mathbf{A}_i \boldsymbol{x}_i^* \geq \sum_{i=1}^{n} \bar{\boldsymbol{\mu}}^\top \mathbf{A}_i \boldsymbol{d}_i = 0$; (4) uses the notation $\boldsymbol{\mu}^*$ to represent any equilibrium strategy optimizing the objective $\boldsymbol{c}^\top \boldsymbol{\mu}$; and (5) follows from the fact that, by assumption, $\boldsymbol{\mu}^*$ satisfies the equilibrium constraint: $\max_{\boldsymbol{x}_i^* \in X_i} (\boldsymbol{\mu}^*)^\top \mathbf{A}_i \boldsymbol{x}_i^* \leq 0$ for any player $i \in [\![n]\!]$, as well as the nonnegativity of the Lagrange multiplier. This establishes Item 1 of the statement.

Next, we analyze the equilibrium gap of $\bar{\boldsymbol{\mu}}$. Consider any mediator strategy $\boldsymbol{\mu} \in \Xi$ such that $\boldsymbol{\mu}^\top \mathbf{A}_i \boldsymbol{x}_i \leq 0$ for any $\boldsymbol{x}_i \in X_i$ and player $i \in [\![n]\!]$. By (6),

$$\boldsymbol{c}^\top \boldsymbol{\mu} - \lambda \sum_{i=1}^{n} \boldsymbol{\mu}^\top \mathbf{A}_i \bar{\boldsymbol{x}}_i - \boldsymbol{c}^\top \bar{\boldsymbol{\mu}} + \lambda \sum_{i=1}^{n} \max_{\boldsymbol{x}_i^* \in X_i} \bar{\boldsymbol{\mu}}^\top \mathbf{A}_i \boldsymbol{x}_i^* \leq \frac{1}{T} \left( \sum_{i=1}^{n} \mathrm{Reg}_{X_i}^T + \mathrm{Reg}_\Xi^T \right). \quad (7)$$

But, by the equilibrium constraint for $\boldsymbol{\mu}$, it follows that $\boldsymbol{\mu}^\top \mathbf{A}_i \boldsymbol{x}_i \leq 0$ for any $\boldsymbol{x}_i \in X_i$ and player $i \in [\![n]\!]$, in turn implying that $\sum_{i=1}^{n} \boldsymbol{\mu}^\top \mathbf{A}_i \boldsymbol{x}_i \leq 0$. So, combining with (7),

$$\lambda \sum_{i=1}^{n} \max_{\boldsymbol{x}_i^* \in X_i} \bar{\boldsymbol{\mu}}^\top \mathbf{A}_i \boldsymbol{x}_i^* \leq \boldsymbol{c}^\top \bar{\boldsymbol{\mu}} - \boldsymbol{c}^\top \boldsymbol{\mu} + \frac{1}{T} \left( \sum_{i=1}^{n} \mathrm{Reg}_{X_i}^T + \mathrm{Reg}_\Xi^T \right). \quad (8)$$

Finally, given that $\max_{\boldsymbol{x}_{i'}^* \in X_{i'}} \bar{\boldsymbol{\mu}}^\top \mathbf{A}_{i'} \boldsymbol{x}_{i'}^* \geq \bar{\boldsymbol{\mu}}^\top \mathbf{A}_{i'} \boldsymbol{d}_{i'} = 0$ for any player $i'$, it follows that

$$\sum_{i=1}^{n} \max_{\boldsymbol{x}_i^* \in X_i} \bar{\boldsymbol{\mu}}^\top \mathbf{A}_i \boldsymbol{x}_i^* \geq \max_{i \in [\![n]\!]} \max_{\boldsymbol{x}_i^* \in X_i} \bar{\boldsymbol{\mu}}^\top \mathbf{A}_i \boldsymbol{x}_i^*,$$

and (8) implies Item 2 of the statement. $\qquad\square$

**Bounding the regret of the players** To instantiate Theorem D.1 for our problem, we first bound the regret of Player min in (L1) in terms of the magnitude of the Lagrange multiplier. As we explained in Appendix D, the regret minimization problem faced by Player min can be decomposed into subproblems over the sequence-form polytope, one for each player. To keep the exposition self-contained, let us first recall the standard regret guarantee of CFR under the sequence-form polytope.

**Proposition E.1** ([106])**.** *Let* $\mathrm{Reg}_{X_i}^T$ *be the regret cumulated by CFR [106] over the sequence-form polytope* $X_i$*. Then, for any* $T \in \mathbb{N}$*,*

$$\mathrm{Reg}_{X_i}^T \leq C D_i |\Sigma_i| \sqrt{T},$$

*where* $D_i > 0$ *is the range of utilities observed by player* $i \in [\![n]\!]$ *and* $C > 0$ *is an absolute constant.*

An analogous regret guarantee holds for online mirror descent [91]. As a result, given that the range of observed utilities for each player is $O(\lambda)$, for a fixed Lagrange multiplier $\lambda$, we arrive at the following result.

**Corollary E.2.** *If all players employ CFR, the regret of Player min in* (L1) *can be bounded as*

$$\mathrm{Reg}_X^T = \sum_{i=1}^{n} \mathrm{Reg}_{X_i}^T = O(\lambda \sqrt{T}).$$

Here, we used the simple fact that regret over a Cartesian product can be expressed as the sum of the regrets over each individual set [29].

**Bounding the regret of the mediator** We next turn our attention to the regret minimization problem faced by the mediator. The complexity of this problem depends on the underlying notion of equilibrium at hand. In particular, for the correlated equilibrium concepts studied in this paper—namely, NFCCE, EFCCE and EFCE, we employ the framework of *DAG-form sequential decision problem (DFSDP)* [103]. In particular, DFSDP is a sequential decision process over a DAG. We will denote by $E$ the set of edges of the DAG, and by $\mathcal{S}$ the set of its nodes; we refer to Zhang et al. [103] for the precise definitions. A crucial structural observation is that a DFSDP can be derived from the probability simplex after repeated Cartesian products and *scaled-extensions* operations; suitable DGFs arising from such operations have been documented [32]. As such, we can use the following guarantee shown by Zhang et al. [103].

**Proposition E.3** ([103]). *Let* $\text{Reg}_\Xi^T$ *be the regret cumulated by the regret minimization algorithm* $\mathcal{R}_\Xi$ *used by the mediator (Player* max *in* (L1)) *up to time* $T \in \mathbb{N}$. *Then, if* $\mathcal{R}_\Xi$ *is instantiated using CFR, or suitable variants thereof,* $\text{Reg}_\Xi^T = O(|\mathcal{S}|\sqrt{T}D)$, *where* $D$ *is the range of the utilities observed by the mediator. Further, the iteration complexity is* $O(|E|)$.

As a result, combining Theorem D.1 with Proposition E.3 and Corollary E.2, and setting the Lagrange multiplier $\lambda := T^{1/4}$, we establish the statement of Corollary 3.3.

**Faster rates through optimism**   Next, to obtain Corollary 3.4, let us parameterize the regret of optimistic gradient descent in terms of the maximum utility, which can be directly extracted from the work of Rakhlin and Sridharan [87].

**Proposition E.4.** *If both agents in the saddle-point problem* (L1) *employ optimistic gradient descent with a sufficiently small learning rate* $\eta > 0$, *then the sum of their regrets is bounded by* $O(\lambda)$, *for any fixed* $\lambda > 0$.

*Proof Sketch.* By the RVU bound [94, 87], the sum of the agents' regrets can be bounded as $(\text{diam}_\Xi^2 + \text{diam}_X^2)/\eta$, for a sufficiently small $\eta = O(1/\lambda)$, where $\text{diam}_\Xi$ and $\text{diam}_X$ denote the $\ell_2$-diameter of $\Xi$ and $X$, respectively. Thus, taking $\eta = \Theta(1/\lambda)$ to be sufficiently small implies the statement.   $\square$

As a result, taking $\lambda := T^{1/2}$ and applying Theorem D.1 leads to the bound claimed in Corollary 3.4.

**Last-iterate convergence**   Finally, let us explain how known guarantees can be applied to establish Theorem 3.5. By applying [4], it follows that for a sufficiently small learning rate $\eta = O(1/\lambda)$ there is an iterate of optimistic gradient descent with $O\left(\frac{1}{\eta\sqrt{T}}\right)$ duality gap. Thus, setting $\eta = \Theta(1/\lambda)$ to be sufficiently small we get that the duality gap is bounded by $O\left(\frac{\lambda}{\sqrt{T}}\right)$. As a result, for $\lambda := T^{1/4}$ Theorem D.1 implies a rate of $T^{-1/4}$, as claimed in Theorem 3.5. We remark that while the guarantee of Theorem D.1 has been expressed in terms of the sum of the agents' regrets, the conclusion readily applies for any pair of strategies $(\bar{\boldsymbol{\mu}}, \bar{\boldsymbol{x}}) \in \Xi \times X$ by replacing the term $\text{Reg}_\Xi^T + \sum_{i=1}^n \text{Reg}_{X_i}$ with the duality gap of $(\bar{\boldsymbol{\mu}}, \bar{\boldsymbol{x}})$ with respect to (L1) (for the fixed value of $\lambda$). We further note that once the desirable duality gap $O\left(\frac{1}{\eta\sqrt{T}}\right)$ has been reached, one can fix the players' strategies to obtain a last-iterate guarantee as well.

## E.1   An alternative approach

In this subsection, we highlight an alternative approach for solving the saddle-point (L1) using regret minimization. In particular, we first observe that it can expressed as the saddle-point problem

$$\max_{\boldsymbol{\mu} \in \Xi} \min_{\bar{\boldsymbol{x}}_i \in \bar{X}_i : i \in [\![n]\!]} \quad \boldsymbol{c}^\top \boldsymbol{\mu} - \sum_{i=1}^n \boldsymbol{\mu}^\top \mathbf{A}_i \bar{\boldsymbol{x}}_i, \tag{9}$$

where $\bar{X}_i := \{\lambda_i \boldsymbol{x}_i : \lambda_i \in [0, K], \boldsymbol{x}_i \in X_i\}$ is the *conic hull* of $X_i$ truncated to a sufficiently large parameter $K > 0$. Analogously to our approach in Appendix D, suitably tuning the value of $K$ will allow us trade off between the optimality gap and the equilibrium gap. In this context, we point out below that how to construct a regret minimizer over a conic hull.

**Regret minimization over conic hulls**   Suppose that $\mathcal{R}_{X_i}$ is a regret minimizer over $X_i$ and $\mathcal{R}_+$ is a regret minimizer over the interval $[0, K]$. Based on those two regret minimizers, Algorithm 2 shows how to construct a regret minimizer over the conic hull $\bar{X}_i$. More precisely, Algorithm 2 follows the convention that a generic regret minizer $\mathcal{R}$ interacts with its environment via the following two subroutines:

- $\mathcal{R}.\text{NEXTSTRATEGY}$: $\mathcal{R}$ returns the next strategy based on its internal state; and

- $\mathcal{R}.\text{OBSERVEUTILITY}(\boldsymbol{u}^{(t)})$: $\mathcal{R}$ receives as input from the environment a (compatible) utility vector $\boldsymbol{u}^{(t)}$ at time $t \in \mathbb{N}$.

---
**ALGORITHM 2:** Regret minimization over a conic hull
---
**1 function** NEXTSTRATEGY()
**2**     $\lambda_i \leftarrow \mathcal{R}_+.\text{NEXTSTRATEGY}()$
**3**     $\boldsymbol{x}_i \leftarrow \mathcal{R}_{X_i}.\text{NEXTSTRATEGY}()$
**4**     **return** $\bar{\boldsymbol{x}}_i \coloneqq \lambda_i \boldsymbol{x}_i$
**5 function** OBSERVEUTILITY($\boldsymbol{u}_i$)
**6**     $\mathcal{R}_{X_i}.\text{OBSERVEUTILITY}(\boldsymbol{u}_i)$
**7**     $\mathcal{R}_+.\text{OBSERVEUTILITY}(\boldsymbol{u}_i^\top \boldsymbol{x}_i)$
---

The formal statement regarding the cumulated regret of Algorithm 2 below is cast in the framework of *regret circuits* [29].

**Proposition E.5** (Regret circuit for the conic hull). *Suppose that* $\text{Reg}_{X_i}^T$ *and* $\text{Reg}_+^T$ *is the cumulative regret incurred by* $\mathcal{R}_{X_i}$ *and* $\mathcal{R}_+$*, respectively, up to a time horizon* $T \in \mathbb{N}$*. Then, the regret* $\text{Reg}_{\bar{X}_i}^T$ *of* $\mathcal{R}_{\bar{X}_i}$ *constructed based on Algorithm 2 can be bounded as*

$$\text{Reg}_{\bar{X}_i}^T \leq K \max\{0, \text{Reg}_{X_i}^T\} + \text{Reg}_+^T.$$

*Proof.* By construction, we have that $\text{Reg}_{\bar{X}_i}^T$ is equal to

$$\max_{\bar{\boldsymbol{x}}_i^* \in \bar{X}_i} \left\{ \sum_{t=1}^T \langle \bar{\boldsymbol{x}}_i^* - \bar{\boldsymbol{x}}_i^{(t)}, \boldsymbol{u}^{(t)} \rangle \right\} = \max_{\lambda_i^* \boldsymbol{x}_i^* \in \bar{X}_i} \left\{ \sum_{t=1}^T \langle \lambda_i^* \boldsymbol{x}_i^* - \lambda_i^{(t)} \boldsymbol{x}_i^{(t)}, \boldsymbol{u}_i^{(t)} \rangle \right\}$$

$$= \max_{\lambda_i^* \boldsymbol{x}_i^* \in \bar{X}_i} \left\{ \lambda_i^* \sum_{t=1}^T \langle \boldsymbol{x}_i^* - \boldsymbol{x}_i^{(t)}, \boldsymbol{u}_i^{(t)} \rangle + (\lambda_i^* - \lambda_i^{(t)})(\boldsymbol{u}_i^{(t)})^\top \boldsymbol{x}_i^{(t)} \right\}$$

$$\leq K \max\{0, \text{Reg}_{X_i}^T\} + \text{Reg}_+^T,$$

where the last derivation uses that $\lambda_i^* \in [0, K]$. $\qquad\qquad\square$

As a result, by suitable instantiating $\mathcal{R}_{X_i}$ and $\mathcal{R}_+$ (*e.g.*, using Proposition E.1), the regret circuit of Proposition E.5 enables us to construct a regret minimizer over $\bar{X}_i$ with regret bounded as $O(K\sqrt{T})$. In turn, this directly leads to a regret minimizer for Player min in (9) with regret bounded by $O(K\sqrt{T})$. We further remark that Theorem D.1 can be readily cast in terms of the saddle-point problem (9) as well, parameterized now by $K$ instead of $\lambda$. As a result, convergence bounds such as Corollary 3.3 also apply to regret minimizers constructed via conic hulls.

# F Description of game instances

In this section, we provide a detailed description of the game instances used in our experiments in Section 4.1.

## F.1 Liar's dice (D), Goofspiel (GL), Kuhn poker (K), and Leduc poker (L)

**Liar's dice**    At the start of the game, each of the three players rolls a fair $k$-sided die privately. Then, the players take turns making claims about the outcome of their roll. The first player starts by stating any number from 1 to $k$ and the minimum number of dice they believe are showing that value among all players. On their turn, each player has the option to make a higher claim or challenge the previous claim by calling the previous player a "liar." A claim is higher if the number rolled is higher or the number of dice showing that number is higher. If a player challenges the previous claim and the claim is found to be false, the challenger is rewarded +1 and the last bidder receives a penalty of -1. If the claim is true, the last bidder is rewarded +1, and the challenger receives -1. All other players receive 0 reward. We consider two instances of the game, one with $k = 2$ (D32) and one with $k = 3$ (D33).

**Goofspiel**   This is a variant of Goofspiel with limited information. In this variation, in each turn the players do not reveal the cards that they have played. Instead, players show their cards to a neutral umpire, who then decides the winner of the round by determining which card is the highest. In the event of a tie, the umpire directs the players to divide the prize equally among the tied players, similar to the Goofspiel game. The instance GL3 which we employ has 3 players, 3 ranks, and imperfect information.

**Kuhn poker**   Three-player Kuhn Poker, an extension of the original two-player version proposed by Kuhn [61], is played with three players and $r$ cards. Each player begins by paying one chip to the pot and receiving a single private card. The first player can check or bet (*i.e.*, putting an additional chip in the pot). Then, the second player can check or bet after a first player's check, or fold/call the first player's bet. The third player can either check or bet if no previous bet was made, otherwise they must fold or call. At the showdown, the player with the highest card who has not folded wins all the chips in the pot. We use the instance K35 which has rank $r = 5$.

**Leduc poker**   In our instances of the three-player Leduc poker the deck consists of $s$ suits with $r$ cards each. Our instances are parametric in the maximum number of bets $b$, which in limit hold'em is not necessarily tied to the number of players. The maximum number of raise per betting round can be either 1, 2 or 3. At the beginning of the game, players each contribute one chip to the pot. The game proceeds with two rounds of betting. In the first round, each player is dealt a private card, and in the second round, a shared board card is revealed. The minimum raise is set at 2 chips in the first round and 4 chips in the second round. We denote by L3$brs$ an instance with three players with $b$ bets per round, $r$ ranks, and $s$ suits. We employ the following five instances: L3132, L3133, L3151, L3223, L3523.

### F.2   Battleship game (B) and Sheriff game (S)

**Battleship**   The game is a general-sum version of the classic game Battleship, where two players take turns placing ships of varying sizes and values on two separate grids of size $h \times w$, and then take turns firing at their opponent. Ships which have been hit at all their tiles are considered destroyed. The game ends when one player loses all their ships, or after each player has fired $r$ shots. Each player's payoff is determined by the sum of the value of the opponent's destroyed ships minus $\gamma \geq 1$ times the number of their own lost ships. We denote by B$phwr$ an instance with $p$ players on a grid of size $h \times w$, one unit-size ship for each player, and $r$ rounds. We consider the following four instances: B2222, B2322, B2323, B2324.

**Sheriff**   This game is a simplified version of the *Sheriff of Nottingham* board game, which models the interaction between a *Smuggler*—who is trying to smuggle illegal items in their cargo—and the *Sheriff*—who's goal is stopping the Smuggler. First, the Smuggler has to decide the number $n \in \{0, \ldots, N\}$ of illegal items to load on the cargo. Then, the Sheriff decides whether to inspect the cargo. If they choose to inspect, and find illegal goods, the Smuggler has to pay $p \cdot n$ to the Sheriff. Otherwise, the Sheriff has to compensate the Smuggler with a reward of $s$. If the Sheriff decides not to inspect the cargo, the Sheriff's utility is 0, and the Smuggler's utility is $v \cdot n$. After the Smuggler has loaded the cargo, and before the Sheriff decides whether to inspect, the Smuggler can try to bribe the Sheriff to avoid the inspection. In particular, they engage in $r$ rounds of bargaining and, for each round $i$, the Smuggler proposes a bribe $b_i \in \{0, \ldots, B\}$, and the Sheriff accepts or declines it. Only the proposal and response from the final round $r$ are executed. If the Sheriff accepts a bribe $b_r$ then they get $b_r$, while the Smuggler's utility is $vn - b_r$. Further details on the game can be found in Farina et al. [30]. An instance S$pNBr$ has $p$ players, $N$ illegal items, a maximum bribe of $B$, and $r$ rounds of bargaining. The other parameters are $v = 5$, $p = 1$, $s = 1$ and they are fixed across all instances. We employ the following five instances: S2122, S2123, S2133, S2254, S2264.

### F.3   The double-dummy bridge endgame (TP)

The double-dummy bridge endgame is a benchmark introduced by Zhang et al. [101] which simulates a bridge endgame scenario. The game uses a fixed deck of playing cards that includes three ranks (2, 3, 4) of each of four suits (spades, hearts, diamonds, clubs). Spades are designated as the trump suit. There are four players involved: two defenders sitting across from each other, the dummy, and the declarer. The dummy's actions will be controlled by the declarer, so there are only

three players actively participating. However, for clarity, we will refer to all four players throughout this section.

The entire deck of cards is randomly dealt to the four players. We study the version of the game that has perfect information, meaning that all players' cards are revealed to everyone, creating a game in which all information is public (*i.e.*, a *double-dummy game*). The game is played in rounds called *tricks*. The player to the left of the declarer starts the first trick by playing a card. The suit of this card is known as the *lead suit*. Going in clockwise order, the other three players play a card from their hand. Players must play a card of the lead suit if they have one, otherwise, they can play any card. If a spade is played, the player with the highest spade wins the trick. Otherwise, the highest card of the lead suit wins the trick. The winner of each trick then leads the next one. At the end of the game, each player earns as many points as the number of tricks they won. In this adversarial team game, the two defenders are teammates and play against the declarer, who controls the dummy.

The specific instance that we use (*i.e.*, TP3) has 3 ranks and perfect information. The dummy's hand is fixed as 2♠ 2♥ 3♥.

## F.4 Ridesharing game (RS)

This benchmark was first introduced by Zhang et al. [101], and it models the interaction between two drivers competing to serve requests on a road network. The network is defined as an undirected graph $G^{\mathtt{U}} = (V^{\mathtt{U}}, E^{\mathtt{U}})$, where each vertex $v \in V^{\mathtt{U}}$ corresponds to a ride request to be served. Each request has a reward in $\mathbb{R}_{\geq 0}$, and each edge in the network has some cost. The first driver who arrives on node $v \in V^{\mathtt{U}}$ serves the corresponding ride, and receives the corresponding reward. Once a node has been served, it stays clean until the end of the game. The game terminates when all nodes have been cleared, or when a timeout is met (*i.e.*, there's a fixed time horizon $T$). If the two drivers arrive simultaneously on the same vertex they both get reward 0. The final utility of each driver is computed as the sum of the rewards obtained from the beginning until the end of the game. The initial position of the two drivers is randomly selected at the beginning of the game. Finally, the two drivers can observe each other's position only when they are simultaneously on the same node, or they are in adjacent nodes.

Ridesharing games are particularly well-suited to study the computation of optimal equilibria because they are *not* triangle-free [28].

**Setup** We denote by RS $p$ $i$ $T$ a ridesharing instance with $p$ drivers, network configuration $i$, and horizon $T$. Parameter $i \in \{1, 2\}$ specifies the graph configuration. We consider the two network configurations of Zhang et al. [101], their structure is reported in Figure 2. All edges are given unitary cost. We consider a total of four instances: RS212, RS222, RS213, RS223.

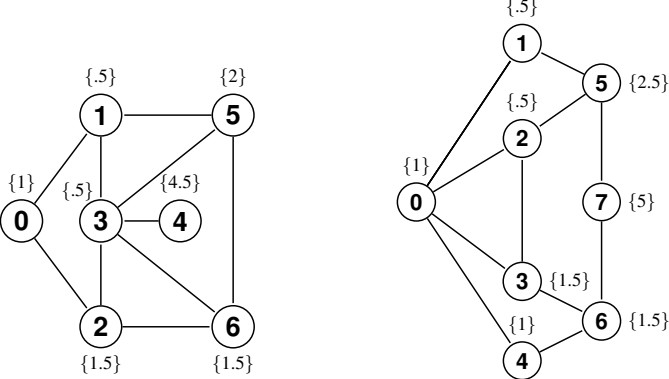

Figure 2: *Left*: configuration 1 (used for RS212, RS213). *Right*: configuration 2 (used for RS222, RS223). In both cases the position of the two drivers is randomly chosen at the beginning of the game, edge costs are unitary, and the reward for each node is indicated between curly brackets.

# G Additional experimental results

## G.1 Investigation of lower equilibrium approximation

In Table 2 we show results, using the same format as Table 1 shown in the body, for the case in which the approximation $\varepsilon$ is set to be $0.1\%$ of the payoff range of the game, as opposed to the $1\%$ threshold of the body.

Table 2: Comparison between the linear-programming-based algorithm ('LP') of Zhang and Sandholm [100] and our learning-based approach ('Ours'), for the problem of computing an approximate optimal equilibrium within tolerance $\varepsilon$ set to $0.1\%$ of the payoff range of the game.

| Game | # Nodes | NFCCE LP | NFCCE Ours | EFCCE LP | EFCCE Ours | EFCE LP | EFCE Ours | COMM LP | COMM Ours | CERT LP | CERT Ours |
|---|---|---|---|---|---|---|---|---|---|---|---|
| B2222 | 1573 | 0.00s | 0.00s | 0.00s | 0.02s | 0.00s | 0.03s | 3.00s | 1m 5s | 0.00s | 0.04s |
| B2322 | 23 839 | 1.00s | 0.02s | 3.00s | 1.42s | 9.00s | 4.11s | timeout | 17m 30s | 2.00s | 2.82s |
| B2323 | 254 239 | 6.00s | 0.66s | 1m 29s | 30.04s | 3m 40s | 1m 28s | timeout | timeout | 39.00s | 1m 24s |
| B2324 | 1 420 639 | 41.00s | 5.25s | timeout | 5m 49s | timeout | timeout | timeout | timeout | timeout | timeout |
| D32 | 1017 | 0.00s | 0.03s | 0.00s | 0.04s | 14.00s | 0.92s | 1.00s | 0.26s | 0.00s | 0.03s |
| D33 | 27 622 | 3m 22s | 44.41s | timeout | 10m 27s | timeout | timeout | timeout | 16m 38s | 6.00s | 6.87s |
| GL3 | 7735 | 0.00s | 0.06s | 1.00s | 0.07s | 0.00s | 0.06s | timeout | 36.83s | 0.00s | 0.11s |
| K35 | 1501 | 55.00s | 2.46s | 53.00s | 3.05s | 1m 5s | 2.99s | 1.00s | 0.09s | 0.00s | 0.02s |
| L3132 | 8917 | 28.00s | 2.13s | 11m 26s | 22.14s | 9m 41s | 26.68s | 13.00s | 15.41s | 1.00s | 0.62s |
| L3133 | 12 688 | 45.00s | 2.83s | timeout | 35.86s | 26m 52s | 22.31s | 17.00s | 15.27s | 1.00s | 1.25s |
| L3151 | 19 981 | timeout | 54.66s | timeout | timeout | timeout | timeout | timeout | 1m 15s | 2.00s | 0.91s |
| L3223 | 15 659 | 5.00s | 1.73s | 1m 21s | 8.58s | 2m 38s | 20.44s | 26.00s | 1m 43s | 1.00s | 2.00s |
| L3523 | 1 299 005 | timeout | 4m 4s | timeout | timeout | timeout | timeout | timeout | timeout | timeout | timeout |
| S2122 | 705 | 0.00s | 0.00s | 0.00s | 0.02s | 0.00s | 0.07s | 3.00s | 6.14s | 0.00s | 0.03s |
| S2123 | 4269 | 0.00s | 0.02s | 1.00s | 0.14s | 1.00s | 0.37s | 1m 51s | 10m 8s | 1.00s | 0.41s |
| S2133 | 9648 | 1.00s | 0.05s | 3.00s | 0.17s | 4.00s | 0.95s | timeout | timeout | 3.00s | 1.99s |
| S2254 | 712 552 | 2m 0s | 32.14s | timeout | 42.65s | timeout | 9m 2s | timeout | timeout | timeout | 6m 50s |
| S2264 | 1 303 177 | 3m 48s | 57.76s | timeout | 1m 16s | timeout | timeout | timeout | timeout | timeout | timeout |
| TP3 | 910 737 | 1m 43s | 14.28s | timeout | 20.81s | timeout | 26.28s | timeout | timeout | timeout | 52.76s |
| RS212 | 598 | 0.00s | 0.00s | 0.00s | 0.00s | 0.00s | 0.00s | 2.00s | 0.02s | 0.00s | 0.00s |
| RS222 | 734 | 0.00s | 0.01s | 0.00s | 0.01s | 0.00s | 0.02s | 3.00s | 0.03s | 0.00s | 0.00s |
| RS213 | 6274 | timeout | 43.46s | timeout | 45.00s | timeout | 2m 28s | 7m 30s | 27.19s | 0.00s | 0.03s |
| RS223 | 6238 | timeout | timeout | timeout | timeout | timeout | timeout | 9m 16s | 15.68s | 1.00s | 0.05s |

We observe that none of the results change qualitatively when this increased precision is considered.

## G.2 Game values and size of Mediator's strategy space

Tables 3 and 4 reports optimal equilibrium values for all games and all equilibrium concepts for which the LP algorithm was able to compute an exact value (We restrict to the cases solvable by LP because the Lagrangian relaxations only compute an $\varepsilon$-equilibrium, but the mediator objective in an $\varepsilon$-equilibrium could be arbitrarily far away from the mediator objective in an exact equilibrium). We hope that these will be good references for future researchers interested in this topic. Table 5 reports the size of the strategy space of the mediator player in the two-player zero-sum game that captures the computation of optimal equilibria, in terms of the number of decision points and edges. For correlated notions, this number may be exponential in the original game size; for communication and certification notions, it will always be polynomial.

Table 3: Optimal equilibrium value for correlated equilibrium concepts. 'Pl. 1' is the utility for Player 1 in the Player 1-optimal equilibrium. 'Pl. 2' and 'Pl. 3' are similar. In two-player games, 'SW' is the welfare of the welfare-maximizing equilibrium. (these three values, of course, may come from three different equilibria.) The three-player games are zero-sum, so optimizing welfare makes no sense (the welfare is always zero).

| | NFCCE | | | EFCCE | | | EFCE | | |
|---|---|---|---|---|---|---|---|---|---|
| Game | Pl. 1 | Pl. 2 | SW | Pl. 1 | Pl. 2 | SW | Pl. 1 | Pl. 2 | SW |
| B2222 | 0.281 | 0.094 | 0.000 | $-0.027$ | $-0.338$ | $-0.525$ | $-0.031$ | $-0.338$ | $-0.525$ |
| B2322 | 0.181 | 0.097 | 0.000 | $-0.043$ | $-0.123$ | $-0.317$ | $-0.045$ | $-0.123$ | $-0.317$ |
| B2323 | 0.250 | 0.125 | 0.000 | 0.000 | $-0.125$ | $-0.375$ | $-0.001$ | $-0.125$ | $-0.375$ |
| B2324 | 0.306 | 0.139 | 0.000 | — | — | — | — | — | — |
| S2122 | 11.636 | 5.999 | 13.636 | 7.652 | 5.043 | 9.565 | 7.262 | 3.841 | 9.078 |
| S2123 | 11.636 | 5.999 | 13.636 | 8.000 | 5.191 | 10.000 | 8.000 | 4.611 | 10.000 |
| S2133 | 15.182 | 6.992 | 18.182 | 12.000 | 6.557 | 15.000 | 12.000 | 6.407 | 15.000 |
| S2254 | 23.571 | 12.830 | 28.571 | — | — | — | — | — | — |
| S2264 | 27.333 | 13.840 | 33.333 | — | — | — | — | — | — |
| U212 | 3.123 | 3.123 | 6.010 | 3.071 | 3.071 | 6.010 | 3.071 | 3.071 | 6.010 |
| U213 | — | — | — | — | — | — | — | — | — |
| U222 | 3.765 | 3.765 | 7.188 | 3.719 | 3.719 | 7.176 | 3.719 | 3.719 | 7.176 |
| U223 | — | — | — | — | — | — | — | — | — |
| | Pl. 1 | Pl. 2 | Pl. 3 | Pl. 1 | Pl. 2 | Pl. 3 | Pl. 1 | Pl. 2 | Pl. 3 |
| D32 | 0.250 | 0.250 | 0.131 | 0.250 | 0.250 | 0.000 | 0.250 | 0.250 | 0.000 |
| D33 | 0.422 | 0.284 | 0.239 | — | — | — | — | — | — |
| GL3 | 2.505 | 2.505 | 2.505 | 2.476 | 2.476 | 2.476 | 2.467 | 2.467 | 2.467 |
| K35 | $-0.011$ | 0.017 | 0.057 | $-0.016$ | 0.015 | 0.052 | $-0.016$ | 0.013 | 0.052 |
| L3132 | 0.571 | 0.504 | 0.606 | 0.519 | — | — | 0.467 | 0.422 | — |
| L3133 | 0.419 | 0.348 | 0.416 | — | — | — | — | — | — |
| L3151 | — | — | — | — | — | — | — | — | — |
| L3223 | 1.079 | 0.992 | 1.146 | 0.984 | 0.959 | 1.033 | 0.887 | 0.883 | 0.861 |
| L3523 | — | — | — | — | — | — | — | — | — |
| TP3 | 1.466 | 1.477 | 1.037 | — | — | — | — | — | — |

Table 4: Optimal equilibrium value for communication and certification equilibrium concepts. 'Pl. 1', 'Pl. 2', 'Pl. 3', and 'SW' have the same meaning as in the previous table.

| | COMM | | | NFCCERT | | | CCERT | | | CERT | | |
|---|---|---|---|---|---|---|---|---|---|---|---|---|
| Game | Pl. 1 | Pl. 2 | SW | Pl. 1 | Pl. 2 | SW | Pl. 1 | Pl. 2 | SW | Pl. 1 | Pl. 2 | SW |
| B2222 | $-0.187$ | $-0.562$ | $-0.750$ | 0.281 | 0.094 | 0.000 | $-0.027$ | $-0.338$ | $-0.525$ | $-0.027$ | $-0.338$ | $-0.525$ |
| B2322 | — | — | — | 0.181 | 0.097 | 0.000 | $-0.043$ | $-0.123$ | $-0.317$ | $-0.043$ | $-0.123$ | $-0.317$ |
| B2323 | — | — | — | 0.250 | 0.125 | 0.000 | 0.000 | $-0.125$ | $-0.375$ | 0.000 | $-0.125$ | $-0.375$ |
| B2324 | — | — | — | 0.306 | 0.139 | 0.000 | — | — | — | — | — | — |
| S2122 | 0.820 | 0.000 | 0.820 | 50.000 | 8.508 | 50.000 | 8.000 | 5.191 | 10.000 | 8.000 | 4.611 | 10.000 |
| S2123 | 0.820 | 0.000 | 0.820 | 50.000 | 8.508 | 50.000 | 8.000 | 5.191 | 10.000 | 8.000 | 4.611 | 10.000 |
| S2133 | — | — | — | 50.000 | 8.671 | 50.000 | 12.000 | 6.557 | 15.000 | 12.000 | 6.407 | 15.000 |
| S2254 | — | — | — | 100.000 | 17.284 | 100.000 | 20.000 | 12.190 | 25.000 | — | — | — |
| S2264 | — | — | — | 100.000 | 17.442 | 100.000 | — | — | — | — | — | — |
| U212 | 3.184 | 3.143 | 6.173 | 3.184 | 3.173 | 6.173 | 3.184 | 3.159 | 6.173 | 3.184 | 3.143 | 6.173 |
| U213 | 5.160 | 5.171 | 9.592 | 5.316 | 5.429 | 9.622 | 5.204 | 5.298 | 9.622 | 5.196 | 5.276 | 9.622 |
| U222 | 4.023 | 3.812 | 7.594 | 4.023 | 3.930 | 7.594 | 4.023 | 3.905 | 7.594 | 4.023 | 3.839 | 7.594 |
| U223 | 6.537 | 6.326 | 11.464 | 6.867 | 6.783 | 11.516 | 6.631 | 6.582 | 11.513 | 6.576 | 6.398 | 11.485 |
| | Pl. 1 | Pl. 2 | Pl. 3 | Pl. 1 | Pl. 2 | Pl. 3 | Pl. 1 | Pl. 2 | Pl. 3 | Pl. 1 | Pl. 2 | Pl. 3 |
| D32 | 0.250 | 0.250 | 0.042 | 0.500 | 0.250 | 0.250 | 0.250 | 0.250 | 0.250 | 0.250 | 0.250 | 0.250 |
| D33 | — | — | — | 0.580 | 0.296 | 0.284 | 0.444 | 0.296 | 0.284 | 0.432 | 0.296 | 0.272 |
| GL3 | — | — | — | 2.505 | 2.505 | 2.505 | 2.505 | 2.505 | 2.505 | 2.467 | 2.468 | 2.468 |
| K35 | 0.022 | 0.050 | 0.088 | 0.092 | 0.106 | 0.169 | 0.092 | 0.090 | 0.169 | 0.086 | 0.090 | 0.169 |
| L3132 | 0.646 | 0.618 | 0.723 | 0.853 | 0.779 | 0.802 | 0.853 | 0.779 | 0.802 | 0.853 | 0.779 | 0.802 |
| L3133 | 0.441 | 0.459 | 0.590 | 0.646 | 0.654 | 0.709 | 0.646 | 0.654 | 0.709 | 0.646 | 0.654 | 0.709 |
| L3151 | — | — | — | 0.179 | 0.197 | 0.222 | 0.179 | 0.182 | 0.222 | 0.171 | 0.182 | 0.222 |
| L3223 | 1.011 | 0.915 | 1.020 | 1.379 | 1.556 | 1.451 | 1.379 | 1.556 | 1.451 | 1.379 | 1.556 | 1.451 |
| L3523 | — | — | — | 2.000 | 2.000 | 2.000 | — | — | — | — | — | — |
| TP3 | — | — | — | 1.739 | 1.506 | 1.083 | — | — | — | — | — | — |

Table 5: Dimension of the mediator's decision space in terms of number of decision points ('Dec. pts.') and edges.

| Game | NFCCE | | EFCCE | | EFCE | | COMM | | NFCCERT | | CCERT | | CERT | |
|---|---|---|---|---|---|---|---|---|---|---|---|---|---|---|
| | Dec. pts. | Edges | Dec. pts. | Edges | Dec. pts. | Edges | Dec. pts. | Edges | Dec. pts. | Edges | Dec. pts. | Edges | Dec. pts. | Edges |
| B2222 | 1429 | 6915 | 5001 | 21868 | 4212 | 20534 | 16341 | 65577 | 663 | 2739 | 1430 | 5854 | 3590 | 14638 |
| B2322 | 11707 | 89519 | 66181 | 340619 | 67219 | 503145 | 681523 | 4090261 | 5661 | 34227 | 13940 | 84080 | 52640 | 317180 |
| B2323 | 164707 | 1022286 | 1067881 | 5446015 | 1032019 | 7271972 | — | — | 77661 | 394227 | 244340 | 1236080 | 959840 | 4853180 |
| B2324 | 1316707 | 6397418 | 8296681 | 41633816 | 8160018 | 49264667 | | | 596061 | 2467827 | 2188340 | 9012080 | 8476640 | 34920380 |
| D32 | 3956 | 33823 | 5381 | 51593 | 53402 | 536485 | 12794 | 45070 | 472 | 1796 | 504 | 1844 | 2484 | 9176 |
| D33 | 417625 | 11165451 | 1599919 | 71372690 | — | — | 2854524 | 10450812 | 11292 | 44382 | 18396 | 68937 | 135504 | 520665 |
| GL3 | 8898 | 30021 | 10680 | 37041 | 5637 | 16950 | 182289 | 547086 | 2343 | 7104 | 3138 | 9474 | 5058 | 15234 |
| K35 | 52277 | 3592121 | 60257 | 3826201 | 61217 | 4535281 | 9745 | 29235 | 1005 | 3015 | 1075 | 3225 | 2315 | 6945 |
| L3132 | 131012 | 1222128 | 689890 | 15329595 | 694381 | 8400513 | 326730 | 980190 | 7773 | 23319 | 15789 | 47367 | 32055 | 96165 |
| L3133 | 155297 | 1500087 | 1002685 | 26166405 | 1010749 | 14519676 | 365187 | 1095561 | 9960 | 29880 | 21534 | 64602 | 44868 | 134604 |
| L3151 | 1697120 | 34405970 | — | — | — | — | 1784965 | 5354895 | 18285 | 54855 | 36115 | 108345 | 72395 | 217185 |
| L3223 | 91735 | 614847 | 405691 | 5617510 | 678365 | 4999142 | 1234394 | 4004046 | 14186 | 46656 | 36298 | 118576 | 83786 | 273276 |
| L3523 | 7595335 | 58635336 | — | — | — | — | | | 1115978 | 3887736 | 5617402 | 19357364 | 14863826 | 51214448 |
| S2122 | 651 | 2903 | 2061 | 7847 | 1629 | 6227 | 4071 | 12629 | 408 | 1396 | 825 | 2627 | 1749 | 5465 |
| S2123 | 4413 | 19049 | 17883 | 72377 | 13113 | 52559 | 146631 | 454565 | 2496 | 8452 | 6873 | 21959 | 15717 | 49349 |
| S2133 | 9424 | 45931 | 40960 | 171915 | 38732 | 165859 | 778108 | 2425875 | 5112 | 18566 | 15000 | 49607 | 44084 | 139851 |
| S2254 | 617056 | 3758737 | 3974008 | 18655297 | 5470186 | 25303237 | — | — | 327992 | 1300694 | 1332064 | 4615207 | 6089698 | 19348765 |
| S2264 | 1103369 | 7284509 | 7291859 | 34837769 | — | — | | | 579182 | 2358134 | 2402695 | 8425369 | 12809119 | 40551631 |
| TP3 | 2355864 | 7145312 | 3574464 | 11720048 | 2211712 | 6714256 | | — | 1070544 | 3273072 | 1739488 | 5273184 | 3594352 | 10896416 |
| RS212 | 658 | 15410 | 658 | 10604 | 538 | 9000 | 3317 | 14338 | 213 | 902 | 182 | 768 | 182 | 768 |
| RS222 | 1625 | 55123 | 1625 | 35047 | 1495 | 32643 | 4142 | 15914 | 290 | 1098 | 252 | 952 | 252 | 952 |
| RS213 | 61122 | 95194268 | 62704 | 95250604 | 97070 | 124453191 | 365621 | 1638786 | 2459 | 10870 | 2808 | 12416 | 4288 | 19160 |
| RS223 | — | — | — | — | — | — | 299162 | 1184114 | 2778 | 10854 | 3224 | 12600 | 4500 | 17776 |

## G.3 Detailed breakdown by equilibrium and objective function (two-player games)

For each two-player game, we try three different objective functions: maximizing the utility of Player 1, maximizing the utility of Player 2, and maximizing social welfare. For each objective, we stop the optimization at the approximation level defined as 1% of the payoff range of the game.

We use online optimistic gradient descent to update the Lagrange multipliers of each player (see Appendix E.1). For each objective, we report the following information:

- The runtime of the linear-programming-based algorithm ('LP') of Zhang and Sandholm [100].

- The runtime of our algorithm where each agent (player or mediator) uses the Discounted CFR ('DCFR') algorithm set up with the hyperparameters recommended in the work by Brown and Sandholm [12]. In the table, we report the best runtime across all choices of the stepsize hyperparameter $\eta \in \{0.01, 0.1, 1.0, 10.0\}$ used in online optimistic gradient descent to update the Lagrange multipliers. The value of $\eta$ that produces the reported runtime is noted in square brackets.

- The runtime of our algorithm where each agent (player or mediator) uses the Predictive CFR$^+$ ('PCFR$^+$') algorithm of [33]. In the table, we report the best runtime across all choices of the stepsize hyperparameter $\eta \in \{0.01, 0.1, 1.0, 10.0\}$ used in online optimistic gradient descent to update the Lagrange multipliers. The value of $\eta$ that produces the reported runtime is again noted in square brackets.

### G.3.1 Results for NFCCE solution concept

| Game | Maximize Player 1's utility | | | Maximize Player 2's utility | | | Maximize social welfare | | |
|---|---|---|---|---|---|---|---|---|---|
| | LP | Ours (DCFR) | Ours (PCFR$^+$) | LP | Ours (DCFR) | Ours (PCFR$^+$) | LP | Ours (DCFR) | Ours (PCFR$^+$) |
| B2222 | 0.00s | 0.00s [1.0] | 0.00s [0.1] | 0.00s | 0.00s [1.0] | 0.00s [0.1] | 0.00s | 0.00s [1.0] | 0.00s [0.1] |
| B2322 | 0.00s | 0.03s [0.1] | 0.07s [0.1] | 0.00s | 0.04s [0.1] | 0.04s [1.0] | 0.00s | 0.01s [0.1] | 0.01s [10.0] |
| B2323 | 7.00s | 1.05s [0.1] | 1.61s [0.1] | 6.00s | 1.01s [0.1] | 1.63s [0.1] | 6.00s | 0.33s [0.1] | 0.53s [10.0] |
| B2324 | 50.00s | 15.57s [0.1] | 20.01s [0.1] | 37.00s | 12.80s [0.1] | 20.91s [0.1] | 38.00s | 2.73s [0.1] | 4.57s [1.0] |
| S2122 | 0.00s | 0.00s [0.1] | 0.00s [0.1] | 0.00s | 0.00s [0.1] | 0.00s [0.1] | 0.00s | 0.00s [0.1] | 0.00s [0.1] |
| S2123 | 0.00s | 0.01s [1.0] | 0.01s [0.1] | 0.00s | 0.01s [0.1] | 0.01s [0.1] | 0.00s | 0.01s [0.1] | 0.01s [0.1] |
| S2133 | 1.00s | 0.02s [1.0] | 0.02s [0.1] | 1.00s | 0.02s [0.1] | 0.03s [0.1] | 1.00s | 0.02s [1.0] | 0.02s [0.1] |
| S2254 | 2m 1s | 6.96s [0.1] | 11.43s [0.1] | 1m 14s | 10.43s [0.1] | 17.72s [0.1] | 1m 58s | 7.43s [0.1] | 11.88s [0.1] |
| S2264 | 3m 36s | 13.96s [0.1] | 23.25s [0.1] | 2m 24s | 18.46s [0.1] | 35.04s [0.1] | 3m 43s | 11.74s [0.1] | 17.91s [0.1] |
| RS212 | 0.00s | 0.00s [1.0] | 0.00s [1.0] | 0.00s | 0.00s [10.0] | 0.00s [1.0] | 0.00s | 0.00s [10.0] | 0.00s [1.0] |
| RS222 | 0.00s | 0.00s [1.0] | 0.01s [0.1] | 0.00s | 0.00s [1.0] | 0.01s [1.0] | 0.00s | 0.00s [0.01] | 0.01s [0.01] |
| RS213 | timeout | 34.52s [1.0] | 1m 9s [0.1] | timeout | 20.29s [1.0] | 41.66s [0.1] | timeout | 14.68s [0.1] | 35.36s [0.1] |
| RS223 | timeout | timeout [—] | timeout [—] | timeout | timeout [—] | timeout [—] | timeout | timeout [—] | timeout [—] |

### G.3.2 Results for EFCCE solution concept

| Game | Maximize Player 1's utility | | | Maximize Player 2's utility | | | Maximize social welfare | | |
|---|---|---|---|---|---|---|---|---|---|
| | LP | Ours (DCFR) | Ours (PCFR$^+$) | LP | Ours (DCFR) | Ours (PCFR$^+$) | LP | Ours (DCFR) | Ours (PCFR$^+$) |
| B2222 | 0.00s | 0.01s [1.0] | 0.02s [1.0] | 0.00s | 0.01s [1.0] | 0.02s [1.0] | 0.00s | 0.01s [1.0] | 0.01s [1.0] |
| B2322 | 3.00s | 0.41s [1.0] | 1.13s [1.0] | 3.00s | 0.69s [1.0] | 1.19s [1.0] | 3.00s | 0.69s [1.0] | 1.33s [1.0] |
| B2323 | 1m 35s | 9.11s [1.0] | 22.38s [1.0] | 1m 30s | 12.39s [1.0] | 23.59s [1.0] | 1m 21s | 14.23s [1.0] | 20.30s [1.0] |
| B2324 | timeout | 1m 53s [1.0] | 3m 35s [1.0] | timeout | 2m 44s [1.0] | 4m 29s [1.0] | timeout | 3m 1s [1.0] | 4m 16s [1.0] |
| S2122 | 0.00s | 0.01s [1.0] | 0.01s [1.0] | 0.00s | 0.00s [0.1] | 0.01s [0.1] | 0.00s | 0.01s [1.0] | 0.02s [1.0] |
| S2123 | 1.00s | 0.03s [1.0] | 0.19s [0.1] | 1.00s | 0.09s [0.1] | 0.05s [0.1] | 1.00s | 0.06s [1.0] | 0.16s [1.0] |
| S2133 | 3.00s | 0.12s [1.0] | 0.31s [1.0] | 2.00s | 0.20s [0.1] | 0.22s [0.1] | 3.00s | 0.11s [1.0] | 0.41s [1.0] |
| S2254 | timeout | 28.81s [1.0] | 27.31s [0.1] | timeout | 37.43s [0.1] | 53.08s [0.1] | timeout | 22.01s [1.0] | 36.83s [0.1] |
| S2264 | timeout | 46.07s [0.1] | 1m 24s [0.1] | timeout | 2m 18s [0.1] | 2m 26s [0.1] | timeout | 39.23s [0.1] | 1m 9s [0.1] |
| RS212 | 0.00s | 0.00s [1.0] | 0.00s [1.0] | 0.00s | 0.00s [1.0] | 0.00s [1.0] | 0.00s | 0.00s [0.01] | 0.00s [0.1] |
| RS222 | 0.00s | 0.00s [1.0] | 0.01s [0.1] | 0.00s | 0.00s [1.0] | 0.01s [1.0] | 0.00s | 0.00s [0.01] | 0.00s [0.1] |
| RS213 | timeout | 28.80s [1.0] | 1m 50s [1.0] | timeout | 31.02s [1.0] | 1m 11s [1.0] | timeout | 15.54s [0.1] | 37.73s [0.1] |
| RS223 | timeout | timeout [—] | timeout [—] | timeout | timeout [—] | timeout [—] | timeout | timeout [—] | timeout [—] |

### G.3.3 Results for EFCE solution concept

| Game | Maximize Player 1's utility | | | Maximize Player 2's utility | | | Maximize social welfare | | |
|---|---|---|---|---|---|---|---|---|---|
| | LP | Ours (DCFR) | Ours (PCFR$^+$) | LP | Ours (DCFR) | Ours (PCFR$^+$) | LP | Ours (DCFR) | Ours (PCFR$^+$) |
| B2222 | 0.00s | 0.01s [1.0] | 0.03s [1.0] | 0.00s | 0.06s [1.0] | 0.05s [0.1] | 0.00s | 0.03s [10.0] | 0.02s [1.0] |
| B2322 | 9.00s | 1.23s [1.0] | 2.97s [1.0] | 9.00s | 4.63s [1.0] | 4.68s [1.0] | 9.00s | 1.60s [10.0] | 2.88s [1.0] |
| B2323 | 3m 54s | 48.40s [1.0] | 1m 28s [1.0] | 4m 9s | 1m 38s [1.0] | 1m 27s [1.0] | 3m 40s | 45.12s [10.0] | 44.87s [1.0] |
| B2324 | timeout | 9m 3s [1.0] | 13m 8s [1.0] | timeout | timeout [—] | 10m 21s [1.0] | timeout | 14m 30s [1.0] | 10m 48s [1.0] |
| S2122 | 0.00s | 0.01s [0.1] | 0.02s [1.0] | 0.00s | 0.02s [0.1] | 0.04s [0.1] | 0.00s | 0.02s [0.1] | 0.02s [1.0] |
| S2123 | 1.00s | 0.09s [1.0] | 0.23s [0.1] | 1.00s | 0.34s [0.1] | 0.43s [1.0] | 1.00s | 0.15s [0.1] | 0.25s [0.1] |
| S2133 | 4.00s | 0.52s [1.0] | 0.77s [0.1] | 3.00s | 1.86s [0.1] | 1.31s [0.1] | 3.00s | 0.49s [1.0] | 0.96s [1.0] |
| S2254 | timeout | 2m 17s [0.1] | 2m 10s [0.1] | timeout | timeout [—] | timeout [—] | timeout | 3m 34s [0.1] | timeout [—] |
| S2264 | timeout | timeout [—] | timeout [—] | timeout | timeout [—] | timeout [—] | timeout | timeout [—] | timeout [—] |
| RS212 | 0.00s | 0.00s [1.0] | 0.00s [1.0] | 0.00s | 0.00s [1.0] | 0.00s [1.0] | 0.00s | 0.00s [0.01] | 0.00s [0.01] |
| RS222 | 0.00s | 0.00s [1.0] | 0.01s [0.1] | 0.00s | 0.00s [1.0] | 0.01s [1.0] | 0.00s | 0.00s [1.0] | 0.00s [0.1] |
| RS213 | timeout | 35.37s [1.0] | 1m 28s [0.1] | timeout | 32.49s [1.0] | 1m 27s [1.0] | timeout | 23.37s [0.01] | 57.68s [0.01] |
| RS223 | timeout | timeout [—] | timeout [—] | timeout | timeout [—] | timeout [—] | timeout | timeout [—] | timeout [—] |

### G.3.4 Results for COMM solution concept

| Game | Maximize Player 1's utility | | | Maximize Player 2's utility | | | Maximize social welfare | | |
|---|---|---|---|---|---|---|---|---|---|
| | LP | Ours (DCFR) | Ours (PCFR$^+$) | LP | Ours (DCFR) | Ours (PCFR$^+$) | LP | Ours (DCFR) | Ours (PCFR$^+$) |
| B2222 | 2.00s | 0.88s [1.0] | 1.14s [1.0] | 2.00s | 1.23s [10.0] | 0.89s [1.0] | 2.00s | 1.49s [10.0] | 2.33s [1.0] |
| B2322 | timeout | 5m 47s [1.0] | 10m 17s [1.0] | timeout | 3m 45s [1.0] | 5m 2s [1.0] | timeout | 4m 41s [1.0] | 7m 6s [1.0] |
| B2323 | timeout | timeout [—] | timeout [—] | timeout | timeout [—] | timeout [—] | timeout | timeout [—] | timeout [—] |
| B2324 | timeout | timeout [—] | timeout [—] | timeout | timeout [—] | timeout [—] | timeout | timeout [—] | timeout [—] |
| S2122 | 2.00s | 0.21s [0.1] | 0.36s [0.1] | 2.00s | 0.48s [0.01] | 0.48s [0.1] | 2.00s | 0.36s [0.01] | 0.35s [0.1] |
| S2123 | 1m 30s | 38.95s [0.1] | 1m 7s [0.1] | 1m 36s | 1m 10s [0.01] | 1m 52s [0.01] | 1m 33s | 59.63s [0.01] | 1m 30s [0.1] |
| S2133 | timeout | 7m 34s [0.01] | 4m 26s [0.1] | timeout | 7m 27s [0.01] | 14m 12s [0.01] | timeout | 12m 11s [0.01] | 13m 40s [0.01] |
| S2254 | timeout | timeout [—] | timeout [—] | timeout | timeout [—] | timeout [—] | timeout | timeout [—] | timeout [—] |
| S2264 | timeout | timeout [—] | timeout [—] | timeout | timeout [—] | timeout [—] | timeout | timeout [—] | timeout [—] |
| RS212 | 2.00s | 0.01s [1.0] | 0.01s [1.0] | 2.00s | 0.01s [1.0] | 0.03s [10.0] | 2.00s | 0.01s [1.0] | 0.01s [1.0] |
| RS222 | 3.00s | 0.01s [1.0] | 0.01s [0.1] | 3.00s | 0.02s [10.0] | 0.04s [1.0] | 3.00s | 0.02s [10.0] | 0.01s [0.01] |
| RS213 | 6m 51s | 11.05s [1.0] | 10.24s [1.0] | 6m 27s | 14.66s [10.0] | 12.83s [1.0] | 6m 25s | 9.00s [1.0] | 8.74s [1.0] |
| RS223 | 8m 41s | 5.79s [10.0] | 10.49s [1.0] | 8m 24s | 8.45s [1.0] | 11.14s [1.0] | 8m 54s | 4.00s [1.0] | 7.02s [1.0] |

### G.3.5 Results for NFCCERT solution concept

| Game | Maximize Player 1's utility | | | Maximize Player 2's utility | | | Maximize social welfare | | |
|---|---|---|---|---|---|---|---|---|---|
| | LP | Ours (DCFR) | Ours (PCFR$^+$) | LP | Ours (DCFR) | Ours (PCFR$^+$) | LP | Ours (DCFR) | Ours (PCFR$^+$) |
| B2222 | 0.00s | 0.00s [1.0] | 0.00s [0.1] | 0.00s | 0.00s [1.0] | 0.00s [0.1] | 0.00s | 0.00s [0.01] | 0.00s [10.0] |
| B2322 | 0.00s | 0.02s [1.0] | 0.02s [0.1] | 0.00s | 0.02s [0.1] | 0.02s [0.1] | 0.00s | 0.01s [0.01] | 0.01s [0.1] |
| B2323 | 2.00s | 0.62s [0.1] | 0.63s [0.1] | 1.00s | 0.48s [0.1] | 0.73s [0.1] | 2.00s | 0.14s [1.0] | 0.18s [0.1] |
| B2324 | 11.00s | 4.88s [0.1] | 11.06s [0.1] | 11.00s | 5.24s [0.1] | 9.92s [0.1] | 10.00s | 1.82s [0.01] | 2.51s [1.0] |
| S2122 | 0.00s | 0.00s [0.01] | 0.00s [0.01] | 0.00s | 0.00s [0.01] | 0.00s [0.1] | 0.00s | 0.00s [1.0] | 0.00s [0.01] |
| S2123 | 0.00s | 0.00s [0.01] | 0.00s [0.01] | 0.00s | 0.01s [0.01] | 0.00s [0.1] | 0.00s | 0.00s [0.01] | 0.00s [0.1] |
| S2133 | 0.00s | 0.01s [0.01] | 0.01s [0.1] | 0.00s | 0.01s [1.0] | 0.01s [0.1] | 0.00s | 0.01s [0.1] | 0.01s [0.1] |
| S2254 | 25.00s | 1.23s [0.01] | 2.49s [0.01] | 24.00s | 3.02s [0.1] | 3.02s [0.1] | 28.00s | 1.32s [0.1] | 2.00s [0.01] |
| S2264 | 56.00s | 2.71s [0.1] | 4.93s [0.01] | 42.00s | 5.43s [0.1] | 6.88s [0.01] | 50.00s | 2.73s [0.1] | 3.65s [0.01] |
| RS212 | 0.00s | 0.00s [0.1] | 0.00s [10.0] | 0.00s | 0.00s [10.0] | 0.00s [0.1] | 0.00s | 0.00s [0.1] | 0.00s [0.01] |
| RS222 | 0.00s | 0.00s [10.0] | 0.00s [1.0] | 0.00s | 0.00s [10.0] | 0.00s [1.0] | 0.00s | 0.00s [1.0] | 0.00s [0.1] |
| RS213 | 0.00s | 0.00s [0.1] | 0.00s [0.01] | 0.00s | 0.00s [10.0] | 0.00s [0.1] | 0.00s | 0.00s [1.0] | 0.00s [0.01] |
| RS223 | 0.00s | 0.00s [1.0] | 0.00s [0.1] | 0.00s | 0.00s [10.0] | 0.01s [1.0] | 0.00s | 0.00s [0.01] | 0.01s [1.0] |

### G.3.6 Results for CCERT solution concept

| Game | Maximize Player 1's utility | | | Maximize Player 2's utility | | | Maximize social welfare | | |
|---|---|---|---|---|---|---|---|---|---|
| | LP | Ours (DCFR) | Ours (PCFR$^+$) | LP | Ours (DCFR) | Ours (PCFR$^+$) | LP | Ours (DCFR) | Ours (PCFR$^+$) |
| B2222 | 0.00s | 0.00s [1.0] | 0.01s [1.0] | 0.00s | 0.01s [1.0] | 0.01s [1.0] | 0.00s | 0.01s [1.0] | 0.01s [1.0] |
| B2322 | 0.00s | 0.07s [1.0] | 0.22s [1.0] | 0.00s | 0.15s [1.0] | 0.34s [1.0] | 0.00s | 0.22s [1.0] | 0.33s [1.0] |
| B2323 | 7.00s | 3.75s [1.0] | 7.37s [1.0] | 6.00s | 4.69s [1.0] | 9.79s [1.0] | 8.00s | 5.39s [1.0] | 6.08s [1.0] |
| B2324 | timeout | 54.92s [1.0] | 1m 2s [1.0] | timeout | 59.02s [1.0] | 1m 28s [1.0] | timeout | 1m 31s [1.0] | 1m 41s [1.0] |
| S2122 | 0.00s | 0.00s [1.0] | 0.01s [0.1] | 0.00s | 0.00s [0.1] | 0.00s [0.1] | 0.00s | 0.00s [1.0] | 0.01s [0.1] |
| S2123 | 0.00s | 0.01s [1.0] | 0.05s [1.0] | 0.00s | 0.02s [1.0] | 0.03s [0.1] | 0.00s | 0.01s [1.0] | 0.05s [1.0] |
| S2133 | 1.00s | 0.04s [1.0] | 0.07s [1.0] | 1.00s | 0.08s [0.1] | 0.09s [0.1] | 1.00s | 0.05s [1.0] | 0.11s [1.0] |
| S2254 | 1m 41s | 8.61s [0.1] | 15.31s [0.1] | 1m 47s | 25.38s [0.1] | 23.28s [0.1] | 2m 3s | 8.22s [0.1] | 16.08s [0.1] |
| S2264 | timeout | 20.11s [1.0] | 23.99s [0.1] | timeout | 1m 9s [0.01] | 1m 5s [0.1] | timeout | 16.50s [0.1] | 29.02s [0.1] |
| RS212 | 0.00s | 0.00s [10.0] | 0.00s [1.0] | 0.00s | 0.00s [10.0] | 0.00s [1.0] | 0.00s | 0.00s [1.0] | 0.00s [1.0] |
| RS222 | 0.00s | 0.00s [0.01] | 0.00s [10.0] | 0.00s | 0.00s [1.0] | 0.00s [1.0] | 0.00s | 0.00s [0.01] | 0.00s [0.01] |
| RS213 | 0.00s | 0.01s [10.0] | 0.01s [1.0] | 0.00s | 0.01s [1.0] | 0.01s [1.0] | 0.00s | 0.01s [0.1] | 0.01s [0.01] |
| RS223 | 0.00s | 0.01s [10.0] | 0.01s [1.0] | 0.00s | 0.01s [1.0] | 0.01s [1.0] | 0.00s | 0.01s [10.0] | 0.01s [0.1] |

### G.3.7 Results for CERT solution concept

| Game | Maximize Player 1's utility | | | | | Maximize Player 2's utility | | | | | Maximize social welfare | | | | |
|---|---|---|---|---|---|---|---|---|---|---|---|---|---|---|---|
| | LP | Ours (DCFR) | | Ours (PCFR$^+$) | | LP | Ours (DCFR) | | Ours (PCFR$^+$) | | LP | Ours (DCFR) | | Ours (PCFR$^+$) | |
| B2222 | 0.00s | 0.01s | [1.0] | 0.01s | [1.0] | 0.00s | 0.02s | [1.0] | 0.02s | [1.0] | 0.00s | 0.02s | [1.0] | 0.03s | [1.0] |
| B2322 | 2.00s | 1.05s | [1.0] | 1.16s | [1.0] | 2.00s | 1.16s | [1.0] | 2.29s | [1.0] | 2.00s | 1.24s | [1.0] | 1.43s | [1.0] |
| B2323 | 40.00s | 47.11s | [10.0] | 1m 2s | [1.0] | 33.00s | 58.20s | [1.0] | 2m 14s | [0.1] | 37.00s | 46.51s | [1.0] | 40.45s | [1.0] |
| B2324 | timeout | 8m 29s | [0.1] | timeout | [—] | timeout | timeout | [—] | timeout | [—] | timeout | 6m 14s | [1.0] | timeout | [—] |
| S2122 | 0.00s | 0.02s | [0.1] | 0.02s | [1.0] | 0.00s | 0.02s | [0.1] | 0.02s | [0.1] | 0.00s | 0.02s | [1.0] | 0.02s | [1.0] |
| S2123 | 1.00s | 0.19s | [0.1] | 0.37s | [0.1] | 1.00s | 0.28s | [1.0] | 0.31s | [0.1] | 1.00s | 0.15s | [1.0] | 0.35s | [0.1] |
| S2133 | 3.00s | 0.96s | [1.0] | 1.06s | [1.0] | 2.00s | 1.10s | [0.1] | 1.18s | [0.1] | 2.00s | 0.92s | [0.1] | 1.26s | [0.1] |
| S2254 | timeout | 3m 26s | [0.1] | 5m 35s | [0.1] | timeout | 6m 23s | [0.1] | 6m 15s | [0.1] | timeout | 2m 42s | [0.1] | 8m 2s | [0.1] |
| S2264 | timeout | timeout | [—] | timeout | [—] | timeout | timeout | [—] | timeout | [—] | timeout | timeout | [—] | timeout | [—] |
| RS212 | 0.00s | 0.00s | [0.1] | 0.00s | [1.0] | 0.00s | 0.00s | [10.0] | 0.00s | [1.0] | 0.00s | 0.00s | [0.1] | 0.00s | [0.1] |
| RS222 | 0.00s | 0.00s | [1.0] | 0.00s | [10.0] | 0.00s | 0.00s | [1.0] | 0.00s | [1.0] | 0.00s | 0.00s | [0.01] | 0.00s | [0.01] |
| RS213 | 0.00s | 0.02s | [10.0] | 0.03s | [1.0] | 0.00s | 0.02s | [1.0] | 0.03s | [1.0] | 0.00s | 0.02s | [10.0] | 0.02s | [0.1] |
| RS223 | 1.00s | 0.01s | [10.0] | 0.02s | [1.0] | 1.00s | 0.02s | [1.0] | 0.03s | [1.0] | 1.00s | 0.01s | [0.1] | 0.02s | [0.01] |

## G.4 Detailed breakdown by equilibrium and objective function (three-player games)

For each two-player game, we try three different objective functions: maximizing the utility of Player 1, maximizing the utility of Player 2, and maximizing the utility of Player 3. As in Appendix G.3, for each objective we stop the optimization at the approximation level defined as 1% of the payoff range of the game.

For each game and objective, we report the same information as Appendix G.3.

### G.4.1 Results for NFCCE solution concept

| Game | Maximize Player 1's utility | | | | | Maximize Player 2's utility | | | | | Maximize Player 3's utility | | | | |
|---|---|---|---|---|---|---|---|---|---|---|---|---|---|---|---|
| | LP | Ours (DCFR) | | Ours (PCFR$^+$) | | LP | Ours (DCFR) | | Ours (PCFR$^+$) | | LP | Ours (DCFR) | | Ours (PCFR$^+$) | |
| D32 | 0.00s | 0.01s | [1.0] | 0.01s | [1.0] | 0.00s | 0.01s | [1.0] | 0.01s | [1.0] | 0.00s | 0.01s | [1.0] | 0.02s | [1.0] |
| D33 | 2m 17s | 12.93s | [0.1] | 43.45s | [0.1] | 2m 8s | 16.40s | [0.1] | 33.57s | [0.1] | 2m 23s | 17.57s | [0.1] | 36.55s | [1.0] |
| GL3 | 0.00s | 0.01s | [1.0] | 0.02s | [1.0] | 0.00s | 0.01s | [1.0] | 0.02s | [1.0] | 0.00s | 0.02s | [0.1] | 0.02s | [0.1] |
| K35 | 49.00s | 0.76s | [1.0] | 0.95s | [0.1] | 55.00s | 0.56s | [1.0] | 1.15s | [0.1] | 55.00s | 0.64s | [1.0] | 1.34s | [0.1] |
| L3132 | 26.00s | 0.59s | [0.1] | 1.09s | [0.1] | 24.00s | 0.79s | [0.1] | 1.28s | [0.1] | 24.00s | 0.77s | [0.1] | 1.18s | [0.1] |
| L3133 | 38.00s | 0.94s | [0.1] | 1.93s | [0.1] | 38.00s | 0.89s | [0.1] | 2.33s | [0.1] | 37.00s | 1.11s | [0.1] | 1.79s | [0.1] |
| L3151 | timeout | 15.12s | [0.1] | 17.94s | [0.1] | timeout | 12.74s | [0.1] | 31.74s | [0.1] | timeout | 18.03s | [0.1] | 31.69s | [0.1] |
| L3223 | 4.00s | 0.44s | [0.1] | 0.92s | [0.1] | 4.00s | 0.45s | [0.1] | 0.98s | [0.1] | 4.00s | 0.52s | [0.1] | 1.26s | [0.1] |
| L3523 | timeout | 1m 7s | [0.01] | 1m 59s | [0.01] | timeout | 1m 2s | [0.01] | 1m 44s | [0.01] | timeout | 1m 9s | [0.01] | 1m 44s | [0.01] |
| TP3 | 1m 38s | 7.44s | [1.0] | 10.71s | [10.0] | 1m 40s | 7.63s | [1.0] | 11.09s | [10.0] | 1m 44s | 11.45s | [10.0] | 11.90s | [10.0] |

### G.4.2 Results for EFCCE solution concept

| Game | Maximize Player 1's utility | | | | | Maximize Player 2's utility | | | | | Maximize Player 3's utility | | | | |
|---|---|---|---|---|---|---|---|---|---|---|---|---|---|---|---|
| | LP | Ours (DCFR) | | Ours (PCFR$^+$) | | LP | Ours (DCFR) | | Ours (PCFR$^+$) | | LP | Ours (DCFR) | | Ours (PCFR$^+$) | |
| D32 | 0.00s | 0.02s | [1.0] | 0.02s | [1.0] | 0.00s | 0.01s | [1.0] | 0.02s | [1.0] | 0.00s | 0.02s | [1.0] | 0.03s | [1.0] |
| D33 | timeout | 1m 46s | [1.0] | 4m 31s | [0.1] | timeout | 1m 16s | [1.0] | 3m 56s | [1.0] | timeout | 1m 44s | [1.0] | 4m 56s | [1.0] |
| GL3 | 1.00s | 0.02s | [1.0] | 0.04s | [1.0] | 1.00s | 0.03s | [1.0] | 0.04s | [1.0] | 1.00s | 0.03s | [1.0] | 0.04s | [10.0] |
| K35 | 46.00s | 0.67s | [10.0] | 1.69s | [0.1] | 55.00s | 0.75s | [1.0] | 1.69s | [0.1] | 51.00s | 0.68s | [1.0] | 2.02s | [0.1] |
| L3132 | 8m 43s | 5.13s | [0.1] | 9.57s | [1.0] | 9m 17s | 6.27s | [0.1] | 12.37s | [0.1] | 9m 44s | 7.76s | [0.1] | 14.66s | [0.1] |
| L3133 | 20m 26s | 8.88s | [0.1] | 18.30s | [0.1] | timeout | 8.19s | [1.0] | 18.09s | [0.1] | 23m 15s | 10.86s | [1.0] | 18.52s | [0.1] |
| L3151 | timeout | timeout | [—] | timeout | [—] | timeout | timeout | [—] | timeout | [—] | timeout | timeout | [—] | timeout | [—] |
| L3223 | 1m 10s | 2.94s | [0.1] | 4.85s | [0.1] | 1m 10s | 3.22s | [0.1] | 4.73s | [0.1] | 1m 2s | 3.24s | [0.1] | 4.78s | [0.1] |
| L3523 | timeout | timeout | [—] | timeout | [—] | timeout | timeout | [—] | timeout | [—] | timeout | timeout | [—] | timeout | [—] |
| TP3 | timeout | 13.76s | [10.0] | 24.94s | [1.0] | timeout | 15.03s | [10.0] | 28.28s | [1.0] | timeout | 15.27s | [10.0] | 27.29s | [1.0] |

### G.4.3 Results for EFCE solution concept

| Game | Maximize Player 1's utility | | | Maximize Player 2's utility | | | Maximize Player 3's utility | | |
|---|---|---|---|---|---|---|---|---|---|
| | LP | Ours (DCFR) | Ours (PCFR$^+$) | LP | Ours (DCFR) | Ours (PCFR$^+$) | LP | Ours (DCFR) | Ours (PCFR$^+$) |
| D32 | 12.00s | 0.76s [1.0] | 0.40s [1.0] | 11.00s | 0.35s [1.0] | 0.85s [1.0] | 10.00s | 0.66s [1.0] | 0.80s [1.0] |
| D33 | timeout | timeout [—] | timeout [—] | timeout | timeout [—] | timeout [—] | timeout | timeout [—] | timeout [—] |
| GL3 | 0.00s | 0.01s [1.0] | 0.02s [1.0] | 0.00s | 0.01s [1.0] | 0.02s [0.1] | 0.00s | 0.01s [1.0] | 0.02s [1.0] |
| K35 | 57.00s | 0.55s [1.0] | 1.08s [0.1] | 55.00s | 1.03s [1.0] | 1.47s [0.1] | 60.00s | 1.26s [1.0] | 1.77s [0.1] |
| L3132 | 8m 18s | 6.10s [0.1] | 7.85s [0.1] | 8m 57s | 7.65s [0.1] | 12.08s [0.1] | 7m 35s | 6.78s [0.1] | 12.88s [0.1] |
| L3133 | 21m 25s | 6.84s [0.1] | 12.97s [0.1] | 21m 43s | 10.76s [0.1] | 18.44s [0.1] | 19m 58s | 10.28s [0.1] | 15.69s [0.1] |
| L3151 | timeout | timeout [—] | timeout [—] | timeout | timeout [—] | timeout [—] | timeout | timeout [—] | timeout [—] |
| L3223 | 2m 2s | 5.52s [0.1] | 8.74s [0.1] | 1m 50s | 6.46s [0.1] | 10.70s [0.1] | 2m 0s | 5.94s [0.1] | 10.65s [0.1] |
| L3523 | timeout | timeout [—] | timeout [—] | timeout | timeout [—] | timeout [—] | timeout | timeout [—] | timeout [—] |
| TP3 | timeout | 13.46s [10.0] | 20.25s [1.0] | timeout | 14.25s [1.0] | 22.19s [10.0] | timeout | 14.48s [10.0] | 21.28s [1.0] |

### G.4.4 Results for COMM solution concept

| Game | Maximize Player 1's utility | | | Maximize Player 2's utility | | | Maximize Player 3's utility | | |
|---|---|---|---|---|---|---|---|---|---|
| | LP | Ours (DCFR) | Ours (PCFR$^+$) | LP | Ours (DCFR) | Ours (PCFR$^+$) | LP | Ours (DCFR) | Ours (PCFR$^+$) |
| D32 | 0.00s | 0.06s [1.0] | 0.12s [1.0] | 1.00s | 0.06s [1.0] | 0.18s [0.1] | 1.00s | 0.18s [1.0] | 0.17s [1.0] |
| D33 | timeout | 4m 37s [1.0] | 9m 46s [1.0] | timeout | 1m 31s [1.0] | 3m 5s [1.0] | timeout | 2m 38s [1.0] | 3m 57s [0.1] |
| GL3 | timeout | 7.72s [0.1] | 11.24s [0.1] | timeout | 7.50s [1.0] | 11.02s [1.0] | timeout | 11.42s [1.0] | 18.22s [0.1] |
| K35 | 1.00s | 0.03s [1.0] | 0.03s [0.1] | 1.00s | 0.02s [1.0] | 0.03s [0.1] | 1.00s | 0.03s [1.0] | 0.04s [1.0] |
| L3132 | 8.00s | 3.46s [0.1] | 5.65s [0.1] | 8.00s | 3.37s [0.1] | 5.89s [0.1] | 7.00s | 4.02s [0.1] | 8.38s [0.1] |
| L3133 | 12.00s | 3.40s [0.1] | 7.98s [0.1] | 12.00s | 3.54s [0.1] | 7.89s [0.1] | 11.00s | 3.52s [0.1] | 10.52s [0.1] |
| L3151 | timeout | 16.73s [0.1] | 18.42s [0.1] | timeout | 15.80s [0.1] | 29.51s [0.1] | timeout | 18.00s [0.1] | 22.54s [0.1] |
| L3223 | 19.00s | 18.19s [0.1] | 29.24s [0.01] | 18.00s | 15.30s [0.1] | 27.38s [0.1] | 21.00s | 18.77s [0.1] | 25.91s [0.01] |
| L3523 | timeout | timeout [—] | timeout [—] | timeout | timeout [—] | timeout [—] | timeout | timeout [—] | timeout [—] |
| TP3 | timeout | timeout [—] | timeout [—] | timeout | timeout [—] | timeout [—] | timeout | timeout [—] | timeout [—] |

### G.4.5 Results for NFCCERT solution concept

| Game | Maximize Player 1's utility | | | Maximize Player 2's utility | | | Maximize Player 3's utility | | |
|---|---|---|---|---|---|---|---|---|---|
| | LP | Ours (DCFR) | Ours (PCFR$^+$) | LP | Ours (DCFR) | Ours (PCFR$^+$) | LP | Ours (DCFR) | Ours (PCFR$^+$) |
| D32 | 0.00s | 0.00s [10.0] | 0.00s [1.0] | 0.00s | 0.00s [1.0] | 0.00s [1.0] | 0.00s | 0.00s [1.0] | 0.00s [1.0] |
| D33 | 0.00s | 0.04s [1.0] | 0.05s [1.0] | 0.00s | 0.06s [1.0] | 0.06s [1.0] | 0.00s | 0.04s [10.0] | 0.05s [10.0] |
| GL3 | 0.00s | 0.00s [1.0] | 0.00s [1.0] | 0.00s | 0.00s [1.0] | 0.00s [1.0] | 0.00s | 0.01s [10.0] | 0.00s [1.0] |
| K35 | 0.00s | 0.00s [1.0] | 0.00s [1.0] | 0.00s | 0.00s [1.0] | 0.00s [1.0] | 0.00s | 0.00s [1.0] | 0.00s [1.0] |
| L3132 | 0.00s | 0.02s [0.1] | 0.02s [1.0] | 0.00s | 0.02s [0.1] | 0.02s [0.1] | 0.00s | 0.02s [0.1] | 0.02s [0.1] |
| L3133 | 0.00s | 0.02s [0.1] | 0.02s [0.1] | 0.00s | 0.02s [1.0] | 0.02s [0.1] | 0.00s | 0.02s [0.1] | 0.03s [1.0] |
| L3151 | 0.00s | 0.03s [0.1] | 0.08s [0.1] | 0.00s | 0.04s [0.1] | 0.04s [0.1] | 0.00s | 0.03s [0.1] | 0.04s [0.1] |
| L3223 | 0.00s | 0.03s [0.1] | 0.04s [0.1] | 0.00s | 0.05s [0.1] | 0.04s [0.1] | 0.00s | 0.05s [0.1] | 0.05s [0.1] |
| L3523 | 13.00s | 7.41s [0.1] | 14.31s [0.1] | 15.00s | 8.17s [0.1] | 9.36s [0.1] | 17.00s | 10.60s [0.01] | 13.52s [0.01] |
| TP3 | 47.00s | 4.56s [1.0] | 10.06s [10.0] | 48.00s | 4.97s [10.0] | 8.55s [1.0] | 45.00s | 7.38s [10.0] | 9.40s [1.0] |

### G.4.6 Results for CCERT solution concept

| Game | Maximize Player 1's utility | | | Maximize Player 2's utility | | | Maximize Player 3's utility | | |
|------|---|---|---|---|---|---|---|---|---|
| | LP | Ours (DCFR) | Ours (PCFR$^+$) | LP | Ours (DCFR) | Ours (PCFR$^+$) | LP | Ours (DCFR) | Ours (PCFR$^+$) |
| D32 | 0.00s | 0.00s [1.0] | 0.00s [1.0] | 0.00s | 0.00s [1.0] | 0.00s [1.0] | 0.00s | 0.00s [1.0] | 0.00s [1.0] |
| D33 | 1.00s | 0.20s [1.0] | 0.29s [1.0] | 1.00s | 0.13s [1.0] | 0.30s [1.0] | 1.00s | 0.31s [0.1] | 0.19s [0.1] |
| GL3 | 0.00s | 0.01s [0.1] | 0.01s [1.0] | 0.00s | 0.02s [1.0] | 0.01s [1.0] | 0.00s | 0.01s [0.1] | 0.01s [1.0] |
| K35 | 0.00s | 0.00s [1.0] | 0.00s [1.0] | 0.00s | 0.00s [0.1] | 0.00s [1.0] | 0.00s | 0.00s [1.0] | 0.00s [1.0] |
| L3132 | 0.00s | 0.06s [0.1] | 0.05s [0.1] | 0.00s | 0.05s [0.1] | 0.07s [0.1] | 0.00s | 0.05s [0.1] | 0.06s [0.1] |
| L3133 | 0.00s | 0.04s [0.1] | 0.05s [0.1] | 0.00s | 0.08s [0.1] | 0.11s [0.1] | 0.00s | 0.12s [0.1] | 0.15s [0.1] |
| L3151 | 1.00s | 0.12s [0.1] | 0.27s [0.1] | 1.00s | 0.16s [0.1] | 0.35s [0.1] | 1.00s | 0.16s [0.1] | 0.33s [0.1] |
| L3223 | 1.00s | 0.23s [0.1] | 0.52s [0.1] | 1.00s | 0.26s [0.1] | 0.43s [0.1] | 1.00s | 0.24s [0.1] | 0.41s [0.01] |
| L3523 | timeout | 1m 17s [0.01] | 2m 47s [0.01] | timeout | 1m 16s [0.01] | 3m 6s [0.01] | timeout | 1m 13s [0.01] | 2m 26s [0.01] |
| TP3 | timeout | 10.23s [10.0] | 18.00s [1.0] | timeout | 10.38s [10.0] | 19.88s [10.0] | timeout | 11.95s [10.0] | 25.89s [10.0] |

### G.4.7 Results for CERT solution concept

| Game | Maximize Player 1's utility | | | Maximize Player 2's utility | | | Maximize Player 3's utility | | |
|------|---|---|---|---|---|---|---|---|---|
| | LP | Ours (DCFR) | Ours (PCFR$^+$) | LP | Ours (DCFR) | Ours (PCFR$^+$) | LP | Ours (DCFR) | Ours (PCFR$^+$) |
| D32 | 0.00s | 0.01s [1.0] | 0.01s [1.0] | 0.00s | 0.01s [1.0] | 0.01s [1.0] | 0.00s | 0.01s [1.0] | 0.02s [1.0] |
| D33 | 4.00s | 3.14s [1.0] | 6.08s [1.0] | 4.00s | 2.02s [1.0] | 3.18s [0.1] | 4.00s | 2.10s [1.0] | 3.89s [0.1] |
| GL3 | 0.00s | 0.02s [1.0] | 0.03s [0.1] | 0.00s | 0.02s [1.0] | 0.03s [1.0] | 0.00s | 0.03s [0.1] | 0.03s [0.1] |
| K35 | 0.00s | 0.01s [1.0] | 0.01s [1.0] | 0.00s | 0.00s [1.0] | 0.01s [0.1] | 0.00s | 0.00s [1.0] | 0.01s [0.1] |
| L3132 | 1.00s | 0.15s [0.1] | 0.10s [0.1] | 1.00s | 0.18s [0.1] | 0.22s [0.1] | 0.00s | 0.18s [0.1] | 0.35s [0.1] |
| L3133 | 1.00s | 0.22s [0.1] | 0.42s [1.0] | 1.00s | 0.19s [0.1] | 0.33s [0.1] | 1.00s | 0.25s [0.1] | 0.38s [0.1] |
| L3151 | 2.00s | 0.21s [0.1] | 0.42s [0.1] | 2.00s | 0.22s [0.1] | 0.44s [0.1] | 2.00s | 0.29s [0.1] | 0.53s [0.1] |
| L3223 | 1.00s | 0.61s [0.1] | 1.43s [0.01] | 1.00s | 0.61s [0.1] | 1.13s [0.01] | 1.00s | 0.68s [0.1] | 0.98s [0.01] |
| L3523 | timeout | 2m 58s [0.01] | timeout [—] | timeout | 4m 33s [0.01] | timeout [—] | timeout | 3m 55s [0.01] | timeout [—] |
| TP3 | timeout | 26.70s [1.0] | 40.73s [1.0] | timeout | 25.08s [1.0] | 35.99s [10.0] | timeout | 36.36s [10.0] | 1m 0s [1.0] |

# H   Experimental results for binary search-based Lagrangian

In this section, we compare the performance of our "direct" Lagrangian approach against our binary search-based Lagrangian approach for computing several equilibrium concepts at the approximation level defined as 1% of the payoff range of the game.

For each solution concept, we identify the same three objectives as Appendix G (maximizing each player's individual utility, and maximizing the social welfare in our two-player general-sum games). For each objective, each of the following tables compares three runtimes:

- The time required by the linear program (column 'LP');

- The time required by the "direct" (non-binary search-based) Lagrangian approach, taking the fastest between the implementations using DCFR and PCFR$^+$ as the underlying no-regret algorithms (column 'Lagrangian').

- The time required by the binary search-based Lagrangian approach, taking the fastest between the implementations using DCFR and PCFR$^+$ as the underlying no-regret algorithms (column 'Bin.Search').

We observe that our two approaches behave similarly in small games. In larger games, especially with three players, the direct Lagrangian tends to be 2-4 times faster.

## H.1 Detailed Breakdown by Equilibrium and Objective Function (Two-Player Games)

### H.1.1 Results for NFCCE solution concept

| Game | Maximize Pl. 1's utility | | | Maximize Pl. 2's utility | | | Maximize social welfare | | |
|------|-----|------------|------------|-----|------------|------------|-----|------------|------------|
| | LP | Lagrangian | Bin.Search | LP | Lagrangian | Bin.Search | LP | Lagrangian | Bin.Search |
| B2222 | 0.00s | 0.00s | 0.00s | 0.00s | 0.00s | 0.00s | 0.00s | 0.00s | 0.00s |
| B2322 | 0.00s | 0.03s | 0.04s | 0.00s | 0.04s | 0.03s | 0.00s | 0.01s | 0.01s |
| B2323 | 7.00s | 1.05s | 1.67s | 6.00s | 1.01s | 1.23s | 6.00s | 0.33s | 0.17s |
| B2324 | 50.00s | 15.57s | 18.23s | 37.00s | 12.80s | 11.11s | 38.00s | 2.73s | 1.41s |
| S2122 | 0.00s | 0.00s | 0.00s | 0.00s | 0.00s | 0.00s | 0.00s | 0.00s | 0.00s |
| S2123 | 0.00s | 0.01s | 0.01s | 0.00s | 0.01s | 0.01s | 0.00s | 0.01s | 0.01s |
| S2133 | 1.00s | 0.02s | 0.03s | 1.00s | 0.02s | 0.02s | 1.00s | 0.02s | 0.02s |
| S2254 | 2m 1s | 6.96s | 6.03s | 1m 14s | 10.43s | 7.12s | 1m 58s | 7.43s | 3.64s |
| S2264 | 3m 36s | 13.96s | 7.64s | 2m 24s | 18.46s | 14.57s | 3m 43s | 11.74s | 7.69s |
| RS212 | 0.00s | 0.00s | 0.00s | 0.00s | 0.00s | 0.00s | 0.00s | 0.00s | 0.00s |
| RS213 | timeout | 34.52s | 28.65s | timeout | 20.29s | 26.25s | timeout | 14.68s | 7.44s |
| RS222 | 0.00s | 0.00s | 0.00s | 0.00s | 0.00s | 0.00s | 0.00s | 0.00s | 0.00s |
| RS223 | timeout | timeout | timeout | timeout | timeout | timeout | timeout | timeout | timeout |

### H.1.2 Results for EFCCE solution concept

| Game | Maximize Pl. 1's utility | | | Maximize Pl. 2's utility | | | Maximize social welfare | | |
|------|-----|------------|------------|-----|------------|------------|-----|------------|------------|
| | LP | Lagrangian | Bin.Search | LP | Lagrangian | Bin.Search | LP | Lagrangian | Bin.Search |
| B2222 | 0.00s | 0.01s | 0.01s | 0.00s | 0.01s | 0.01s | 0.00s | 0.01s | 0.01s |
| B2322 | 3.00s | 0.41s | 0.54s | 3.00s | 0.69s | 0.60s | 3.00s | 0.69s | 0.41s |
| B2323 | 1m 35s | 9.11s | 16.32s | 1m 30s | 12.39s | 8.62s | 1m 21s | 14.23s | 6.89s |
| B2324 | timeout | 1m 53s | 2m 5s | timeout | 2m 44s | 3m 14s | timeout | 3m 1s | 1m 27s |
| S2122 | 0.00s | 0.01s | 0.00s | 0.00s | 0.00s | 0.00s | 0.00s | 0.01s | 0.00s |
| S2123 | 1.00s | 0.03s | 0.04s | 1.00s | 0.05s | 0.04s | 1.00s | 0.06s | 0.03s |
| S2133 | 3.00s | 0.12s | 0.11s | 2.00s | 0.20s | 0.32s | 3.00s | 0.11s | 0.22s |
| S2254 | timeout | 27.31s | 30.66s | timeout | 37.43s | 1m 17s | timeout | 22.01s | 36.62s |
| S2264 | timeout | 46.07s | 1m 9s | timeout | 2m 18s | 1m 26s | timeout | 39.23s | 1m 4s |
| RS212 | 0.00s | 0.00s | 0.00s | 0.00s | 0.00s | 0.00s | 0.00s | 0.00s | 0.00s |
| RS213 | timeout | 28.80s | 36.02s | timeout | 31.02s | 29.54s | timeout | 15.54s | 9.84s |
| RS222 | 0.00s | 0.00s | 0.00s | 0.00s | 0.00s | 0.00s | 0.00s | 0.00s | 0.00s |
| RS223 | timeout | timeout | timeout | timeout | timeout | timeout | timeout | timeout | timeout |

### H.1.3 Results for EFCE solution concept

| Game | Maximize Pl. 1's utility | | | Maximize Pl. 2's utility | | | Maximize social welfare | | |
|------|-----|------------|------------|-----|------------|------------|-----|------------|------------|
| | LP | Lagrangian | Bin.Search | LP | Lagrangian | Bin.Search | LP | Lagrangian | Bin.Search |
| B2222 | 0.00s | 0.01s | 0.02s | 0.00s | 0.05s | 0.01s | 0.00s | 0.02s | 0.02s |
| B2322 | 9.00s | 1.23s | 4.49s | 9.00s | 4.63s | 2.04s | 9.00s | 1.60s | 1.57s |
| B2323 | 3m 54s | 48.40s | 1m 44s | 4m 9s | 1m 27s | 40.91s | 3m 40s | 44.87s | 1m 19s |
| B2324 | timeout | 9m 3s | 16m 45s | timeout | 10m 21s | 13m 15s | timeout | 10m 48s | 8m 12s |
| S2122 | 0.00s | 0.01s | 0.01s | 0.00s | 0.02s | 0.01s | 0.00s | 0.02s | 0.01s |
| S2123 | 1.00s | 0.09s | 0.06s | 1.00s | 0.34s | 0.10s | 1.00s | 0.15s | 0.08s |
| S2133 | 4.00s | 0.52s | 0.32s | 3.00s | 1.31s | 0.77s | 3.00s | 0.49s | 0.43s |
| S2254 | timeout | 2m 10s | 2m 19s | timeout | timeout | 4m 1s | timeout | 3m 34s | 2m 10s |
| S2264 | timeout | timeout | timeout | timeout | timeout | timeout | timeout | timeout | timeout |
| RS212 | 0.00s | 0.00s | 0.00s | 0.00s | 0.00s | 0.00s | 0.00s | 0.00s | 0.00s |
| RS213 | timeout | 35.37s | 34.00s | timeout | 32.49s | 36.52s | timeout | 23.37s | 22.16s |
| RS222 | 0.00s | 0.00s | 0.00s | 0.00s | 0.00s | 0.00s | 0.00s | 0.00s | 0.00s |
| RS223 | timeout | timeout | timeout | timeout | timeout | timeout | timeout | timeout | timeout |

## H.1.4 Results for COMM solution concept

| Game | Maximize Pl. 1's utility | | | Maximize Pl. 2's utility | | | Maximize social welfare | | |
|---|---|---|---|---|---|---|---|---|---|
| | LP | Lagrangian | Bin.Search | LP | Lagrangian | Bin.Search | LP | Lagrangian | Bin.Search |
| B2222 | 2.00s | 0.88s | 3.39s | 2.00s | 0.89s | 2.70s | 2.00s | 1.49s | 2.80s |
| B2322 | timeout | 5m 47s | 7m 28s | timeout | 3m 45s | 10m 31s | timeout | 4m 41s | 7m 20s |
| B2323 | timeout | timeout | timeout | timeout | timeout | timeout | timeout | timeout | timeout |
| B2324 | timeout | timeout | timeout | timeout | timeout | timeout | timeout | timeout | timeout |
| S2122 | 2.00s | 0.21s | 0.42s | 2.00s | 0.48s | 0.28s | 2.00s | 0.35s | 0.40s |
| S2123 | 1m 30s | 38.95s | 1m 36s | 1m 36s | 1m 10s | 1m 7s | 1m 33s | 59.63s | 51.54s |
| S2133 | timeout | 4m 26s | 10m 23s | timeout | 7m 27s | 6m 17s | timeout | 12m 11s | 5m 57s |
| S2254 | timeout | timeout | timeout | timeout | timeout | timeout | timeout | timeout | timeout |
| S2264 | timeout | timeout | timeout | timeout | timeout | timeout | timeout | timeout | timeout |
| RS212 | 2.00s | 0.01s | 0.01s | 2.00s | 0.01s | 0.01s | 2.00s | 0.01s | 0.00s |
| RS213 | 6m 51s | 10.24s | 18.82s | 6m 27s | 12.83s | 24.61s | 6m 25s | 8.74s | 8.18s |
| RS222 | 3.00s | 0.01s | 0.01s | 3.00s | 0.02s | 0.02s | 3.00s | 0.01s | 0.01s |
| RS223 | 8m 41s | 5.79s | 9.71s | 8m 24s | 8.45s | 12.94s | 8m 54s | 4.00s | 4.54s |

## H.1.5 Results for NFCCERT solution concept

| Game | Maximize Pl. 1's utility | | | Maximize Pl. 2's utility | | | Maximize social welfare | | |
|---|---|---|---|---|---|---|---|---|---|
| | LP | Lagrangian | Bin.Search | LP | Lagrangian | Bin.Search | LP | Lagrangian | Bin.Search |
| B2222 | 0.00s | 0.00s | 0.00s | 0.00s | 0.00s | 0.00s | 0.00s | 0.00s | 0.00s |
| B2322 | 0.00s | 0.02s | 0.03s | 0.00s | 0.02s | 0.02s | 0.00s | 0.01s | 0.00s |
| B2323 | 2.00s | 0.62s | 0.64s | 1.00s | 0.48s | 0.51s | 2.00s | 0.14s | 0.08s |
| B2324 | 11.00s | 4.88s | 9.86s | 11.00s | 5.24s | 6.26s | 10.00s | 1.82s | 0.70s |
| S2122 | 0.00s | 0.00s | 0.00s | 0.00s | 0.00s | 0.00s | 0.00s | 0.00s | 0.00s |
| S2123 | 0.00s | 0.00s | 0.00s | 0.00s | 0.00s | 0.00s | 0.00s | 0.00s | 0.00s |
| S2133 | 0.00s | 0.01s | 0.00s | 0.00s | 0.01s | 0.01s | 0.00s | 0.01s | 0.00s |
| S2254 | 25.00s | 1.23s | 0.84s | 24.00s | 3.02s | 2.41s | 28.00s | 1.32s | 0.70s |
| S2264 | 56.00s | 2.71s | 1.72s | 42.00s | 5.43s | 3.81s | 50.00s | 2.73s | 1.24s |
| U212 | 0.00s | 0.00s | 0.00s | 0.00s | 0.00s | 0.00s | 0.00s | 0.00s | 0.00s |
| U213 | 0.00s | 0.00s | 0.00s | 0.00s | 0.00s | 0.00s | 0.00s | 0.00s | 0.00s |
| U222 | 0.00s | 0.00s | 0.00s | 0.00s | 0.00s | 0.00s | 0.00s | 0.00s | 0.00s |
| U223 | 0.00s | 0.00s | 0.01s | 0.00s | 0.00s | 0.00s | 0.00s | 0.00s | 0.00s |

## H.1.6 Results for CCERT solution concept

| Game | Maximize Pl. 1's utility | | | Maximize Pl. 2's utility | | | Maximize social welfare | | |
|---|---|---|---|---|---|---|---|---|---|
| | LP | Lagrangian | Bin.Search | LP | Lagrangian | Bin.Search | LP | Lagrangian | Bin.Search |
| B2222 | 0.00s | 0.00s | 0.01s | 0.00s | 0.01s | 0.00s | 0.00s | 0.01s | 0.00s |
| B2322 | 0.00s | 0.07s | 0.11s | 0.00s | 0.15s | 0.09s | 0.00s | 0.22s | 0.09s |
| B2323 | 7.00s | 3.75s | 3.55s | 6.00s | 4.69s | 3.54s | 8.00s | 5.39s | 2.64s |
| B2324 | timeout | 54.92s | 49.66s | timeout | 59.02s | 59.88s | timeout | 1m 31s | 37.51s |
| S2122 | 0.00s | 0.00s | 0.00s | 0.00s | 0.00s | 0.00s | 0.00s | 0.00s | 0.00s |
| S2123 | 0.00s | 0.01s | 0.02s | 0.00s | 0.02s | 0.02s | 0.00s | 0.01s | 0.02s |
| S2133 | 1.00s | 0.04s | 0.04s | 1.00s | 0.08s | 0.06s | 1.00s | 0.05s | 0.03s |
| S2254 | 1m 41s | 8.61s | 11.29s | 1m 47s | 23.28s | 16.40s | 2m 3s | 8.22s | 16.31s |
| S2264 | timeout | 20.11s | 21.42s | timeout | 1m 5s | 38.76s | timeout | 16.50s | 20.62s |
| RS212 | 0.00s | 0.00s | 0.00s | 0.00s | 0.00s | 0.00s | 0.00s | 0.00s | 0.00s |
| RS213 | 0.00s | 0.01s | 0.00s | 0.00s | 0.01s | 0.01s | 0.00s | 0.01s | 0.00s |
| RS222 | 0.00s | 0.00s | 0.00s | 0.00s | 0.00s | 0.00s | 0.00s | 0.00s | 0.00s |
| RS223 | 0.00s | 0.01s | 0.01s | 0.00s | 0.01s | 0.01s | 0.00s | 0.01s | 0.00s |

### H.1.7 Results for CERT solution concept

| Game | Maximize Pl. 1's utility | | | Maximize Pl. 2's utility | | | Maximize social welfare | | |
|------|------|------------|------------|------|------------|------------|------|------------|------------|
|      | LP | Lagrangian | Bin.Search | LP | Lagrangian | Bin.Search | LP | Lagrangian | Bin.Search |
| B2222 | 0.00s | 0.01s | 0.01s | 0.00s | 0.02s | 0.01s | 0.00s | 0.02s | 0.02s |
| B2322 | 2.00s | 1.05s | 0.95s | 2.00s | 1.16s | 0.66s | 2.00s | 1.24s | 1.20s |
| B2323 | 40.00s | 47.11s | 21.54s | 33.00s | 58.20s | 22.16s | 37.00s | 40.45s | 59.67s |
| B2324 | timeout | 8m 29s | 4m 38s | timeout | timeout | 3m 40s | timeout | 6m 14s | 4m 1s |
| S2122 | 0.00s | 0.02s | 0.01s | 0.00s | 0.02s | 0.01s | 0.00s | 0.02s | 0.01s |
| S2123 | 1.00s | 0.19s | 0.09s | 1.00s | 0.28s | 0.20s | 1.00s | 0.15s | 0.12s |
| S2133 | 3.00s | 0.96s | 0.43s | 2.00s | 1.10s | 0.81s | 2.00s | 0.92s | 0.41s |
| S2254 | timeout | 3m 26s | 2m 54s | timeout | 6m 15s | 4m 14s | timeout | 2m 42s | 2m 53s |
| S2264 | timeout | timeout | timeout | timeout | timeout | timeout | timeout | timeout | timeout |
| RS212 | 0.00s | 0.00s | 0.00s | 0.00s | 0.00s | 0.00s | 0.00s | 0.00s | 0.00s |
| RS213 | 0.00s | 0.02s | 0.03s | 0.00s | 0.02s | 0.04s | 0.00s | 0.02s | 0.01s |
| RS222 | 0.00s | 0.00s | 0.00s | 0.00s | 0.00s | 0.00s | 0.00s | 0.00s | 0.00s |
| RS223 | 1.00s | 0.01s | 0.02s | 1.00s | 0.02s | 0.02s | 1.00s | 0.01s | 0.01s |

## H.2 Detailed Breakdown by Equilibrium and Objective Function (Three-Player Games)

### H.2.1 Results for NFCCE solution concept

| Game | Maximize Pl. 1's utility | | | Maximize Pl. 2's utility | | | Maximize Pl. 3's utility | | |
|------|------|------------|------------|------|------------|------------|------|------------|------------|
|      | LP | Lagrangian | Bin.Search | LP | Lagrangian | Bin.Search | LP | Lagrangian | Bin.Search |
| D32 | 0.00s | 0.01s | 0.02s | 0.00s | 0.01s | 0.02s | 0.00s | 0.01s | 0.02s |
| D33 | 2m 17s | 12.93s | 21.46s | 2m 8s | 16.40s | 28.63s | 2m 23s | 17.57s | 19.74s |
| GL3 | 0.00s | 0.01s | 0.02s | 0.00s | 0.01s | 0.02s | 0.00s | 0.02s | 0.02s |
| K35 | 49.00s | 0.76s | 1.52s | 55.00s | 0.56s | 1.39s | 55.00s | 0.64s | 1.67s |
| L3132 | 26.00s | 0.59s | 1.72s | 24.00s | 0.79s | 1.63s | 24.00s | 0.77s | 1.91s |
| L3133 | 38.00s | 0.94s | 1.79s | 38.00s | 0.89s | 1.83s | 37.00s | 1.11s | 1.90s |
| L3151 | timeout | 15.12s | 30.01s | timeout | 12.74s | 35.58s | timeout | 18.03s | 30.87s |
| L3223 | 4.00s | 0.44s | 0.91s | 4.00s | 0.45s | 1.84s | 4.00s | 0.52s | 1.58s |
| L3523 | timeout | 1m 7s | 4m 21s | timeout | 1m 2s | 4m 27s | timeout | 1m 9s | 4m 35s |
| TP3 | 1m 38s | 7.44s | 8.43s | 1m 40s | 7.63s | 9.41s | 1m 44s | 11.45s | 11.85s |

### H.2.2 Results for EFCCE solution concept

| Game | Maximize Pl. 1's utility | | | Maximize Pl. 2's utility | | | Maximize Pl. 3's utility | | |
|------|------|------------|------------|------|------------|------------|------|------------|------------|
|      | LP | Lagrangian | Bin.Search | LP | Lagrangian | Bin.Search | LP | Lagrangian | Bin.Search |
| D32 | 0.00s | 0.02s | 0.03s | 0.00s | 0.01s | 0.03s | 0.00s | 0.02s | 0.03s |
| D33 | timeout | 1m 46s | 2m 32s | timeout | 1m 16s | 2m 28s | timeout | 1m 44s | 3m 7s |
| GL3 | 1.00s | 0.02s | 0.04s | 1.00s | 0.03s | 0.05s | 1.00s | 0.03s | 0.05s |
| K35 | 46.00s | 0.67s | 2.09s | 55.00s | 0.75s | 1.90s | 51.00s | 0.68s | 1.85s |
| L3132 | 8m 43s | 5.13s | 18.75s | 9m 17s | 6.27s | 13.48s | 9m 44s | 7.76s | 18.47s |
| L3133 | 20m 26s | 8.88s | 19.52s | timeout | 8.19s | 23.17s | 23m 15s | 10.86s | 27.48s |
| L3151 | timeout | timeout | timeout | timeout | timeout | timeout | timeout | timeout | timeout |
| L3223 | 1m 10s | 2.94s | 14.79s | 1m 10s | 3.22s | 12.23s | 1m 2s | 3.24s | 13.68s |
| L3523 | timeout | timeout | timeout | timeout | timeout | timeout | timeout | timeout | timeout |
| TP3 | timeout | 13.76s | 15.36s | timeout | 15.03s | 13.64s | timeout | 15.27s | 15.10s |

### H.2.3 Results for EFCE solution concept

| Game | Maximize Pl. 1's utility | | | Maximize Pl. 2's utility | | | Maximize Pl. 3's utility | | |
| | LP | Lagrangian | Bin.Search | LP | Lagrangian | Bin.Search | LP | Lagrangian | Bin.Search |
|---|---|---|---|---|---|---|---|---|---|
| D32 | 12.00s | 0.40s | 1.11s | 11.00s | 0.35s | 1.32s | 10.00s | 0.66s | 1.23s |
| D33 | timeout | timeout | timeout | timeout | timeout | timeout | timeout | timeout | timeout |
| GL3 | 0.00s | 0.01s | 0.03s | 0.00s | 0.01s | 0.03s | 0.00s | 0.01s | 0.06s |
| K35 | 57.00s | 0.55s | 1.50s | 55.00s | 1.03s | 2.37s | 60.00s | 1.26s | 2.22s |
| L3132 | 8m 18s | 6.10s | 14.02s | 8m 57s | 7.65s | 16.25s | 7m 35s | 6.78s | 19.11s |
| L3133 | 21m 25s | 6.84s | 25.83s | 21m 43s | 10.76s | 26.37s | 19m 58s | 10.28s | 20.65s |
| L3151 | timeout | timeout | timeout | timeout | timeout | timeout | timeout | timeout | timeout |
| L3223 | 2m 2s | 5.52s | 28.79s | 1m 50s | 6.46s | 29.37s | 2m 0s | 5.94s | 17.13s |
| L3523 | timeout | timeout | timeout | timeout | timeout | timeout | timeout | timeout | timeout |
| TP3 | timeout | 13.46s | 13.78s | timeout | 14.25s | 13.71s | timeout | 14.48s | 11.48s |

### H.2.4 Results for COMM solution concept

| Game | Maximize Pl. 1's utility | | | Maximize Pl. 2's utility | | | Maximize Pl. 3's utility | | |
| | LP | Lagrangian | Bin.Search | LP | Lagrangian | Bin.Search | LP | Lagrangian | Bin.Search |
|---|---|---|---|---|---|---|---|---|---|
| D32 | 0.00s | 0.06s | 0.24s | 1.00s | 0.06s | 0.21s | 1.00s | 0.17s | 0.32s |
| D33 | timeout | 4m 37s | 5m 35s | timeout | 1m 31s | 4m 27s | timeout | 2m 38s | 5m 21s |
| GL3 | timeout | 7.72s | 26.48s | timeout | 7.50s | 21.91s | timeout | 11.42s | 25.44s |
| K35 | 1.00s | 0.03s | 0.07s | 1.00s | 0.02s | 0.07s | 1.00s | 0.03s | 0.05s |
| L3132 | 8.00s | 3.46s | 10.64s | 8.00s | 3.37s | 8.44s | 7.00s | 4.02s | 9.57s |
| L3133 | 12.00s | 3.40s | 12.82s | 12.00s | 3.54s | 10.70s | 11.00s | 3.52s | 12.35s |
| L3151 | timeout | 16.73s | 55.51s | timeout | 15.80s | 57.66s | timeout | 18.00s | 56.83s |
| L3223 | 19.00s | 18.19s | 1m 0s | 18.00s | 15.30s | 1m 17s | 21.00s | 18.77s | 57.11s |
| L3523 | timeout | timeout | timeout | timeout | timeout | timeout | timeout | timeout | timeout |
| TP3 | timeout | timeout | timeout | timeout | timeout | timeout | timeout | timeout | timeout |

### H.2.5 Results for NFCCERT solution concept

| Game | Maximize Pl. 1's utility | | | Maximize Pl. 2's utility | | | Maximize Pl. 3's utility | | |
| | LP | Lagrangian | Bin.Search | LP | Lagrangian | Bin.Search | LP | Lagrangian | Bin.Search |
|---|---|---|---|---|---|---|---|---|---|
| D32 | 0.00s | 0.00s | 0.00s | 0.00s | 0.00s | 0.00s | 0.00s | 0.00s | 0.00s |
| D33 | 0.00s | 0.04s | 0.05s | 0.00s | 0.06s | 0.05s | 0.00s | 0.04s | 0.03s |
| GL3 | 0.00s | 0.00s | 0.01s | 0.00s | 0.00s | 0.01s | 0.00s | 0.00s | 0.01s |
| K35 | 0.00s | 0.00s | 0.00s | 0.00s | 0.00s | 0.00s | 0.00s | 0.00s | 0.00s |
| L3132 | 0.00s | 0.02s | 0.03s | 0.00s | 0.02s | 0.02s | 0.00s | 0.02s | 0.03s |
| L3133 | 0.00s | 0.02s | 0.03s | 0.00s | 0.02s | 0.04s | 0.00s | 0.02s | 0.04s |
| L3151 | 0.00s | 0.03s | 0.04s | 0.00s | 0.04s | 0.07s | 0.00s | 0.03s | 0.07s |
| L3223 | 0.00s | 0.03s | 0.06s | 0.00s | 0.04s | 0.05s | 0.00s | 0.05s | 0.08s |
| L3523 | 13.00s | 7.41s | 13.51s | 15.00s | 8.17s | 16.32s | 17.00s | 10.60s | 22.93s |
| TP3 | 47.00s | 4.56s | 4.98s | 48.00s | 4.97s | 8.12s | 45.00s | 7.38s | 7.50s |

### H.2.6 Results for CCERT solution concept

| Game | Maximize Pl. 1's utility | | | Maximize Pl. 2's utility | | | Maximize Pl. 3's utility | | |
|------|------|-----------|------------|------|-----------|------------|------|-----------|------------|
| | LP | Lagrangian | Bin.Search | LP | Lagrangian | Bin.Search | LP | Lagrangian | Bin.Search |
| D32 | 0.00s | 0.00s | 0.00s | 0.00s | 0.00s | 0.00s | 0.00s | 0.00s | 0.00s |
| D33 | 1.00s | 0.20s | 0.27s | 1.00s | 0.13s | 0.32s | 1.00s | 0.19s | 0.25s |
| GL3 | 0.00s | 0.01s | 0.02s | 0.00s | 0.01s | 0.02s | 0.00s | 0.01s | 0.01s |
| K35 | 0.00s | 0.00s | 0.00s | 0.00s | 0.00s | 0.00s | 0.00s | 0.00s | 0.00s |
| L3132 | 0.00s | 0.05s | 0.08s | 0.00s | 0.05s | 0.09s | 0.00s | 0.05s | 0.07s |
| L3133 | 0.00s | 0.04s | 0.13s | 0.00s | 0.08s | 0.20s | 0.00s | 0.12s | 0.09s |
| L3151 | 1.00s | 0.12s | 0.36s | 1.00s | 0.16s | 0.32s | 1.00s | 0.16s | 0.31s |
| L3223 | 1.00s | 0.23s | 0.72s | 1.00s | 0.26s | 0.58s | 1.00s | 0.24s | 0.67s |
| L3523 | timeout | 1m 17s | 13m 19s | timeout | 1m 16s | 16m 19s | timeout | 1m 13s | 6m 41s |
| TP3 | timeout | 10.23s | 16.79s | timeout | 10.38s | 12.96s | timeout | 11.95s | 11.91s |

### H.2.7 Results for CERT solution concept

| Game | Maximize Pl. 1's utility | | | Maximize Pl. 2's utility | | | Maximize Pl. 3's utility | | |
|------|------|-----------|------------|------|-----------|------------|------|-----------|------------|
| | LP | Lagrangian | Bin.Search | LP | Lagrangian | Bin.Search | LP | Lagrangian | Bin.Search |
| D32 | 0.00s | 0.01s | 0.01s | 0.00s | 0.01s | 0.01s | 0.00s | 0.01s | 0.02s |
| D33 | 4.00s | 3.14s | 7.11s | 4.00s | 2.02s | 2.26s | 4.00s | 2.10s | 9.88s |
| GL3 | 0.00s | 0.02s | 0.04s | 0.00s | 0.02s | 0.05s | 0.00s | 0.03s | 0.05s |
| K35 | 0.00s | 0.01s | 0.01s | 0.00s | 0.00s | 0.01s | 0.00s | 0.00s | 0.01s |
| L3132 | 1.00s | 0.10s | 0.28s | 1.00s | 0.18s | 0.37s | 0.00s | 0.18s | 0.35s |
| L3133 | 1.00s | 0.22s | 0.41s | 1.00s | 0.19s | 0.44s | 1.00s | 0.25s | 0.30s |
| L3151 | 2.00s | 0.21s | 0.86s | 2.00s | 0.22s | 0.75s | 2.00s | 0.29s | 0.72s |
| L3223 | 1.00s | 0.61s | 1.48s | 1.00s | 0.61s | 1.65s | 1.00s | 0.68s | 1.95s |
| L3523 | timeout | 2m 58s | timeout | timeout | 4m 33s | timeout | timeout | 3m 55s | timeout |
| TP3 | timeout | 26.70s | 34.23s | timeout | 25.08s | 28.22s | timeout | 36.36s | 27.68s |

