# OpenReview forum: "Computing Optimal Equilibria and Mechanisms via Learning in Zero-Sum Extensive-Form Games"
_NeurIPS.cc/2023/Conference — NeurIPS 2023 poster_

### Official Review · Reviewer_dGJ5 · 2023-07-04

**Soundness:** 3 good
**Presentation:** 3 good
**Contribution:** 3 good
**Rating:** 7
**Confidence:** 3

**Summary:**

The authors proposed a learning-based framework to solve for optimal equilibria in n-player general-sum extensive-form games. The key idea is to leverage Lagrangian relaxation and convert such games to 2-player zero-sum extensive-form games which can then be solved using known learning methods that solve for minimax equilibria.

**Strengths:**

* significance
I believe this work could be a significant contribution to the field as it proposes a computationally practical method that solves for optimal equilibria in n-player general-sum games. This is significant as it offers a language to speak about optimality of equilibria in addition to stability and could make game theoretic methods attractive in wider classes of practical applications.

* originality
I believe the proposed method is novel.

* clarity
The paper is well written and relatively easy to follow.

* quality
The theoretical derivation of the method appears sound; the empirical results seem extensive though it would be interesting to understanding additional details which I have commented below.

**Weaknesses:**

The presentation of the empirical results seems to be primarily focused on the efficiency and feasibility of converting the problem of solving for optimal equilibria in n-player general-sum EFGs to solving minimax equilibria in two-player zero-sum games. For instance Table 1 presented runtime results of the learning-based method compared to solving LP directly.

Would it be possible to provide additional details on optimality as well especially in general-sum games such as Sheriff where we know the value of max-welfare equilibria (e.g. https://www.cs.cmu.edu/~gfarina/2020/efcce-aaai20/coarse-correlation.aaai20.pdf)? Would the proposed method scale up to these instances of sheriff game?

**Questions:**

I find the presentation relatively easy to follow. Some minor clarifying questions below:

L42: "... including information design, and solution concepts such as ....". This statement appears several times but it is a bit confusing to follow. I can understand the statement after reading the paper in full but consider rephrasing it or splitting it up to make it more readable?

L63 nit: clarify what $T$ stands for.

L145: I find the discussion on the revelation principle interesting but it's not clear how general it would apply to games. Could you comment on this and if the set of benchmark games studied in this work would satisfy this property?

L191: "All players, ... are constrained to [act/play, missing] according to $d_i$..." is mediator considered to be included in "All players" in this statement? If so, what does playing according to $d_i$ mean for the mediator? Is the mediator limited to deterministic policies in this two-player game?

L192: "If nature picked ..., the utility is ...", --> "the utility [to the mediator] is ..."?


**Limitations:**

The authors have discussed practical limitations of the method in snippets of the main text.

---

> ### Author Rebuttal · Authors · 2023-08-10
>
> Thanks for the review!
>
> > Would it be possible to provide additional details on optimality as well especially in general-sum games such as Sheriff where we know the value of max-welfare equilibria (e.g. https://www.cs.cmu.edu/~gfarina/2020/efcce-aaai20/coarse-correlation.aaai20.pdf)?
>
> Sheriff games are known to be triangle-free, since they are two-player games with no chance moves. For this class of games, the polytope of all EFCE can be described using a polynomial number of constraints. Both our paper and the paper by Farina et al. that you linked are able to take advantage of this fact and lead to a polynomial-time optimization algorithm. For our algorithm, the majority of the time is spent allocating memory for constructing the internal representation of the set of equilibria used in the learning procedure. We similarly guess that the algorithm by Farina et al. a large fraction of time is spent computing the constraints that define the same set, though expressed as linear constraints passed to the linear programming solver.
>
> As an example, we run our framework on one of the large Sheriff instances used by Farina et al. In particular, we employed the instance S2-20-5-4 (line 4 of Table 1 in the paper linked). Computing a social-welfare-maximizing NFCCE using our framework requires a learning phase of 110 sec (3 minutes if we consider also the time to construct the representation). The paper by Farina et al. requires around 13min to solve the same instance. On the same instance, we compute a SW-maximizing EFCCE in 99 seconds (6 minutes if we consider the construction of the internal representation), while Farina et al. report an execution time of 1h34min. For both solution concepts, we obtain a final value matching exactly the optimal social welfare reported in Table 1 of Farina et al.
>
> We will include additional experiments on the exact instances of Farina et al. in the final version.
>
>
> *On the questions raised by the reviewer (several of which are helpful reformulations):*
>
> Re L42: We agree. We will split the sentence into two sentences. The first sentence introduces the solution concepts that we cover, the second information and mechanism design with a separate footnote outlining the construction around L153.
>
> Re L63: We will clarify that this refers to the time horizon, i.e., the number of iterations of the algorithm.
>
> Re L145: The revelation principle depends on an equilibrium/”implementation” concept. Hence, it depends on which equilibrium we consider. In particular, our revelation principle applies to all games we consider with the equilibrium concepts we discuss in L51-54. See also the response to JN1n for more discussion of the revelation principle.
>
> Re L191: According to footnote 1, we refer to the mediator only by “agent”, not “player”. Hence “all players” does not include the mediator.
>
> Re L192: Thank you. This is correct, and we will make this change.

---

> > ### Comment · Reviewer_dGJ5 · 2023-08-16
> >
> > Thank you for the rebuttals.
> >
> > > For both solution concepts, we obtain a final value matching exactly the optimal social welfare reported in Table 1 of Farina et al.
> > I think it would be beneficial for the community to see reference results on optimal equilibria solutions in these games. Would it be possible to include them in the appendix?
> >
> > Good work! I will keep my recommendation for accepting.

---

> > > ### Author Response · Authors · 2023-08-16
> > >
> > > Yes, we'll do this in the final version. Thank you for the suggestion.

---

### Official Review · Reviewer_JN1n · 2023-07-04

**Soundness:** 3 good
**Presentation:** 3 good
**Contribution:** 3 good
**Rating:** 6
**Confidence:** 4

**Summary:**

The paper focuses on addressing the challenge of computing optimal equilibria in extensive-form games. The authors introduce the revelation principle, which transforms the problem into a linear programming (LP) task.

They propose using Lagrange relaxations to solve the LP, treating the resulting saddle-point problem as a Nash equilibrium in a zero-sum two-player game. The authors explicitly construct such zero-sum game. To efficiently find the solution, the authors employ regret minimization over conic hulls. Additionally, they highlight the flexibility of their approach by showing that other algorithms for zero-sum two-player games can be utilized, offering different convergence rate guarantees or achieving last-iteration convergence.

The paper concludes with thorough experimental evaluations, demonstrating the effectiveness and scalability of their proposed methods across various simple games.

**Strengths:**

- The paper addresses the significant problem of computing optimal equilibria in game theory. By reducing this problem to zero-sum games, the authors provide a framework for finding optimal equilibria. The key contribution lies in the fact that existing algorithms for zero-sum extensive form games can be directly applied to this problem. This reduction greatly enhances the understanding and applicability of the proposed approach.

- The experimental evaluation conducted by the authors is robust and contributes to the strength of their work. They provide evidence of the scalability of their methods through preliminary experiments. This not only showcases the effectiveness of the proposed techniques but also demonstrates their potential for real-world applications. The inclusion of experimental results further reinforces the practical relevance of the research and adds value to the overall paper.

**Weaknesses:**

My only concern is about the revelation principle. The reduction in the paper relies on the revelation principle, which is a fundamental concept. However, it is not clear regarding the specific conditions under which a fixed pure strategy d_i exists for different players and within which games and equilibria this concept applies.  Additionally, how to effectively find such d_i strategies is also a problem.

**Questions:**

- See the above weakness.

- What does 'optimal direct equilibria' mean in line 162. Moerover, if the revelation principle fails, there's no fixed pure strategy d_i. In this case, how to get a similar problem of (G).

- In the experiment of auction design, to demonstrate scalability, is it possible to conduct an experiment if each bidder's valuation is sampled from uniform distribution in [0,1].

**Limitations:**

The authors have partially addressed the limitations of their work, though there is space for improvement (see the section Weaknesses and Questions).

---

> ### Author Rebuttal · Authors · 2023-08-10
>
> Thanks for the review!
>
> > My only concern is about the revelation principle. The reduction in the paper relies on the revelation principle, which is a fundamental concept. However, it is not clear regarding the specific conditions under which a fixed pure strategy d_i exists for different players and within which games and equilibria this concept applies. Additionally, how to effectively find such d_i strategies is also a problem.
>
> The revelation principle holds for all the equilibrium concepts referenced in the paper, and indeed in all of those cases, the direct strategy d_i is trivial to define (report honest information and play recommended actions).
>
> > What does 'optimal direct equilibria' mean in line 162. Moerover, if the revelation principle fails, there's no fixed pure strategy d_i. In this case, how to get a similar problem of (G).
>
> The pure strategy d_i can be defined regardless of whether the revelation principle holds: it is, as above, simply “report honest information and play recommended actions”. An “optimal direct equilibrium” is an equilibrium that is optimal among direct equilibria, that is, equilibria of the form (μ, d). Without the revelation principle, such equilibria can still exist, and our method will still recover the optimal such equilibrium (if one exists). However, without the revelation principle, there would be no guarantee that the optimal direct equilibrium will exist or that it will also be the (overall) optimal equilibrium, as there could be an indirect equilibrium that is better. We leave the problem of generalizing our methods to such cases as an interesting future research direction.
>
> > In the experiment of auction design, to demonstrate scalability, is it possible to conduct an experiment if each bidder's valuation is sampled from uniform distribution in [0,1].
>
> Thank you for this experiment idea. While our current paper's scope does not include such extensive experimentation, we agree that this experiment could provide additional evidence for our method being scalable in future work. We believe that our existing methodology adequately supports our findings, but this additional analysis could be a valuable extension in subsequent research that scales our method to much larger games.

---

> > ### Comment · Reviewer_JN1n · 2023-08-20
> >
> > Thank you for answering my questions.

---

### Official Review · Reviewer_c3NQ · 2023-07-05

**Soundness:** 3 good
**Presentation:** 3 good
**Contribution:** 3 good
**Rating:** 5
**Confidence:** 2

**Summary:**

This paper studies the computation of optimal equilibria in multi-player extensive-form games via no-regret learning algorithms. The key idea is to take the constrained LP formulation of the optimal equilibrium problem proposed by Zhang and Sandholm and consider the saddle point formulation, which can then be solved using no-regret learning algorithms. In order to alleviate the large $\lambda^*$, a binary-search based algorithm is proposed. Finally, the algorithms are compared with a LP-based method on several tabular games as well as for an auction-design problem.

**Strengths:**

- The paper is well-written. The results presented clearly and the writing is easy to follow.

- The experimental results seem compelling as it has an order of magnitude advantage in wall-clock time over the LP-based method. This advantage is consistent across a set of diverse instances.

**Weaknesses:**

- I have some concerns about the LP formulation (G). Consider a $N$ player normal form game with $A$ actions per player for instance; it seems that for the correlated equilibrium in such a game, the $\mu$ in (G) would need to have dimension $A^N$. However, no swap-regret algorithms can find an approximate correlated equilibrium with $\tilde{O}(NA^2/\epsilon^2)$ samples. It seems that the LP formulation (G) can lead to suboptimal dependence on the number of players.

- Perhaps related the previous point, while in Line 79 the authors claim that "Our algorithm significantly outperforms existing LP-based methods", in the experiments it is only compared against the LP based solution to (G) proposed by Zhagn and Sandholm. Is (G) the only known LP formulation of the optimal equilibrium problem? If not, wouldn't it make sense to compare against those as well?

**Questions:**

Have the authors tried gradient based methods (such as extra-gradient or gradient descent-ascent) on the saddle-point problem (L1)?

**Limitations:**

This paper does not have foreseeable negative societal impact.

---

> ### Author Rebuttal · Authors · 2023-08-10
>
> Thanks for the review!
>
> > I have some concerns about the LP formulation (G).
>
> There is an important detail that might have been missed here. The cited no-swap-regret dynamics compute one equilibrium, but this paper is concerned with optimizing over the space of equilibria (for example, to compute a social-welfare-maximizing equilibrium). The latter problem is harder than the former, and explains the difference in dependence.
>
> > Perhaps related the previous point, while in Line 79 the authors claim that "Our algorithm significantly outperforms existing LP-based methods", in the experiments it is only compared against the LP based solution to (G) proposed by Zhagn and Sandholm. Is (G) the only known LP formulation of the optimal equilibrium problem? If not, wouldn't it make sense to compare against those as well?
>
> Yes, the LP of Zhang and Sandholm is, to our knowledge, the only LP formulation of the optimal equilibrium problem in its full generality. That said, for several of the concepts (e.g., mechanism design in single-stage settings [2]), the LP of Zhang and Sandholm is equivalent to previously-known techniques. The paper of Zhang and Sandholm itself contains more details.
>
> [2] V Conitzer and T Sandholm, “Complexity of mechanism design”, UAI 2002
>
> > Have the authors tried gradient based methods (such as extra-gradient or gradient descent-ascent) on the saddle-point problem (L1)?
>
> CFR itself can be viewed as a gradient-based method. Other gradient-based methods, such as those that you mention, certainly can be used to solve any zero-sum game, including (L1). In practice, there is a long line of research (e.g., [3, 4] and references therein) that consistently establishes the CFR variants that we chose as the fastest methods of solving large tabular extensive-form games, so in our experiments we focus on these methods.
>
> [3] G Farina, C Kroer, and T Sandholm, “Faster Game Solving via Predictive Blackwell Approachability: Connecting Regret Matching and Mirror Descent”, AAAI 2021
>
> [4] N Brown and T Sandholm, “Solving Imperfect-Information Games via Discounted Regret Minimization”, AAAI 2019

---

### Official Review · Reviewer_dPXL · 2023-07-08

**Soundness:** 3 good
**Presentation:** 3 good
**Contribution:** 3 good
**Rating:** 6
**Confidence:** 3

**Summary:**

This paper presents a novel approach to compute optimal equilibria in multi-player extensive-form games through the use of Lagrangian relaxation as a two-player zero-sum extensive-form game. Building upon the mediator augmentation game framework, the proposed computing approach significantly contributes to the reduction of zero-sum games, thereby holding substantial implications for mechanism design and information design. Furthermore, it plays a pivotal role in fostering a well-balanced hierarchy of concepts.



**Strengths:**

-- The investigation of computing minimax equilibria in extensive-form games is an interesting and well-motivated problem.

-- Most correlation equilibrium notions pose challenges in finding optimal equilibria, whereas the framework presented in this paper offers a learning-based algorithm to compute them.



**Weaknesses:**

-- The paper lies heavily on notation that lacks intuitive explanations. To enhance reader understanding, a schematic representation of the mediator-augmented game framework should be included in the main body.

-- The approach proposed in this paper resembles a transformation of the original game into a mediator-led Stackerberg game. It would be beneficial to discuss and compare existing research on equilibrium solutions for zero-sum games in Stackelberg games to provide a contextual background for the paper's contribution. Lack controlled experiments involving additional algorithms for equilibrium solutions in Stackelberg games.


**Questions:**

Why does the space of messages can be restricted by using the revelation principle? I think more details on this are necessary in the paper.

---

> ### Author Rebuttal · Authors · 2023-08-10
>
> Thanks for the review!
>
> > The paper lies heavily on notation that lacks intuitive explanations. To enhance reader understanding, a schematic representation of the mediator-augmented game framework should be included in the main body.
>
> Thank you for the suggestion; we will incorporate this and other clarity recommendations from the other reviews in the final version.
>
> > The approach proposed in this paper resembles a transformation of the original game into a mediator-led Stackerberg game. It would be beneficial to discuss and compare existing research on equilibrium solutions for zero-sum games in Stackelberg games to provide a contextual background for the paper's contribution. Lack controlled experiments involving additional algorithms for equilibrium solutions in Stackelberg games.
>
> Stackelberg equilibria in extensive-form games are hard to find in general [1]. Our Stackelberg game has a much nicer form than general Stackelberg games—in particular, we know in advance what the equilibrium strategies will be for the followers (namely, the direct strategies, $d_i$). This observation is what allows the reduction to zero-sum games, sidestepping the need to use Stackleberg-specific technology or solvers. We will expand the discussion about the relationship with Stackelberg games in Appendix B (L722) to include this and make sure it is comprehensive.
>
> [1] J Letchford and V Conitzer “Computing Optimal Strategies to Commit to in Extensive-Form Games”, EC 2010
>
> > Why does the space of messages can be restricted by using the revelation principle? I think more details on this are necessary in the paper.
>
> The *purpose* of the revelation principle is precisely to reduce the message space: the revelation principle states that, without loss of generality, players in equilibrium will send their true information and obey action recommendations. This allows a reduction of the message space, because then we can assume that messages to the mediator are information reports, and messages from the mediator are action recommendations. We will add a note about this in the final version.

---

> > ### Comment · Reviewer_dPXL · 2023-08-19
> >
> > Thank you for your response.

---

### Decision · Program_Chairs · 2023-09-21

**Decision:**

Accept (poster)

**Comment:**

A new approach is proposed for computing (correlated, communication, and certification) equilibria in zero-sum extensive-form games. It is proven that the no-regret dynamics converge to equilibrium in both average-iterate and final-iterate senses. The key idea is the mediator-augmented transformation, which gives rise to the saddle-point formulation that can be solved by online learning algorithms. The reduction approach is presented clearly and appears to be novel (although it builds on existing LP formulations). Reviewers reached a consensus on the acceptance.